# Refining Dual Spectral Sparsity in Transformed Tensor Singular Values

**Andong Wang** [1]   **Yuning Qiu** [1]   **Haonan Huang** [1]   **Zhong Jin** [2]   **Guoxu Zhou** [3 4]   **Qibin Zhao** [1]

## Abstract

The Tubal Nuclear Norm (TNN), derived from the tensor Singular Value Decomposition (t-SVD), is a widely used low-rank modeling tool that promotes sparsity of frequency-domain singular values. However, as a direct extension of the matrix nuclear norm, TNN applies a uniform element-wise penalty to transformed singular values, without explicitly distinguishing sparsity across frequency components from low-rankness within each component. This can be restrictive for real-world tensor data that exhibit multi-level spectral structures, where spectral energy is concentrated in a subset of frequency components while active components remain low-rank. To overcome this limitation, we propose the tensor $\ell_p$-Schatten-$q$ quasi-norm ($p, q \in (0, 1]$), which enables explicit control of dual spectral sparsity by jointly regularizing inter-frequency sparsity and intra-frequency low-rankness. This formulation includes TNN as a special case and subsumes several existing tensor regularizers by coupling global frequency sparsity with local spectral low-rankness, yielding a more flexible modeling principle. We establish minimax error bounds under the proposed dual spectral sparsity model, develop a reweighted optimization algorithm for the resulting nonconvex problem, and demonstrate its effectiveness and robustness on noisy and Poisson tensor completion as well as image clustering tasks.

[1]RIKEN AIP, Tokyo, Japan. [2] Department of Computer, China University of Petroleum – Beijing at Karamay, Karamay, China. [3]School of Automation, Guangdong University of Technology, Guangzhou, China. [4]Key Laboratory of Intelligent Detection and the Internet of Things in Manufacturing, Ministry of Education, Guangdong University of Technology, Guangzhou, China. Correspondence to: Qibin Zhao <qibin.zhao@riken.jp>.

*Proceedings of the 43$^{rd}$ International Conference on Machine Learning*, Seoul, South Korea. PMLR 306, 2026. Copyright 2026 by the author(s).

## 1. Introduction

Identifying and modeling latent structural patterns in high-dimensional signals is a fundamental challenge across domains such as machine learning and signal processing (Liu et al., 2020; Wang et al., 2023c; 2025a; Li et al., 2019b). Real-world datasets are often inherently multi-way, containing intricate dependencies that cannot be adequately captured by vector- or matrix-based representations (Cichocki et al., 2016). A common strategy to uncover these dependencies is to impose a *low-rank* prior, which extracts latent factors and reduces the degrees of freedom, focusing on the principal components of the signal (Martin et al., 2013; Bergqvist & Larsson, 2010). Traditional tensor decompositions, such as CANDECOMP/PARAFAC (CP) (Carroll & Chang, 1970), Tucker (Tucker, 1966), and Tensor Train (Oseledets, 2011), have been widely used to model multi-way signals (Cichocki et al., 2016; Liu et al., 2013; Imaizumi et al., 2017; Yuan & Zhang, 2016). While effective in many settings, these methods primarily exploit low-rankness in the original tensor domain. For complex real-world data, a suitable transform can often make the underlying correlations more compact and easier to model (Li et al., 2019a; Luo et al., 2022a;b; Wang et al., 2025d). This has motivated *transformed-domain* low-rank models, where linear transformations such as the Discrete Fourier Transform (DFT) are used to reveal more pronounced low-rank patterns. Within this paradigm, the tensor Singular Value Decomposition (t-SVD) (Kilmer et al., 2013) has emerged as a powerful framework with notable success in applications such as image and video analysis (Liu et al., 2020; Zhang et al., 2014; Wang et al., 2025c; Zhang & Aeron, 2017).

Building on the t-SVD framework, the Tubal Nuclear Norm (TNN, Definition 2.4) has become an extensively adopted regularizer for low-rank tensor modeling (Zhang et al., 2014; Song et al., 2020; Zhang & Ng, 2021; Lu, 2021; Zhou & Feng, 2017; Qiu et al., 2026; Wang et al., 2023a; 2025b). By extending the matrix nuclear norm to the tensor setting, TNN promotes low-rankness by enforcing *element-wise sparsity* on singular values in the transformed domain (Li et al., 2019a; Zhang et al., 2014). This formulation captures low-rank dependencies within individual frequency components and has proven effective in practical tasks such as hyperspectral data recovery, video denoising, and multi-view clustering (Wang et al., 2025e; Wu, 2024; Peng et al.,

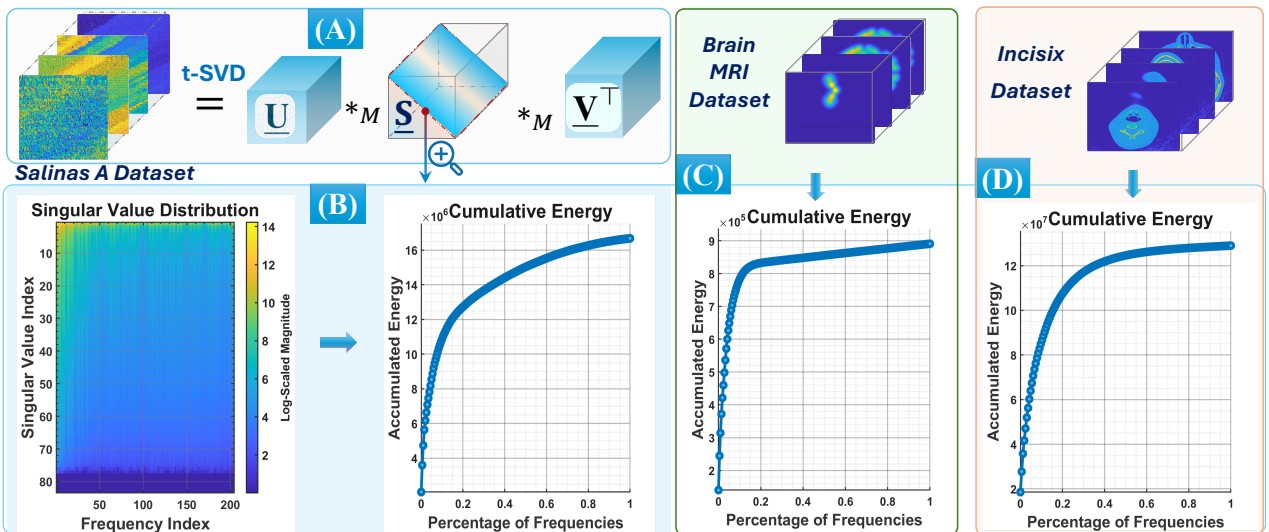

## Most Spectral Energy is Concentrated in Few Frequency Components

*Figure 1.* Empirical illustration of dual spectral sparsity in the transformed domain (DCT) via t-SVD. **(A)** The t-SVD decomposes a tensor into transformed-domain singular structures. **(B)**-Left: Singular value heatmap of the *Salinas A* dataset under t-SVD, with each column represents one frequency slice. Vertical decay reveals intra-frequency low-rankness, while horizontal variation indicates sparsity across frequencies. **(B)**-Right, **(C)**, **(D)**: Cumulative energy curves for *Salinas A*, *Brain MRI*, and *Incisix* datasets show that over 80% of total spectral energy is concentrated in the top 15%–30% frequency components, confirming frequency-wise sparsity.

2025; Zhou et al., 2019; Zheng et al., 2020; 2025; Huang et al., 2025).

**Dual Spectral Sparsity.** Motivated by these observations, we note that TNN sums the singular values of all transformed frontal slices with the same linear penalty. Consequently, a weak frequency slice containing many small singular values and a dominant frequency slice with comparable total singular-value mass are penalized in essentially the same way. This uniform element-wise treatment promotes overall transformed-domain low-rankness, but it does not explicitly separate two different effects: selecting a small number of informative frequency components and enforcing low-rankness within those selected components.

Empirical evidence further shows that transformed tensors tend to exhibit a two-level spectral structure: *most spectral energy is concentrated in a relatively small number of frequency slices, while the active slices are typically low-rank*. This pattern has been observed across a range of datasets, including natural videos, medical imaging volumes (e.g., MRI and CT), hyperspectral cubes, seismic data, and multispectral images. Figure 1 visualizes this behavior, and Tables 4–5 quantify orders-of-magnitude gaps between real tensors and random tensors in both frequency-wise energy concentration and slice-wise effective rank. Formal hypothesis tests in Tables 6 and 7 further confirm complete distributional separation between real data and random baselines. Together, these results suggest that dual spectral sparsity is a widely observed empirical pattern in real tensor data.

This observation motivates a more structured regularization principle beyond the uniform element-wise penalty of TNN, one that explicitly separates inter-frequency sparsity from intra-frequency low-rankness.

**Main Research Questions.** This motivation leads to three central questions:

**RQ1 (Modeling):** *How can a tensor model explicitly represent sparsity across frequency slices and low-rankness within active slices?*

**RQ2 (Theory):** *What statistical guarantees can be established for such a two-level spectral structure?*

**RQ3 (Algorithm):** *Can we develop a tractable optimization scheme for the nonconvex, coupled spectral structure, so that the benefits of modeling this two-level sparsity can be realized in tensor learning tasks?*

**Contributions.** To address these questions, we propose the *tensor $\ell_p$-Schatten-q quasi-norm*, a novel framework introducing *dual spectral sparsity control* to simultaneously model both within-frequency and across-frequency dependencies. While our framework offers promising modeling capabilities, the *coupled nature of this dual spectral sparsity* introduces significant theoretical and computational challenges. Accordingly, our main contributions are as follows:

- **Structural Modeling (RQ1):** To the best of our knowledge, this work is the first to explicitly model and analyze dual spectral sparsity within the t-SVD framework,

separating inter-frequency sparsity from intra-frequency low-rankness (Section 3). The proposed $\ell_p$-Schatten-$q$ quasi-norm jointly models both inter-frequency sparsity and intra-frequency low-rankness, while allowing separate control over each via parameters $p$ and $q$. This framework generalizes TNN, unifying existing methods such as the tensor Schatten-$p$ quasi-norm (Kong et al., 2018) and tensor average rank (Wang et al., 2021b) into a single, versatile framework.

- **Theoretical Guarantees (RQ2):** We establish matched minimax lower and upper bounds for tensor estimation under dual spectral sparsity in a Gaussian location model, covering both hard and soft regimes (Section 4). These bounds characterize how the estimation error depends jointly on frequency sparsity and within-slice spectral decay.

- **Optimization and Empirical Validation (RQ3):** We develop a reweighted optimization scheme for the proposed quasi-norm (Section 5). It combines a reweighted approximation with frequency-wise singular value updates in the transform domain to address the nonconvex and coupled regularization structure. Experiments on real-world tensor recovery and clustering tasks demonstrate the empirical effectiveness of the proposed method (Section 6).

The remainder of the paper is organized as follows. Section 2 reviews basic preliminaries. Section 3 introduces the proposed quasi-norm. Sections 4 and 5 present the theoretical analysis and optimization algorithm, respectively. Experimental results are reported in Section 6, followed by the conclusion in Section 7. Details on related work, proofs, algorithms, and experiments are provided in the appendix.

## 2. Notations and Preliminaries

**Notations.** For any positive integer $d$, let $[d] = \{1, \ldots, d\}$. Vectors are denoted by lowercase bold letters (e.g., $\mathbf{a}$), matrices by uppercase bold letters (e.g., $\mathbf{A}$), and third-order tensors by underlined uppercase letters (e.g., $\underline{\mathbf{A}}$). Constants such as $c$, $c_1$, and $C$ may change from line to line. For a tensor of size $d_1 \times d_2 \times m$, we assume $d_1 \geq d_2$ for convenience. We use $a \wedge b := \min(a, b)$, and $a \asymp b$ means that $a$ and $b$ are equal up to absolute constant factors.

For a matrix $\mathbf{A} \in \mathbb{R}^{d_1 \times d_2}$, let $\boldsymbol{\sigma}(\mathbf{A})$ be its singular values in descending order. The spectral and nuclear norms are $\|\mathbf{A}\|_{\mathrm{spec}}$ and $\|\mathbf{A}\|_*$, defined as the largest and the sum of these values. For any tensor $\underline{\mathbf{A}}$, we set $\|\underline{\mathbf{A}}\|_p := \|\operatorname{vec}(\underline{\mathbf{A}})\|_p$ and $\|\underline{\mathbf{A}}\|_{\mathrm{F}} := \|\operatorname{vec}(\underline{\mathbf{A}})\|_2$, where $\operatorname{vec}(\cdot)$ denotes vectorization (Kolda & Bader, 2009). The inner product is $\langle \underline{\mathbf{A}}, \underline{\mathbf{B}} \rangle := \operatorname{vec}(\underline{\mathbf{A}})^\top \operatorname{vec}(\underline{\mathbf{B}})$. For $\underline{\mathbf{A}} \in \mathbb{R}^{d_1 \times d_2 \times m}$, its $i$-th frontal slice is $\underline{\mathbf{A}}_{:,:,i}$, abbreviated as $\underline{\mathbf{A}}_i$ when clear from context.

**The t-SVD Framework.** The t-SVD framework can be understood as applying a linear transform along the third mode of a tensor and then performing matrix operations slice by slice in the transformed domain. For example, after the transform, a tensor $\underline{\mathbf{T}} \in \mathbb{R}^{d_1 \times d_2 \times m}$ is represented by $m$ frontal slices $M(\underline{\mathbf{T}})_{:,:,1}, \ldots, M(\underline{\mathbf{T}})_{:,:,m}$, each of which can be treated as a matrix. This representation allows tensor multiplication, singular value decomposition, and low-rank regularization to be defined through standard matrix operations on these transformed slices. The t-product is a tensor generalization of matrix multiplication under such an invertible linear transform $M$ (Kernfeld et al., 2015). Suitable transforms can strengthen low-rank structure and capture correlations across slices (Zhang & Ng, 2021; Wang et al., 2021a). Following common practice, we use an orthogonal matrix $\mathbf{M} \in \mathbb{R}^{m \times m}$ for stability and efficiency (Lu, 2021; Wang et al., 2023a). For a tensor $\underline{\mathbf{T}} \in \mathbb{R}^{d_1 \times d_2 \times m}$, the $M$-transform and its inverse are defined by

$$M(\underline{\mathbf{T}}) := \underline{\mathbf{T}} \times_3 \mathbf{M}, \qquad M^{-1}(\underline{\mathbf{T}}) := \underline{\mathbf{T}} \times_3 \mathbf{M}^{-1}, \quad (1)$$

where $\times_3$ is the mode-3 tensor–matrix product (Kernfeld et al., 2015). These operators permit the basic notions of the t-SVD framework.

**Definition 2.1** (t-product (Kernfeld et al., 2015))**.** The t-product of $\underline{\mathbf{A}} \in \mathbb{R}^{d_1 \times d_2 \times m}$ and $\underline{\mathbf{B}} \in \mathbb{R}^{d_2 \times d_3 \times m}$ under the transform $M$ in Eq. (1) is denoted by

$$\underline{\mathbf{A}} *_M \underline{\mathbf{B}} = \underline{\mathbf{C}} \in \mathbb{R}^{d_1 \times d_3 \times m},$$

where

$$M(\underline{\mathbf{C}}) = M(\underline{\mathbf{A}}) \odot M(\underline{\mathbf{B}})$$

in the transform domain, and $\odot$ denotes the frontal-slice-wise product.

**Definition 2.2** ($M$-block-diagonal matrix (Wang et al., 2023a))**.** For a tensor $\underline{\mathbf{T}} \in \mathbb{R}^{d_1 \times d_2 \times m}$, its $M$-block-diagonal matrix $\bar{\mathbf{T}} \in \mathbb{R}^{d_1 m \times d_2 m}$ is defined as

$$\bar{\mathbf{T}} := \operatorname{bdiag}(M(\underline{\mathbf{T}})) = \operatorname{diag}(M(\underline{\mathbf{T}})_{:,:,1}, \ldots, M(\underline{\mathbf{T}})_{:,:,m}),$$

where $M(\underline{\mathbf{T}})$ is the mode-3 transform of $\underline{\mathbf{T}}$ in Eq. (1) and $\operatorname{bdiag}(\cdot)$ places the frontal slices on the diagonal.

Thus, the singular values of $\bar{\mathbf{T}}$ collect the singular values of all transformed frontal slices $M(\underline{\mathbf{T}})_{:,:,i}$.

We now introduce the t-SVD, as illustrated in Figure 1-(A).

**Definition 2.3** (t-SVD and tensor tubal rank (Kernfeld et al., 2015))**.** The tensor Singular Value Decomposition (t-SVD) of a tensor $\underline{\mathbf{T}} \in \mathbb{R}^{d_1 \times d_2 \times m}$ under the invertible linear transform $M$ in Eq. (1) is:

$$\underline{\mathbf{T}} = \underline{\mathbf{U}} *_M \underline{\mathbf{S}} *_M \underline{\mathbf{V}}^\top, \qquad (2)$$

where $\underline{\mathbf{U}} \in \mathbb{R}^{d_1 \times d_1 \times m}$ and $\underline{\mathbf{V}} \in \mathbb{R}^{d_2 \times d_2 \times m}$ are t-orthogonal tensors, and $\underline{\mathbf{S}} \in \mathbb{R}^{d_1 \times d_2 \times m}$ is an f-diagonal tensor. The

tubal rank of $\underline{\mathbf{T}}$ is defined as the number of non-zero tubes in $\underline{\mathbf{S}}$ in the t-SVD, i.e.,

$$r_{\text{tb}}(\underline{\mathbf{T}}) := \#\{i \mid \underline{\mathbf{S}}_{i,i,:} \neq \mathbf{0}, i \leq \min\{d_1, d_2\}\}.$$

To further model the low-rank structure of tensors in the transformed domain, the tubal nuclear norm is proposed as a key regularizer in low-rank tensor learning:

**Definition 2.4** (Tubal nuclear norm (Lu et al., 2019b))**.** The tubal nuclear norm (TNN)[1] of $\underline{\mathbf{T}} \in \mathbb{R}^{d_1 \times d_2 \times m}$ under the transform $M$ is defined as

$$\|\underline{\mathbf{T}}\|_* := \|\bar{\mathbf{T}}\|_* = \|\boldsymbol{\sigma}(\bar{\mathbf{T}})\|_1.$$

By definition, TNN sums all singular values from all transformed frontal slices. Equivalently, it measures the element-wise sparsity of the transformed spectrum $\boldsymbol{\sigma}(\bar{\mathbf{T}}) \in \mathbb{R}^{m \cdot \min\{d_1, d_2\}}$ and promotes low-rankness in the transformed domain. From a slice-wise viewpoint, TNN applies a uniform penalty to the singular values of every transformed slice, without explicitly distinguishing whether a slice carries strong or weak spectral energy. This uniform treatment makes TNN simple and effective for many tensor recovery tasks, such as image/video denoising and inpainting (Lu et al., 2019a), but it also motivates the more flexible dual-spectral model introduced in the next section.

## 3. Modeling Dual Spectral Sparsity

This section answers **RQ1** by proposing a modeling tool for dual spectral sparsity, an empirical spectral pattern that standard TNN regularization cannot explicitly capture.

**Prevalence of Dual Spectral Sparsity.** Real tensor data often exhibit a characteristic two-level spectral structure across a range of modalities. Visual evidences in Figure 1 show that only a small subset of frequency bands carries substantial spectral energy for representative datasets, including *Salinas A*[2], *Brain MRI* (Xu et al., 2015), and *Incisix* (Gandy et al., 2011). In these examples, more than eighty percent of the total energy concentrates in roughly fifteen to thirty percent of the frequency bands. The singular-value heatmap in Figure 1(B) further illustrates two distinct features. Many frequency slices contribute negligibly, and the dominant slices display rapid spectral decay, which reflects clear low-rank organization within each active slice.

To assess whether this behavior extends beyond the illustrated cases, we provide a statistical analysis in Appendix D. The hypothesis tests consistently distinguish real tensors

from matched random baselines, confirming the presence of both inter-frequency sparsity and within-band low-rankness across multispectral imaging and video datasets. Together, these results demonstrate that dual spectral sparsity appears as a recurring empirical pattern across a range of real-world tensor datasets.

**A Group-Sparsity View of TNN.** Definition 2.4 shows that TNN promotes low-rankness by applying an element-wise sparsity-promoting penalty to the transformed singular values $\boldsymbol{\sigma}(\bar{\mathbf{T}})$. From a group-sparsity viewpoint, the spectrum $\boldsymbol{\sigma}(\bar{\mathbf{T}})$ can be decomposed into groups $\{\boldsymbol{\sigma}(M(\underline{\mathbf{T}})_{:,:,i})\}_{i=1}^m$, each corresponding to one transformed frequency slice. TNN treats these groups through the same linear penalty on their individual singular values, which makes the regularizer simple and effective but does not explicitly separate group-level frequency selection from within-group spectral decay. This motivates a more structured framework that can distinguish inter-frequency sparsity from intra-frequency low-rankness.

**Hard Dual Spectral Sparsity.** We begin with a hard dual spectral sparsity model as follows.

**Definition 3.1** (Hard Dual Spectral Sparsity)**.** A tensor $\underline{\mathbf{T}} \in \mathbb{R}^{d_1 \times d_2 \times m}$ is said to exhibit $(s, r)$-dual sparsity under a linear transform $M$ if it satisfies two constraints:

(1) *Inter-frequency sparsity:* The number of active frequency components is limited to at most $s$. Specifically, only $s$ out of the $m$ frequency components can have non-zero singular value vectors:

$$\sum_{i=1}^{m} \mathbb{I}\left(\boldsymbol{\sigma}(M(\underline{\mathbf{T}})_{:,:,i}) \neq \mathbf{0}\right) \leq s,$$

where $\boldsymbol{\sigma}(M(\underline{\mathbf{T}})_{:,:,i})$ denotes the vector of singular values of the $i$-th frontal slice in the transformed domain.

(2) *Intra-frequency low-rankness:* Within each active frequency component, the number of non-zero singular values is constrained to at most $r$. This condition ensures a low-rank structure for each frequency slice ($\forall i \in [m]$):

$$\sum_{j=1}^{\min\{d_1, d_2\}} \mathbb{I}\left(\sigma_j(M(\underline{\mathbf{T}})_{:,:,i}) \neq 0\right) \leq r,$$

where $\sigma_j(M(\underline{\mathbf{T}})_{:,:,i})$ denotes the $j$-th singular value of the $i$-th frontal slice of $M(\underline{\mathbf{T}})$.

This definition is not meant to describe real data generation, but to provide a simple structure for studying dual spectral organization. The hard formulation selects a small set of frequencies and enforces low rank within each slice; although this can be too restrictive when spectral energy decays smoothly, it offers a clear baseline for motivating and analyzing softer models.

---

[1]Since multiple notions of tensor nuclear norm exist in the literature, we follow Zhang & Aeron (2017) and use the term *tubal nuclear norm* to specifically refer to the t-SVD-based tensor nuclear norm defined through the transformed frontal slices.

[2]Dataset available at Salinas A.

**Soft Dual Spectral Sparsity.** In practice, spectral energy typically decays smoothly, making the boundary between active and inactive frequencies both gradual and noise–dependent. To model this softer structure, we introduce the $\ell_p$-Schatten-$q$ quasi-norm, which replaces hard selection and fixed ranks with approximate frequency sparsity and smooth within-slice spectral decay.

**Definition 3.2** (Tensor $\ell_p$-Schatten-$q$ quasi-norm). For rational exponents $(p, q) \in (0, 1]^2$ and a tensor $\underline{\mathbf{T}} \in \mathbb{R}^{d_1 \times d_2 \times m}$, we define the tensor $\ell_p$-Schatten-$q$ quasi-norm (abbreviated as $\ell_p(S_q)$-norm) as:

$$\|\underline{\mathbf{T}}\|_{\ell_p(S_q)} := \left( \sum_{i=1}^{m} \left( \sum_{j=1}^{d_1 \wedge d_2} \sigma_j \big( M(\underline{\mathbf{T}})_{:,:,i} \big)^q \right)^{p/q} \right)^{1/p}. \quad (3)$$

In this quasi-norm, $p$ governs the inter-frequency sparsity by promoting a group-wise regularization across frequency components, effectively highlighting significant groups while suppressing others. Simultaneously, $q$ controls the intra-frequency low-rankness by encouraging sparsity in the singular values within each frequency slice, thereby modeling the intrinsic low-rank structure of the data. This soft dual spectral sparsity formulation provides a unified yet versatile approach to address the hierarchical structure of the tensor spectrum.

*Remark* 3.3. The $\ell_p$-Schatten-$q$ quasi-norm encompasses several existing regularization methods: it recovers TNN when $(p, q) = (1, 1)$(Lu et al., 2019b), approximates the average rank as $(p, q) \to (1, 0)$(Wang et al., 2021b), and reduces to the tensor Schatten-$q$ norm when $p = q$ (Kong et al., 2018), thereby offering greater modeling flexibility. *Despite generalizing these regularizers, it fundamentally differs by jointly enforcing global frequency sparsity and local spectral low-rankness.*

# 4. Estimation under Dual Spectral Sparsity

This section develops the theory of tensor estimation with dual spectral sparsity (**RQ2**).

**Challenges.** The $\ell_p$-Schatten-$q$ quasi-norm, combining inter-frequency sparsity with intra-frequency low-rankness, leads to *a globally coupled structure* that fundamentally differs from classical decoupled models like TNN. It results in a highly non-convex parameter space with nested sparsity patterns. Characterization of the estimation complexity demands novel extensions of covering numbers and metric entropy that jointly capture inter-frequency sparsity and intra-frequency low-rankness under both discrete and continuous constraints.

To uncover the statistical limits of learning under dual spectral sparsity, we analyze a simplified but representative model: the *Gaussian location model*, where the observed

tensor is corrupted by Gaussian noise (Li et al., 2024). This setting preserves the core structural properties, that is, inter-frequency sparsity and intra-frequency low-rankness, while avoiding complications unrelated to sparsity itself. We define structured parameter spaces that capture hard and soft variants of dual spectral sparsity, and establish sharp minimax lower and upper bounds under each. These results reveal how estimation difficulty is jointly governed by the extent of frequency sparsity and the spectral decay within each active slice.

### 4.1. Gaussian Location Model

Consider the Gaussian location model (GLM) (Li et al., 2024), where $n$ independent noisy realizations of the target tensor $\underline{\mathbf{L}}^* \in \mathbb{R}^{d_1 \times d_2 \times m}$ are observed as:

$$\underline{\mathbf{Y}}_i = \underline{\mathbf{L}}^* + \underline{\mathbf{E}}_i, \quad i \in [n], \quad (4)$$

where $\underline{\mathbf{Y}}_i$ is the observed tensor, $\underline{\mathbf{L}}^*$ represents the ground truth tensor of interest, and $\underline{\mathbf{E}}_i$ denotes the noise tensor with *i.i.d.* entries from $\mathcal{N}(0, \sigma^2)$. The parameter $\sigma$ characterizes the noise level. The goal here is to estimate the unknown truth tensor $\underline{\mathbf{L}}^*$ from its noisy observations $\{\underline{\mathbf{Y}}_i\}_{i=1}^n$.

*Remark* 4.1. We adopt GLM to isolate the core effects of dual spectral sparsity and the $\ell_p$-Schatten-$q$ regularization, avoiding additional complications from sampling operators in tensor regression/recovery (Zhang et al., 2020; Wang et al., 2021a; Qiu et al., 2024). This simplified setting enables cleaner analysis and yields insights that extend naturally to regression problems like tensor compressed sensing or tensor completion under standard conditions such as RIP (Zhang et al., 2020) or RSC (Wang et al., 2021a; Qiu et al., 2024; Negahban & Wainwright, 2012). *As such, the results should be interpreted as characterizing how dual spectral structure impacts estimation difficulty, rather than as performance guarantees for any specific downstream task.*

**Dual Spectral Sparsity Assumptions.** We consider two distinct sparsity models for $\underline{\mathbf{L}}^*$:

**A1.** *Hard dual spectral sparsity*: Let $\underline{\mathbf{L}}^*$ belong to the parameter space

$$\mathbf{T}_{0,0}(s, r) = \left\{ \underline{\mathbf{L}} : (s, r)\text{-dual sparse} \right\}.$$

Here $(s, r)$-dual sparse means that $\underline{\mathbf{L}}$ has at most $s$ active frequency slices in the $M$-domain, and each active slice has rank at most $r$, with the understanding that $s$ and $r$ are strictly smaller than their ambient dimensions so that the frequency sparsity and per-slice low-rank structures are nondegenerate.

**A2.** *Soft dual spectral sparsity*: Let $\underline{\mathbf{L}}^*$ belong to the parameter space

$$\mathbf{T}_{p,q}(R) = \left\{ \underline{\mathbf{L}} : \|\underline{\mathbf{L}}\|_{\ell_p(S_q)}^p \leq R \right\}, \quad (5)$$

where the $\ell_p(S_q)$ quasi-norm provides a soft spectral structure combining frequency selection with within-slice spectral decay, and $R$ specifies the radius of the constraint set.

These parameter spaces represent increasing modeling flexibility: the hard model imposes strict structural constraints, while the fully soft model allows gradual spectral decay across frequencies and within slices. Our goal is to estimate $\underline{\mathbf{L}}^*$ in the Gaussian location model (4) and establish minimax bounds under each setting.

### 4.2. Minimax Risk over Dual Spectral Sparsity

A natural problem here is to understand the statistical difficulty of recovering tensors with dual spectral sparsity from noisy observations. Under the Gaussian location model, we study the fundamental difficulty of estimating tensors in a structured parameter space **T**. The minimax risk measures the best achievable worst-case error over this space:

$$\mathfrak{M}(\mathbf{T}) = \inf_{\hat{\underline{\mathbf{L}}}} \sup_{\underline{\mathbf{L}}^* \in \mathbf{T}} \mathbb{E}\left[\|\hat{\underline{\mathbf{L}}} - \underline{\mathbf{L}}^*\|_{\mathrm{F}}^2\right], \qquad (6)$$

where $\underline{\mathbf{L}}^*$ is the unknown ground-truth tensor and $\hat{\underline{\mathbf{L}}}$ ranges over all possible estimators constructed from the observations. The infimum–supremum form captures the performance of the best estimator against the worst-case tensor in **T**. We consider $d_1 = d_2 = d$ for notational simplicity.

**Theorem 4.2** (Minimax Rates with Dual Spectral Sparsity). *Upper and lower bounds are established for the minimax estimation risk of tensors with dual spectral sparsity under the Gaussian location model as follows:*

**I.** *Under hard constraints on both frequency sparsity and per-slice low-rankness, the minimax risk satisfies*

$$\mathfrak{M}(\mathbf{T}_{0,0}(s,r)) \asymp \frac{\sigma^2}{n}\left(s\log\frac{em}{s} + srd\right).$$

**II.** *Under soft $\ell_p(S_q)$ constraints on both frequency sparsity and intra-slice low-rankness, in mild nondegenerate regimes[3], the minimax risk satisfies*

$$\mathfrak{M}(\mathbf{T}_{p,q}(R)) \asymp$$
$$\begin{cases} R\left(\frac{d\sigma^2}{n}\right)^{\frac{2-p}{2}} + R\left(\frac{\sigma^2\log m}{n}\right)^{\frac{2-p}{2}}, & p > q, \\ R^{\frac{q}{p}}\left(\frac{d\sigma^2}{n}\right)^{\frac{2-q}{2}} + R\left(\frac{\sigma^2\log m}{n}\right)^{\frac{2-p}{2}}, & p \leq q,\ m > d^2, \\ R^{\frac{q}{p}}\left(\frac{d\sigma^2}{n}\right)^{\frac{2-q}{2}}, & p \leq q,\ m \leq d^2. \end{cases}$$

Theorem 4.2 characterizes how estimation difficulty arises from the interaction between frequency sparsity and within-frequency low-rankness. The results can be interpreted along two representative regimes.

---
[3]The nondegenerate regimes are specified in Theorem C.4.

**I (Hard dual spectral sparsity).** When both frequency sparsity and per-slice rank constraints are enforced exactly, the minimax risk decomposes into two additive components. Selecting the $s$ informative frequency slices incurs a combinatorial cost of $s\log(em/s)$, while estimating $s$ rank–$r$ matrices contributes a parametric term of order $srd$.

**II (Soft $\ell_p(S_q)$ dual spectral sparsity).** When both structures are relaxed, the estimation complexity is governed by the interplay between the $\ell_p$ frequency regularization and the Schatten–$q$ within-slice decay. If $p > q$, the rate is dominated by the $\ell_p$ component, reflecting the cost of identifying a small set of influential frequencies. If $p \leq q$, the Schatten–$q$ term becomes dominant, and the resulting rate is independent of the ambient frequency dimension $m$ once $m \leq d^2$. This regime explicitly demonstrates how frequency selection and spectral decay jointly determine the effective model complexity.

## 5. Optimization for Dual Spectral Sparsity

Efficiently solving tensor estimation problems with dual spectral sparsity (**RQ3**) is key to leveraging the proposed $\ell_p$-Schatten-$q$ quasi-norm in practice. However, this task presents substantial challenges due to the non-convexity and coupled structure of this regularization.

**Challenges.** Even in the vector setting, optimizing dual-level sparse structures is notoriously difficult due to the combination of *non-convexity* and *structural coupling* (Hu et al., 2017; Lin et al., 2025). In the tensor case, these challenges are further compounded by the need to jointly enforce inter-frequency sparsity and intra-frequency low-rankness. Most existing tensor optimization methods (Lu et al., 2019a; Kong et al., 2018; Wang et al., 2021a) are ill-suited for this setting: they either treat frequency components independently or impose low-rank constraints without spectral sparsity. Moreover, the $\ell_p$-Schatten-$q$ quasi-norm is non-convex whenever $p, q \in (0, 1)$, ruling out standard convex optimization techniques.

To address these issues, we adopt a *proximal update scheme* that exploits the separability of the transform-domain representation $M(\underline{\mathbf{L}})$, together with an iterative reweighting strategy to handle the non-convex regularization.

**Proximal Operator Formulation.** The key idea is to perform shrinkage in the transformed spectral domain, while allowing the shrinkage strength to adapt to both the frequency slice and the singular-value magnitude. Unlike TNN, which applies a uniform linear penalty to all transformed singular values, the proposed $\ell_p(S_q)$ regularizer induces coupled and nonconvex shrinkage across frequency slices and within each slice. We therefore use a reweighted proximal update that preserves the slice-wise structure of the transform domain while approximating the coupled regular-

izer. Specifically, at iteration $t$, the update $\underline{\mathbf{L}}^{t+1}$ is given by solving:

$$\min_{\underline{\mathbf{L}}} \frac{1}{2}\|\underline{\mathbf{L}} - \underline{\mathbf{Z}}^t\|_{\mathrm{F}}^2 + \lambda \sum\nolimits_{k=1}^{m} \|M(\underline{\mathbf{L}})_{:,:,k}\|_{S_q}^p, \quad (7)$$

where $\underline{\mathbf{Z}}^t$ is the intermediate tensor produced by the outer optimization scheme. Since $M$ is orthogonal and the transform-domain representation is slice-wise, Problem (7) reduces to $m$ subproblems over frequency components $k \in [m]$:

$$\min_{\mathbf{A}_k} \frac{1}{2} \left\| \mathbf{A}_k - M(\underline{\mathbf{Z}}^t)_{:,:,k} \right\|_{\mathrm{F}}^2 + \lambda \|\mathbf{A}_k\|_{S_q}^p, \quad (8)$$

where $\mathbf{A}_k := M(\underline{\mathbf{L}})_{:,:,k}$ denotes the $k$-th frontal slice of the transformed tensor $M(\underline{\mathbf{L}})$. Problem (8) is difficult due to the non-convexity and lack of smoothness of the Schatten-$q$ quasi-norm, which admits no closed-form or standard proximal solution in general.

To tackle Problem (8), we adopt a reweighted $\ell_{1/2}$-proxy for $\|\mathbf{A}_k\|_{S_q}^p$ based on singular values:

$$\sum\nolimits_{i=1}^{d} w_{i,k} \cdot \sigma_i(\mathbf{A}_k)^{1/2}, \quad (9)$$

with weights defined as $w_{i,k} = \left( \sum_{j=1}^{d} \varsigma_{j,k}^q + \epsilon \right)^{p/q-1} \cdot \left( \varsigma_{i,k}^{1/2} + \epsilon \right)^{2q-1}$, where $\epsilon$ is a small regularization constant and $\varsigma_{j,k} := \sigma_j(M(\underline{\mathbf{L}}^t)_{:,:,k})$ are the singular values from the previous iterate. The weights adapt the shrinkage to both the overall energy of each frequency slice and the relative magnitude of individual singular values, thereby approximating the coupled effect of inter-frequency sparsity and intra-frequency spectral decay. The update for each singular value then becomes a half-thresholding step:

$$\sigma_i^{(t+1)}(M(\underline{\mathbf{L}})_{:,:,k}) = \mathcal{S}_{\lambda w_{i,k}}^{\ell_{1/2}} \left( \sigma_i(M(\underline{\mathbf{Z}}^t)_{:,:,k}) \right), \quad (10)$$

where $\mathcal{S}^{\ell_{1/2}}$ is the proximal operator for the $\ell_{1/2}$-norm (see Eq. (41) in Appendix E.2).

After singular value shrinkage, we reconstruct each slice $M(\underline{\mathbf{L}}^{t+1})_{:,:,k} = \mathbf{U}_k \cdot \mathrm{diag}(\boldsymbol{\sigma}^{(t+1)}) \cdot \mathbf{V}_k^\top$, where $\mathbf{U}_k$ and $\mathbf{V}_k$ are from the SVD of $M(\underline{\mathbf{Z}}^t)_{:,:,k}$. Finally, applying the inverse transform yields the updated tensor $\underline{\mathbf{L}}^{t+1}$ in the original domain.

**Algorithmic Integration and Convergence.** The proposed weighted $\ell_{1/2}$ proximal operator enables an efficient and fully specified optimization scheme for the nonconvex $\ell_p(S_q)$ regularization. In particular, it yields a closed-form solution to the $\underline{\mathbf{L}}$-subproblem within a scaled Alternating Direction Method of Multipliers (ADMM) framework, which decouples the data fidelity term from the dual spectral regularizer and allows slice-wise spectral updates in the transform domain. The resulting ADMM iterations consist of explicit updates for $\underline{\mathbf{L}}$, $\underline{\mathbf{K}}$, and the scaled dual variable, and are summarized in Algorithm 1. For completeness, we provide the full algorithmic description and computational complexity in Appendix E.2, and complement the algorithmic design with a Lyapunov-based analysis establishing convergence to critical points, rather than global optimality, for the nonconvex objective in Appendix E.3.

# 6. Experiments

The experiments evaluate whether the proposed algorithm effectively realizes the benefits of modeling dual spectral sparsity, as posed in **RQ3**. Specifically, we address the following empirical questions:

- **Performance:** *Does the solver exploit dual spectral sparsity to improve recovery and representation quality?*

- **Robustness:** *Are the gains stable across tasks, noise models, spectral transforms, tensor orders, and parameter choices?*

To this end, we evaluate the proposed method on three representative tasks: *noisy tensor completion*, *Poisson tensor completion*, and *image clustering*. These tasks progressively stress different aspects of the algorithm, from reconstruction accuracy under additive noise, to robustness under signal-dependent noise, and finally to downstream representation learning. Additional ablations and robustness analyses are deferred to Appendix E.

**Noisy Tensor Completion.** We first evaluate the proposed regularizer on noisy tensor completion[4], which serves as the primary benchmark for assessing reconstruction accuracy under limited observations and additive noise. This setting tests the ability of the proposed model to exploit dual spectral sparsity in the presence of missing data and noise.

Given a clean tensor $\underline{\mathbf{L}} \in \mathbb{R}^{d_1 \times d_2 \times d_3}$, we observe noisy and incomplete measurements with i.i.d. Gaussian noise of standard deviation $\sigma = c\sigma_0$, where $c = 0.05$ and $\sigma_0 = \|\underline{\mathbf{L}}\|_{\mathrm{F}}/\sqrt{d_1 d_2 d_3}$. Uniform random sampling is applied with sampling ratios SR $\in \{5\%, 10\%, 15\%\}$. Each experiment is repeated over 10 independent trials, and average PSNR (dB) and SSIM are reported.

We compare $\ell_p(S_q)$ against a broad set of low-rank tensor regularizers, including NN (Candès & Tao, 2010), SNN (Liu et al., 2013), TNN-DFT (Zhang & Aeron, 2017), TNN-DCT (Lu et al., 2019b), the $k$-Support norm (Wang et al., 2021a), the $\ell_{1-2}$ norm (Tan et al., 2023), and Schatten-$p$ minimization ($p = 1/2$) (Kong et al., 2018). Unless otherwise specified, we adopt the DCT as the spectral transform.

---

[4]Example MATLAB code for reproducing the tensor completion experiments is available at https://github.com/pingzaiwang/LpSq-QuasiNorm/tree/main.

*Table 1.* Noisy tensor completion results on benchmark datasets. The proposed $\ell_p(S_q)$ method is tested with three parameter settings: **S1** $(0.60, 0.61)$, **S2** $(0.70, 0.71)$, and **S3** $(0.80, 0.81)$. The best and second-best results are marked in **bold** and underline, respectively.

| Dataset | SR | Metric | NN | SNN | TNN-DFT | TNN-DCT | $k$-Sup | $\ell_{1-2}$ | $S_{1/2}$ | S1 | S2 | S3 |
|---|---|---|---|---|---|---|---|---|---|---|---|---|
| SalinasA | 5% | PSNR | 15.21 | 20.79 | 22.55 | 26.52 | 22.58 | 22.21 | 22.45 | 28.50 | **28.72** | 28.55 |
| | | SSIM | 0.2594 | **0.7547** | 0.5667 | 0.7384 | 0.5689 | 0.5524 | 0.4474 | 0.7399 | 0.7505 | 0.7401 |
| | 10% | PSNR | 20.62 | 25.56 | 25.72 | 29.61 | 25.89 | 26.14 | 25.86 | 31.58 | 31.57 | **31.62** |
| | | SSIM | 0.4775 | 0.8284 | 0.7027 | 0.8403 | 0.7231 | 0.7197 | 0.6058 | 0.8423 | 0.8413 | **0.8428** |
| | 15% | PSNR | 23.09 | 27.99 | 28.06 | 31.32 | 28.09 | 28.13 | 26.98 | 33.16 | 33.17 | **33.28** |
| | | SSIM | 0.5643 | 0.8622 | 0.7804 | 0.8798 | 0.7810 | 0.7795 | 0.6505 | 0.8832 | 0.8846 | **0.8856** |
| IndianPines | 5% | PSNR | 20.44 | 22.01 | 25.68 | 26.26 | 25.70 | 25.73 | 24.68 | 26.68 | 26.86 | **26.87** |
| | | SSIM | 0.3895 | 0.6359 | 0.6293 | **0.6727** | 0.6289 | 0.6316 | 0.5361 | 0.6537 | 0.6658 | 0.6636 |
| | 10% | PSNR | 22.23 | 24.94 | 27.45 | 28.40 | 27.48 | 27.52 | 25.72 | 28.96 | **29.02** | 28.97 |
| | | SSIM | 0.4836 | 0.7171 | 0.7226 | **0.7744** | 0.7219 | 0.7249 | 0.5991 | 0.7657 | 0.7684 | 0.7653 |
| | 15% | PSNR | 23.52 | 26.61 | 28.54 | 29.52 | 28.53 | 28.63 | 26.24 | 29.93 | **30.01** | 29.97 |
| | | SSIM | 0.5438 | 0.7668 | 0.7713 | **0.8177** | 0.7709 | 0.7741 | 0.6258 | 0.8026 | 0.8090 | 0.8039 |
| Cloth | 5% | PSNR | 20.10 | 20.95 | 25.00 | 26.09 | 25.08 | 25.09 | 24.96 | 26.79 | 27.02 | **27.24** |
| | | SSIM | 0.3762 | 0.5096 | 0.6773 | 0.7283 | 0.6792 | 0.6793 | 0.6305 | 0.7227 | 0.7342 | **0.7447** |
| | 10% | PSNR | 21.14 | 22.72 | 28.00 | 29.24 | 28.12 | 28.14 | 27.98 | 30.17 | 30.36 | **30.52** |
| | | SSIM | 0.4341 | 0.5983 | 0.8132 | 0.8540 | 0.8143 | 0.8163 | 0.7668 | 0.8473 | 0.8546 | **0.8576** |
| | 15% | PSNR | 22.05 | 24.18 | 30.03 | 31.36 | 30.08 | 30.11 | 29.50 | 32.31 | 32.51 | **32.61** |
| | | SSIM | 0.4889 | 0.6783 | 0.8722 | **0.9054** | 0.8727 | 0.8733 | 0.8153 | 0.8977 | 0.9025 | 0.9047 |
| Hair | 5% | PSNR | 25.33 | 30.09 | 33.16 | 35.31 | 33.19 | 33.27 | 33.43 | 36.47 | 36.81 | **36.84** |
| | | SSIM | 0.7147 | 0.8631 | 0.8917 | **0.9248** | 0.8921 | 0.8919 | 0.8240 | 0.9151 | 0.9197 | 0.9182 |
| | 10% | PSNR | 29.52 | 33.35 | 36.22 | 38.18 | 36.17 | 36.30 | 35.69 | 39.70 | **39.86** | 39.83 |
| | | SSIM | 0.8008 | 0.9122 | 0.9292 | **0.9535** | 0.9286 | 0.9296 | 0.8640 | 0.9510 | 0.9526 | 0.9515 |
| | 15% | PSNR | 31.12 | 35.24 | 38.00 | 39.88 | 37.91 | 38.07 | 36.46 | 41.31 | 41.47 | **41.51** |
| | | SSIM | 0.8364 | 0.9336 | 0.9449 | 0.9650 | 0.9442 | 0.9448 | 0.8735 | 0.9640 | **0.9653** | 0.9645 |
| JellyBeans | 5% | PSNR | 16.33 | 18.21 | 25.43 | 26.47 | 25.38 | 25.62 | 25.39 | 27.91 | 28.08 | **28.16** |
| | | SSIM | 0.2397 | 0.4942 | 0.6726 | **0.7223** | 0.6714 | 0.6733 | 0.5504 | 0.6916 | 0.7001 | 0.7083 |
| | 10% | PSNR | 18.12 | 22.11 | 28.50 | 30.14 | 28.47 | 28.67 | 28.41 | **31.95** | 31.78 | 31.90 |
| | | SSIM | 0.3169 | 0.6629 | 0.7900 | **0.8518** | 0.7902 | 0.7932 | 0.6905 | 0.8321 | 0.8411 | 0.8461 |
| | 15% | PSNR | 19.92 | 24.67 | 30.51 | 32.33 | 30.52 | 30.61 | 29.96 | 33.39 | 33.81 | **33.89** |
| | | SSIM | 0.4053 | 0.7592 | 0.8489 | **0.9030** | 0.8504 | 0.8499 | 0.7516 | 0.8818 | 0.8932 | 0.8952 |
| Thermal | 5% | PSNR | 13.19 | 15.83 | 28.06 | 27.99 | 28.01 | 28.19 | 28.11 | 30.21 | 30.09 | **30.49** |
| | | SSIM | 0.1848 | 0.4759 | 0.8584 | 0.8707 | 0.8579 | 0.8603 | 0.7928 | 0.8701 | 0.8752 | **0.8763** |
| | 10% | PSNR | 14.67 | 19.75 | 31.30 | 31.62 | 31.28 | 31.60 | 30.51 | **33.96** | 33.76 | 33.85 |
| | | SSIM | 0.2509 | 0.6594 | 0.9151 | 0.9326 | 0.9147 | 0.9168 | 0.8358 | 0.9323 | **0.9331** | 0.9291 |
| | 15% | PSNR | 16.27 | 22.52 | 33.02 | 33.51 | 33.05 | 33.11 | 30.99 | 35.42 | **35.50** | 35.36 |
| | | SSIM | 0.3273 | 0.7621 | 0.9315 | **0.9509** | 0.9321 | 0.9318 | 0.8373 | 0.9473 | 0.9507 | 0.9456 |

For $\ell_p(S_q)$, we explore three configurations of the sparsity parameters $(p, q)$, denoted as: **S1**: $(0.60, 0.61)$, **S2**: $(0.70, 0.71)$, and **S3**: $(0.80, 0.81)$. A comprehensive sensitivity analysis of $(p, q)$ is provided in Appendix E.14.

We conduct experiments on a variety of remote sensing data: hyperspectral (*Indian Pines*, *Salinas A*), multispectral (Columbia MSI: *Cloth*, *Hair*, *Jelly Beans*), and thermal imagery (*OSU Thermal*). Table 1 reports the Peak Signal-to-Noise Ratio (PSNR) and Structural Similarity Index (SSIM) values across methods and sampling rates. The proposed $\ell_p(S_q)$ regularizer consistently achieves the highest PSNR and competitive SSIM across datasets and sampling rates, indicating improved reconstruction accuracy while preserving structural and spectral fidelity.

**Poisson Tensor Completion.** To evaluate robustness beyond Gaussian noise, we extend $\ell_p(S_q)$ to the Poisson tensor completion setting, where observations follow signal-dependent photon statistics and sampling masks are non-uniform. Following Wang et al. (2025a), we embed $\ell_p(S_q)$

into the Bi-Module Tensor Regularization (BTR) framework, yielding a Bi-$\ell_p(S_q)$ model. We compare against established Poisson completion baselines, including PoissonNN (Cao & Xie, 2015), SNN, TMac (Xu et al., 2015), TNN-DFT/DCT (Zhang & Ng, 2021), and BTR-DFT/DCT (Wang et al., 2025a). Figure 2 reports PSNR at a 20% non-uniform sampling ratio across seven benchmark datasets, with full numerical results provided in Appendix E.12. Bi-$\ell_p(S_q)$ consistently attains the highest reconstruction quality, providing evidence that explicitly modeling dual spectral sparsity remains effective under heteroscedastic noise and irregular sampling patterns.

**Image Clustering.** Finally, we evaluate Bi-$\ell_p(S_q)$ on image clustering, where low-rank self-representation plays a central role. Experiments on four benchmark datasets (*FRDUE*, *FRDUE100*, *PIE10*, and *USPS1000*) show consistent improvements in ACC, NMI, and PUR over competing methods (Figure 3), with full numerical results in Appendix E.13. These results indicate that the proposed dual spectral prior induces representations with improved cluster separability,

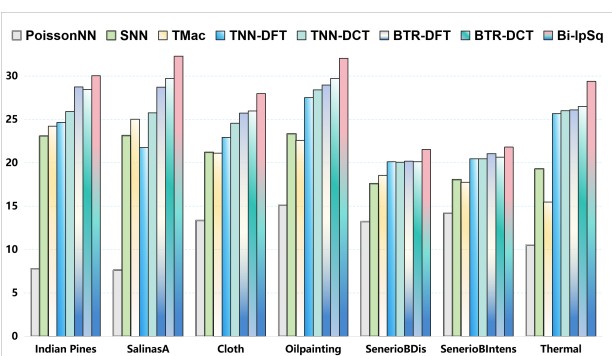

*Figure 2.* PSNR comparison for Poisson tensor completion under 20% non-uniform sampling across seven benchmark datasets.

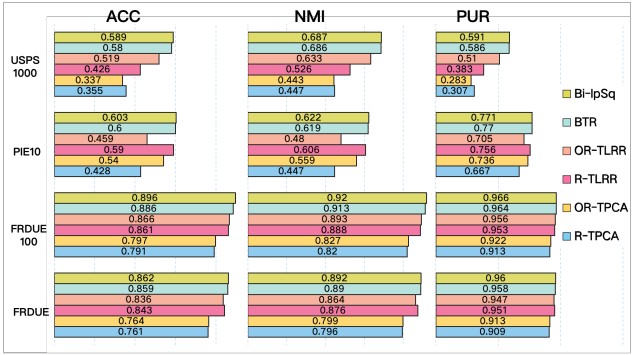

*Figure 3.* Clustering performance measured by accuracy (ACC), normalized mutual information (NMI), and purity (PUR) on four benchmark datasets, compared across competing methods.

and further demonstrate that the benefits of $\ell_p(S_q)$ extend beyond reconstruction tasks to downstream unsupervised representation learning.

**Additional Experimental Coverage.** We also provide supplementary experiments to further examine the algorithmic realization, robustness, and generality of the proposed $\ell_p(S_q)$ framework. These include: (**i**) algorithmic details, computational complexity, and empirical convergence behavior of the ADMM solver (Appendices E.2 and E.3); (**ii**) robustness analyses with respect to modeling and experimental variations, including $(p, q)$ and $\lambda$ sensitivity, different spectral transforms (DFT, random, and adaptive), as well as noiseless and non-Gaussian noise settings (Appendices E.14–E.15 and E.5–E.9); and (**iii**) extensions to additional data modalities and higher-order tensors (Appendices E.10 and E.11).

**Experimental Summary.** Across diverse tasks including noisy and Poisson tensor completion as well as image clustering, the proposed $\ell_p(S_q)$ regularizer consistently matches or outperforms state-of-the-art baselines. These improvements remain stable across sampling ratios, noise models, spectral transforms, and a broad range of $(p, q)$ configurations, indicating that the observed gains are driven by the underlying dual spectral structure rather than delicate parameter tuning or task-specific design.

# 7. Conclusion

We identify a coupled spectral structure in the t-SVD framework, where inter-frequency sparsity and intra-frequency low-rankness coexist. To model this structure, we propose the $\ell_p$-Schatten-$q$ quasi-norm, which unifies and extends classical tensor regularizers. We characterize the fundamental estimation limits under this dual spectral model via minimax analysis, and develop a proximal reweighted algorithm for the resulting nonconvex optimization. Experiments on noisy tensor completion, Poisson tensor completion, and clustering demonstrate that dual spectral regularization provides an effective inductive bias for structured tensor data.

**Limitations and Open Questions.** Although $\ell_p$-quasi-norms (Xu, 2010) and related nonconvex spectral regularizers have been extensively studied, the proposed dual spectral sparsity and the $\ell_p(S_q)$ quasi-norm are new objects. ***Our current understanding of their statistical, geometric, and algorithmic properties is still at an early stage***.

(**i**) Our statistical validation in Appendix D compares real tensors with matched random baselines. This supports the presence of dual spectral sparsity, but leaves a finer question open: *where does the statistical gain mainly come from? Is it driven by element-wise spectral sparsity, group-level frequency sparsity, or the interaction between the two?* Diagnostics that can separate these effects would make dual spectral sparsity more interpretable and practical.

(**ii**) Our experiments suggest that the method often performs stably when $p$ and $q$ are close, with $q$ slightly larger than $p$. This raises a sharper modeling question: *how much does explicit frequency-level weighting actually contribute beyond within-slice spectral shrinkage?* The results suggest that frequency information is useful, but perhaps only in a mild form; overly strong frequency weighting may distort the within-slice spectral geometry. A principled rule for when and how to use frequency information remains open.

(**iii**) The current solver can be more sensitive when the gap between $q$ and $p$ is large. This points to an algorithmic question: *how can one prevent the adaptive thresholds from becoming overly sensitive to small or noisy singular values?* Understanding this issue may lead to more stable reweighted updates for dual spectral regularization.

# Acknowledgements

We would like to thank Dr. Zhifan Li for the exceptionally prompt and helpful discussions on the technical details of Li et al. (2024), especially for patiently clarifying several proof arguments during late-night correspondence. This work was supported in part by JSPS KAKENHI Grant Numbers JP25K21283, JP24K20849, JP23K28109, JP24K03005, JP25K21288, JP26K21319, RIKEN FY2026

Gutaiteki Collaboration Seed Fund and JSPS Bilateral Program Number JPJSBP120257420; and in part by the National Natural Science Foundation of China under Grant Numbers 62562065. Yuning Qiu was supported by the RIKEN Special Postdoctoral Researcher Program.

## Impact Statement

This work is primarily theoretical and relies on computational methods and publicly available datasets. It involves no human subjects, personal data, or sensitive information, and adheres to the ICML Code of Ethics.

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

# Appendix for
## *Refining Dual Spectral Sparsity in Transformed Tensor Singular Values*

This appendix provides a comprehensive supplement to the main paper, elaborating on theoretical foundations, algorithmic components, and additional empirical validations. The materials are organized as follows:

- **Relation to Existing Regularizers (Appendix A.2).** We discuss how the proposed $\ell_p(S_q)$ regularizer relates to TNN models and classical $\ell_u(\ell_q)$-type norms, and clarify the modeling differences introduced by combining frequency-level sparsity with within-slice spectral decay.

- **Additional Notations and Preliminaries (Appendix B).** This section formalizes key notations under the t-SVD framework and presents auxiliary lemmas that support our entropy calculations and complexity analysis.

- **Theoretical Results for Understanding Dual-Level Sparsity (Appendix C).** We provide detailed proofs of the main theorems concerning minimax lower and upper bounds, along with the structural properties of the $\ell_p(S_q)$ space. We also include a numerical illustration of Theorem 4.2 to empirically validate key trends.

- **Statistical Validation of the Dual Spectral Structure (Appendix D).** This section provides a systematic validation of dual spectral sparsity in real-world tensors using quantitative metrics and hypothesis tests against matched Gaussian random baselines.

- **Experimental Details and Additive Results (Appendix E).** This section describes the experimental setup, the ADMM-based solver and the convergence behavior in Appendices E.1-E.3. It also includes ablations on surrogate choices, different spectral transforms (fixed and adaptive), robustness to sampling ratios and noise models, as well as additional noiseless tensor completion benchmarks (Appendices E.4-E.9). We further report extended experiments on diverse 3D and 4D tensors, including YUV videos and seismic volumes (Appendices E.10–E.11), a Poisson tensor completion benchmark (Appendix E.12), image clustering experiments (Appendix E.13), and sensitivity analyses of $(p, q)$ (Appendix E.14) and $\lambda$ (Appendix E.15).

- **Discussion of Limitations and Future Directions (Appendix F).** We reflect on the scope of this work and articulate several potential extensions, including theoretical challenges and promising algorithmic refinements.

## Appendix Contents

# A. Comparison with Prior Work and Unique Technical Challenges

## A.1. General Context and Related Work

Our work intersects with two major research areas: double sparse structures and tensor recovery methods.

**Double Sparse Structures.**   Research on double sparse structures has demonstrated their effectiveness in capturing hierarchical sparsity patterns across various domains. In genomics, these structures model pathway-level and SNP-level sparsity in genome-wide association studies (Silver et al., 2013), while similar hierarchical patterns appear in classification tasks (Rao et al., 2015; Huo et al., 2020) and network analysis (Tugnait, 2021). Methodologically, bi-level selection approaches (Breheny & Huang, 2009) have dominated this field, evolving from the fundamental group bridge method (Huang et al., 2009) to more sophisticated techniques like sparse group lasso (Simon et al., 2013), which unified individual (Tibshirani, 1996) and group-level (Yuan & Lin, 2006) sparsity. Recent theoretical work by Tony Cai et al. (2022) has established minimax bounds for double sparse regression. Li et al. (2024) developed fundamental theoretical bounds for high-dimensional double sparse structures by establishing novel metric entropy bounds over $\ell_u(\ell_q)$-balls using Gilbert-Varshamov techniques, providing insights into the simultaneous estimation of group-wise and element-wise sparsity. However, these approaches, while successful for vector and matrix data, cannot address the fundamental challenge in transformed-domain tensor analysis: *how to simultaneously model sparsity across different frequency components and low-rank structures within each frequency component*, which is a gap our work bridges through the $\ell_p$-Schatten-$q$ framework.

**Tensor Recovery Methods.**   Tensor recovery research has evolved along multiple methodological paths, each addressing different aspects of multi-dimensional data analysis. Traditional approaches utilize various decomposition frameworks: CP decomposition with techniques ranging from sum-of-squares (Barak & Moitra, 2016) to gradient descent (Cai et al., 2019), Tucker decomposition employing nuclear norm minimization (Gandy et al., 2011) and manifold optimization (Xia & Yuan, 2019), and tensor train/ring decompositions using Riemannian methods (Cai et al., 2026). Recently, low-tubal-rank recovery has gained attention, with methods spanning both convex approaches like tubal nuclear norm (TNN) minimization (Lu et al., 2019a; Zhang & Ng, 2021; Qiu et al., 2024) and non-convex alternatives such as Schatten-$p$ norm regularization (Kong et al., 2018). However, existing tensor methods, particularly those based on TNN (Hou et al., 2021; Zhou et al., 2019; Xie et al., 2018; Wu, 2024), apply uniform regularization across all frequencies in the transformed domain. While effective, this single-level sparsity treatment can be further enhanced to better capture the natural hierarchical patterns in real-world tensor data; for example, in hyperspectral images where some frequency bands carry more information than others while each band itself exhibits low-rank structure. *Our $\ell_p$-Schatten-$q$ framework extends TNN's uniform regularization by introducing separate parameters to control sparsity across frequencies ($p$) and low-rank structure within each frequency ($q$).*

## A.2. Relation to Existing Regularizers

We briefly clarify the relation between the proposed $\ell_p(S_q)$ regularizer and two closely related lines of work: the TNN and mixed $\ell_u(\ell_q)$-type norms.

First, TNN can be viewed as a special case of our formulation when $(p, q) = (1, 1)$. It sums the singular values of all transformed frontal slices with a uniform linear penalty and has been widely used in t-SVD-based tensor recovery (Lu et al., 2019b; Zhang et al., 2014). In contrast, the proposed $\ell_p(S_q)$ regularizer separates two levels of spectral structure: sparsity across transformed frequency slices and spectral decay within each slice. This allows the model to distinguish frequency-level selection from within-slice low-rankness, while still reducing to standard TNN in the convex limiting case.

Second, $\ell_u(\ell_q)$-type norms have been studied in vector and matrix-valued double sparse models (Li et al., 2024). Our setting shares a similar hierarchical flavor, but the groups here correspond to transformed frontal slices, and each group is a matrix whose internal structure is described through its singular values. Thus, the proposed model can be regarded as a tensor spectral analogue of double-structured sparsity, adapted to the t-SVD transform-domain representation.

These differences lead to several technical considerations. On the statistical side, the parameter space combines frequency-slice selection with low-rank spectral structure within selected slices, so its complexity cannot be described solely by standard sparse-vector or low-rank-matrix models. Our minimax analysis therefore tracks both the frequency-level and within-slice contributions. On the algorithmic side, the nonconvex $\ell_p(S_q)$ penalty does not generally admit the same closed-form proximal map as TNN. We therefore use a reweighted spectral surrogate, leading to slice-wise singular-value updates within an ADMM framework.

# B. Additional Notations, Preliminaries and Lemmas

**Additional Notations.** We use several asymptotic notations to describe the relationships between functions. For the sake of clarity, we provide their definitions here:

- The notation $f(n) \lesssim g(n)$ means that there exists a positive constant $c$ and a positive integer $n_0$ such that for all $n \geq n_0$, we have $f(n) \leq c \cdot g(n)$. This is equivalent to saying $f(n) = O(g(n))$.

- Similarly, $f(n) \gtrsim g(n)$ means that there exists a positive constant $c$ and a positive integer $n_0$ such that for all $n \geq n_0$, we have $f(n) \geq c \cdot g(n)$. This is equivalent to saying $g(n) = O(f(n))$.

- We write $f(n) \asymp g(n)$ if both $f(n) \lesssim g(n)$ and $f(n) \gtrsim g(n)$ hold. This means that $f(n)$ and $g(n)$ are of the same order.

- The notation $f(n) = o(g(n))$ means that for every positive constant $\epsilon$, there exists a positive integer $n_0$ such that for all $n \geq n_0$, we have $|f(n)| \leq \epsilon \cdot |g(n)|$.

These notations allow us to express the asymptotic behavior of functions concisely, which is particularly useful in our analysis of algorithmic complexity and error bounds.

## B.1. Preliminaries of t-Singular Value Decomposition

Due to space limitations, some concepts related to t-SVD were omitted in the main text. We provide additional notions here.

**Definition B.1** (Frontal-slice-wise product (Lu et al., 2019a)). The frontal-slice-wise product of any two tensors $\underline{\mathbf{A}} \in \mathbb{R}^{d_1 \times d_2 \times m}$ and $\underline{\mathbf{B}} \in \mathbb{R}^{d_1 \times d_2 \times m}$, denoted by $\underline{\mathbf{A}} \odot \underline{\mathbf{B}}$, is defined as a tensor $\underline{\mathbf{T}}$ such that

$$\underline{\mathbf{T}}_{:,:,i} = \underline{\mathbf{A}}_{:,:,i} \cdot \underline{\mathbf{B}}_{:,:,i}, \ i \in [K]$$

where $\cdot$ denotes the standard matrix multiplication. The frontal-slice-wise product performs matrix multiplication on each frontal slice of the tensors, resulting in a new tensor.

**Definition B.2** ($M$-block-diagonal matrix). The $M$-block-diagonal matrix of any tensor $\underline{\mathbf{T}} \in \mathbb{R}^{d_1 \times d_2 \times m}$, denoted by $\bar{\mathbf{T}}$, is the block diagonal matrix whose diagonal blocks are the frontal slices of $M(\underline{\mathbf{T}})$:

$$\bar{\mathbf{T}} := \mathtt{bdiag}(M(\underline{\mathbf{T}})) := \begin{bmatrix} M(\underline{\mathbf{T}})_{:,:,1} & & & \\ & M(\underline{\mathbf{T}})_{:,:,2} & & \\ & & \ddots & \\ & & & M(\underline{\mathbf{T}})^{(K)} \end{bmatrix} \in \mathbb{R}^{d_1 m \times d_2 m}.$$

This concept arranges the slices of a tensor in the frequency domain into a block diagonal matrix, facilitating the theoretical analysis of t-SVD.

We further provide some definitions and properties related to t-SVD:

**Definition B.3.** (Kernfeld et al., 2015) The t-transpose of a tensor $\underline{\mathbf{T}} \in \mathbb{R}^{d_1 \times d_2 \times m}$ under the $M$ transform (as shown in (1)), denoted by $\underline{\mathbf{T}}^\top$, satisfies

$$M(\underline{\mathbf{T}}^\top)_{:,:,i} = \left(M(\underline{\mathbf{T}})_{:,:,i}\right)^\top, \ i \in [K].$$

In other words, the t-transpose performs a transpose on each slice in the frequency domain and then transforms back to the time domain. This operation is one of the foundations of t-SVD theory.

**Definition B.4.** (Kernfeld et al., 2015) The t-identity tensor $\underline{\mathbf{I}} \in \mathbb{R}^{d \times d \times m}$ under the $M$ transform satisfies that each frontal slice of $M(\underline{\mathbf{I}})$ is an $m \times m$ identity matrix, i.e.,

$$M(\underline{\mathbf{I}})_{:,:,i} = \mathbf{I}, \ i \in [K].$$

It is easy to verify that $\underline{\mathbf{T}} *_M \underline{\mathbf{I}} = \underline{\mathbf{T}}$ and $\underline{\mathbf{I}} *_M \underline{\mathbf{T}} = \underline{\mathbf{T}}$ hold for appropriate dimensions. The t-identity tensor plays a role similar to the identity matrix in t-SVD.

**Definition B.5.** (Kernfeld et al., 2015) A tensor $\underline{\mathbf{Q}} \in \mathbb{R}^{d \times d \times m}$ is called t-orthogonal under the $M$ transform if it satisfies

$$\underline{\mathbf{Q}}^\top *_M \underline{\mathbf{Q}} = \underline{\mathbf{Q}} *_M \underline{\mathbf{Q}}^\top = \mathbf{I}.$$

T-orthogonality is an important property of tensor transformations, ensuring that the inner product and norm of tensors remain invariant before and after the transformation.

### B.2. Additional Lemmas

The concept of covering and packing numbers play an important role in our remaining analysis.

**Definition B.6** (Covering and Packing Numbers (Raskutti et al., 2011))**.** Consider a compact metric space consisting of a set $\mathcal{S}$ and a metric $\varrho : \mathcal{S} \times \mathcal{S} \to \mathbb{R}^+$

- An $\epsilon$-covering of $\mathcal{S}$ with respect to the metric $\varrho$ is a collection $\{\underline{\mathbf{L}}^1, \dots, \underline{\mathbf{L}}^N\} \subset \mathcal{S}$ such that for all $\underline{\mathbf{L}} \in \mathcal{S}$, there exists some $\underline{\mathbf{L}}^i, i \in \{1, \dots, N\}$ with $\varrho(\underline{\mathbf{L}}, \underline{\mathbf{L}}^i) \leq \epsilon$. The $\epsilon$-covering number $N(\epsilon; \mathcal{S}, \varrho)$ is the cardinality of the smallest $\epsilon$-covering.

- A $\delta$-packing of $\mathcal{S}$ with repsect to the metric $\varrho$ is a collection $\{\underline{\mathbf{L}}^1, \dots, \underline{\mathbf{L}}^M\} \subset \mathcal{S}$ such that $\varrho(\underline{\mathbf{L}}^i, \underline{\mathbf{L}}^j) > \delta$ for all distinct $i, j$. The $\delta$-packing number $M(\delta; \mathcal{S}, \varrho)$ is the cardinality of the largest $\delta$-packing.

Covering and packing numbers provide essentially the same measure of the massiveness of a set. In particular, the relation between covering number and packing number is described as $M(2\epsilon; \mathcal{S}, \varrho) \leq N(\epsilon; \mathcal{S}, \varrho) \leq M(\epsilon; \mathcal{S}, \varrho)$. These two quantities exhibit the same scaling behavior as $\epsilon \to 0$. Additionally, the logarithm of the covering number $\log N(\epsilon; \mathcal{S}, \varrho)$ is known as the metric entropy of $\mathcal{S}$ with respect to $\varrho$.

**Definition B.7** (entropy number)**.** Consider a quasi-Banach space consisting a compact set $\mathcal{S}$ and a quasi-metric $\varrho$. $N(\epsilon; \mathcal{S}, \varrho)$ denotes the covering number with radius $\epsilon$. For $k = 1, 2, \dots$ the dyadic entropy number is defined as

$$\epsilon_k(\mathcal{S}, \varrho) := \inf\{\epsilon > 0 : N(\epsilon; \mathcal{S}, \varrho) \leq 2^{k-1}\}.$$

**Lemma B.8** (Entropy number of Schatten-$q$ ball (Hinrichs et al., 2017))**.** *Consider a $d \times d$-dimensional vector space. Suppose $\mathcal{S}$ is a $S_q$ unit-ball and $\varrho$ is the metric induced by F-norm. Then, we have the following theorem for $q \leq 2$:*

$$\epsilon_k(\mathbb{B}^d_{S_q}(1), \|\cdot\|_{\mathrm{F}}) \asymp_q \begin{cases} 1 & \text{for } 1 \leq k \leq d & \text{(11a)} \\[2mm] \left(\dfrac{d}{k}\right)^{\frac{1}{q}-\frac{1}{2}} & \text{for } d \leq k \leq d^2 & \text{(11b)} \\[2mm] 2^{-\frac{k}{d^2}} \cdot d^{\frac{1}{2}-\frac{1}{q}} & \text{for } k \geq d^2. & \text{(11c)} \end{cases}$$

## C. Theoretical Results for Understanding of Dual-Level Sparsity for GLM

We focus on two representative modeling regimes for dual spectral sparsity: a fully hard-constrained model that enforces exact frequency sparsity and per-slice low-rankness, and a fully soft $\ell_p(S_q)$ model that allows gradual spectral decay across both levels. These two regimes correspond to the two ends of the modeling spectrum and suffice to capture the core statistical phenomena of dual spectral regularization.

Accordingly, this appendix provides the detailed minimax analysis for these two parameter spaces. The hard model clarifies the separate contributions of frequency selection and per-slice rank constraints, whereas the soft $\ell_p(S_q)$ model characterizes the estimation complexity when both frequency sparsity and within-slice spectral decay are relaxed continuously. Across both regimes, the minimax risk reflects two sources of statistical difficulty: (i) the difficulty of identifying informative frequency components, and (ii) the difficulty of estimating the low-rank or spectrally decaying structure within active components.

**Three Dual Spectral Sparsity Assumptions.** We consider three progressively relaxed sparsity models for $\underline{\mathbf{L}}^*$, which together form a hierarchy of dual spectral structures.

**A1.** *Hard dual spectral sparsity ($\ell_0$-$S_0$ case).* Let $\underline{\mathbf{L}}^*$ belong to the parameter space

$$\mathbf{T}_{0,0}(s,r) = \Big\{ \underline{\mathbf{L}} : (s,r)\text{-dual sparse} \Big\}, \tag{12}$$

meaning that $\underline{\mathbf{L}}$ has at most $s$ active frequency slices in the $M$-domain, and each active slice has rank at most $r$. This model enforces exact inter-frequency sparsity and exact intra-frequency low-rankness.

**A2.** *Fully soft dual spectral sparsity ($\ell_p$-$S_q$ case).* Let $\underline{\mathbf{L}}^*$ belong to the $\ell_p(S_q)$-ball

$$\mathbf{T}_{p,q}(R) = \Big\{ \underline{\mathbf{L}} : \|\underline{\mathbf{L}}\|_{\ell_p(S_q)}^p \le R \Big\}, \tag{13}$$

where the $\ell_p(S_q)$ quasi-norm jointly promotes soft frequency selection and gradual within-slice spectral decay.

Our goal is to characterize the fundamental estimation limits under each setting.

**(I) Minimax Lower Bounds Over $\ell_0$-$S_0$-Constraint.** We first characterizes the *worst-case* error any estimator must incur under hard dual-level sparse constraints.

**Theorem C.1.** *Consider the linear observation model* $\underline{\mathbf{Y}}_i = \underline{\mathbf{L}}^* + \underline{\mathbf{E}}_i$ *under dual-level sparse spectrum, with $n$ i.i.d. observations and noise variance $\sigma^2$. If $\underline{\mathbf{L}}^* \in \mathbf{T}_{0,0}(s,r)$, any measurable estimator $\hat{\underline{\mathbf{L}}}$ satisfies*

$$\inf_{\hat{\underline{\mathbf{L}}}} \sup_{\underline{\mathbf{L}} \in \mathbf{T}_{0,0}(s,r)} \mathbb{P}\Big( \|\hat{\underline{\mathbf{L}}} - \underline{\mathbf{L}}\|_{\mathrm{F}}^2 \ge C_\ell \frac{\sigma^2}{n} \big[ s\, \log\big(\tfrac{em}{s}\big) + s\, r\, d \big] \Big) \ge \tfrac{1}{2},$$

*implying*

$$\mathfrak{M}(\mathbf{T}_{0,0}(s,r)) \ge \tfrac{1}{2}\, C_\ell \frac{\sigma^2}{n} \Big[ s\, \log\big(\tfrac{em}{s}\big) + s\, r\, d \Big].$$

From the expressions in Theorem C.1, we see that the *estimation complexity* has two components:

 (a) A term of order $s\log\big(\tfrac{em}{s}\big)$ captures the combinatorial cost of identifying which $s$ out of $m$ frequency slices are nonzero.

 (b) A second term $(srd)$ quantifies the difficulty of *estimating each slice*.

Hence, Theorem C.1 highlights a fundamental trade-off in learning dual-level sparse structures from noisy observations.

**(II) Minimax Upper Bounds Over $\ell_0$-$S_0$-Ball.** We next confirm that these lower bounds are sharp by analyzing a *constrained least-squares* (CLS) estimator defined via

$$\hat{\underline{\mathbf{L}}}_0 \in \arg\min_{\underline{\mathbf{L}} \in \mathbf{T}_{0,0}(s,r)} \|\underline{\mathbf{L}} - \bar{\underline{\mathbf{Y}}}\|_F^2, \tag{14}$$

where $\bar{\underline{\mathbf{Y}}} = \frac{1}{n}\sum_{i=1}^n \underline{\mathbf{Y}}_i$. One shows that $\hat{\underline{\mathbf{L}}}_0$ attains an $\varepsilon$-accurate solution with high probability as soon as $\varepsilon^2$ is on the same order as the lower-bound terms in Theorem C.1.

**Theorem C.2.** *Under the same dual-level sparse setups and i.i.d. noise model, the following hold: We form the estimator* $\hat{\underline{\mathbf{L}}}_0$ *by minimizing* $\|\underline{\mathbf{L}} - \bar{\underline{\mathbf{Y}}}\|_{\mathrm{F}}^2$ *over* $\mathbf{T}_{0,0}(s,r)$*, then for any* $\epsilon^2 \geq C_u \frac{\sigma^2}{n} \left[ s \log(\frac{em}{s}) + s\,r\,d \right]$*,*

$$\sup_{\underline{\mathbf{L}} \in \mathbf{T}_{0,0}(s,r)} \|\hat{\underline{\mathbf{L}}}_0 - \underline{\mathbf{L}}\|_{\mathrm{F}}^2 \leq \epsilon^2, \tag{15}$$

*with probability at least* $1 - C_1 \exp(-C_2\, n\, \epsilon^2)$.

The upper bound match the lower bound (Theorem C.1) up to constant factors, establishing the $\ell_0$-$S_0$ rate as *optimal*. Consequently, we arrive at the minimax rate:

$$\mathfrak{M}(\mathbf{T}_{0,0}(s,r)) \asymp \frac{\sigma^2}{n} \left[ s\, \log(\tfrac{em}{s}) + s\,r\,d \right].$$

Thus, Theorem C.2 shows that the above CLS estimator is *rate-optimal*.

**(III) Minimax Rates Over General $\ell_p(S_q)$-Balls.** Finally, we move beyond the hard-sparsity constraints to the more flexible $\ell_p(S_q)$ spaces:

$$\mathbf{T}_{p,q}(R) = \left\{ \underline{\mathbf{L}} : \; \left\| \underline{\mathbf{L}} \right\|_{\ell_p(S_q)}^p \leq R \right\}.$$

By combining similar lower/upper bound arguments (now requiring more subtle entropy and covering results) we arrive at Theorem C.4. To avoid an overly detailed discussion of degenerate cases, we follow the analytical strategies adopted in Raskutti et al. (2011) for $\ell_q$ quasi-norms and in Li et al. (2024) for $\ell_u(\ell_q)$ quasi-norms, and impose the following conditions:

$$\begin{cases} \log m < R(\frac{n}{\sigma^2 d})^{\frac{p}{2}} \leq \frac{m}{\log m} \\ \log m < R(\frac{n}{\sigma^2 \log m})^{\frac{p}{2}} \leq \frac{m}{\log m} \\ c_1 d < R^{\frac{q}{\sigma^2 p}}(\frac{n}{d})^{\frac{q}{2}} < C_1 d. \end{cases} \tag{16}$$

*Remark* C.3 (On nondegenerate scaling regimes). The conditions in Eq. (16) are introduced solely to localize the minimax analysis to statistically nondegenerate scaling regimes, and are not additional structural assumptions. They rule out extreme parameter ranges in which the risk either saturates at trivial levels or collapses to uninformative bounds, while preserving the dependence on the ambient dimensions. Such scaling conditions are standard in minimax analyses with quasi-norm constraints, and follow the analytical strategies adopted in Raskutti et al. (2011) for $\ell_q$ balls and in Li et al. (2024) for mixed $\ell_u(\ell_q)$ constraints.

**Theorem C.4** (Minimax Rates for $\ell_p(S_q)$-balls). *Suppose $p, q \in (0, 1]$ and condition Eq. (16) holds to avoid degenerate parameter ranges. Then the minimax risk over the $\ell_p(S_q)$-ball is:*

$$\mathfrak{M}(\mathbf{T}_{p,q}(R)) = \inf_{\hat{\underline{\mathbf{L}}}} \sup_{\underline{\mathbf{L}}^* \in \mathbf{T}_{p,q}(R)} \mathbb{E}\left[ \|\hat{\underline{\mathbf{L}}} - \underline{\mathbf{L}}^*\|_{\mathrm{F}}^2 \right]$$

$$\asymp \begin{cases} R\left(\frac{n}{d\sigma^2}\right)^{\frac{p-2}{2}} + R\left(\frac{n}{\sigma^2 \log m}\right)^{\frac{p-2}{2}}, & p > q, \\[2mm] R^{\frac{q}{p}}\left(\frac{n}{d\sigma^2}\right)^{\frac{q-2}{2}} + R\left(\frac{n}{\sigma^2 \log m}\right)^{\frac{p-2}{2}}, & p \leq q, m > d^2, \\[2mm] R^{\frac{q}{p}}\left(\frac{n}{d\sigma^2}\right)^{\frac{q-2}{2}}, & p \leq q, m \leq d^2. \end{cases} \tag{17}$$

Examining Eq. (17) reveals three distinct regimes:

- $p > q$. The $\ell_p$-type group sparsity dominates, rendering the Schatten-$q$ penalties secondary.

- $p \leq q$, $m > d^2$. Both frequency-domain $\ell_p$-grouping and intra-slice Schatten-$q$-norm jointly control the risk, leading to a sum of two terms.

- $p \leq q$, $m \leq d^2$. The $S_q$-ball effectively saturates the error, making the risk independent of $m$.

Hence, $\ell_p(S_q)$ quasi-norms admit richer, *soft* sparsity structures beyond the hard $\ell_0$-$S_0$ constraint, but yield analogous minimax phenomena once careful covering-entropy or packing arguments are applied.

In conclusion, these results unify the lower and upper bounds for dual-level sparse tensor estimation under both hard-sparsity ($\ell_0$-$S_0$) and soft-sparsity ($\ell_p(S_q)$) constraints. The key takeaway is that the *minimax error* always balances identifying relevant frequencies with estimating each slice's rank structure. Hard- and soft-sparsity assumptions shift how these two aspects interact, but the big-picture story remains consistent: multi-frequency, low-rank modeling carries a fundamental combinatorial cost (for frequency selection) plus a continuous cost (for matrix parameter estimation). These findings rigorously justify the dual-level sparsity approach and characterize the fundamental limits of any estimator hoping to learn such structured tensors from noisy observations.

*Remark* C.5 (On the Practical Validation and Use of Theorem 4.2). While Theorem 4.2 establishes minimax lower bounds for dual-sparse tensor recovery under the $\ell_p$–Schatten-$q$ prior, directly validating these rates through simulation is inherently difficult. This is because minimax guarantees are defined over worst-case estimators, which are often inaccessible—especially under nonconvex regularization. Even for vector-valued estimators with $\ell_p(\ell_q)$ sparsity, constructing such minimax-optimal procedures remains an open challenge when $(p, q)$ are general.

### C.1. Sketch of Minimax Proofs under Dual-Level Sparsity

Our proofs of the minimax bounds proceed by considering three successively more general forms of dual-level sparsity, each requiring distinct technical arguments due to their underlying structural assumptions.

**(1) Hard–Hard Dual-Level Sparsity $\mathsf{T}_{0,0}(s, r)$.** When both the number of active frequency components and the rank of each active slice are bounded by $s$ and $r$, respectively, we construct a *packing set* that simultaneously encodes frequency sparsity and low-rankness:

1. *Support selection:* First, choose which $s$ frequency slices (out of $m$) can be nonzero, ensuring a combinatorial factor $\binom{m}{s}$.

2. *Within-slice low-rank matrices:* Next, for each chosen frequency slice, define a family of low rank $\leq r$ matrices whose entries are set to a suitable scale $\delta$, ensuring the separation properties

$$\|\underline{\mathbf{L}}^i - \underline{\mathbf{L}}^j\|_{\mathrm{F}}^2 \geq c_0\, s\, r\, d\, \delta^2 \quad \text{and} \quad \|\underline{\mathbf{L}}^i - \underline{\mathbf{L}}^j\|_{\mathrm{F}}^2 \leq 2\, s\, r\, d\, \delta^2,$$

while also achieving large enough cardinality for the packing set.

Applying Fano's inequality to this well-separated set establishes a lower bound by appropriately choosing $\delta \propto \sqrt{\frac{\sigma^2}{n}\left[s\log(\frac{em}{s}) + s\, r\, d\right]}$. For the matching upper bound, we employ a constrained least squares estimator and use covering number arguments (based on matrix rank$\leq r$ covers and frequency-support combinatorics) to show it achieves the same rate.

**(2) Soft–Soft Dual-Level Sparsity $\mathsf{T}_{p,q}(R)$.** Finally, in the most general setting, both frequency sparsity and low-rankness are relaxed into an $\ell_p(S_q)$ quasi-norm. Here, the analysis must carefully track how $p$ and $q$ govern *group-level* versus *within-group* regularity:

- When $p > q$, the $\ell_p$ penalty dominates, so the rates follow primarily from frequency-sparsity arguments.

- If $p \leq q$, we observe a phase transition depending on whether $m$ exceeds $d^2$. For $m > d^2$, the two penalties are both active, summing their respective complexities; for $m \leq d^2$, the Schatten-$q$ term saturates the error, rendering it independent of $m$.

Technically, this requires sophisticated entropy tools, in particular Schütt's theorem on entropy numbers for vector-valued sequence spaces, and a careful *chaining* analysis that tracks interactions across frequencies and singular values in each slice.

**Unified Upper Bound Approach.** Although the detailed proofs vary, the estimation upper bounds all follow a similar template via empirical process theory:

- For $\mathbf{T}_{0,0}(s, r)$, we rely on discrete packing/covering of finite-support rank$\leq r$ matrices.

- For $\mathbf{T}_{p,q}(R)$, we perform a chaining argument on $\ell_p(S_q)$ quasi-norms, applying Schütt-type entropy estimates.

In each case, bounding the least squares estimator's risk at the matching lower-bound rate confirms it is optimal up to constants.

Across these two regimes of *hard–hard* and *soft–soft* dual-level sparsity, the main technical challenge is to *precisely quantify the geometric complexity* imposed by the interplay of frequency sparsity and low-rankness (whether hard or soft). Once we derive suitable packing/covering or entropy bounds, standard empirical process arguments transform that complexity into matching lower and upper rates. Consequently, the minimax results show a consistent story: learning multi-frequency structures involves a combinatorial cost from selecting active frequencies plus a continuous cost from estimating each low-rank (or Schatten-$q$) slice, culminating in the optimal rates detailed in our main theorems.

### C.2. Proof of Theorem C.1

*Proof.* Consider the $\frac{srd}{32}$-packing set $\widetilde{\mathbf{T}}_{0,0}(s, r) = \{\underline{\mathbf{L}}^1, \ldots, \underline{\mathbf{L}}^N\}$ constructed in Lemma C.9, where $N$ is its cardinality. We set all nonzero entries of each $\underline{\mathbf{L}}^i \in \widetilde{\mathbf{T}}_{0,0}(s, r)$ to be 1, and define $\vartheta^i = \underline{\mathbf{L}}^i \, \delta$, with $\delta$ a parameter to be determined. Since each $M(\underline{\mathbf{L}}^i)$ has at most $s \, r \, d$ nonzero elements, for any $\vartheta^i \neq \vartheta^j$, it follows that

$$\|\vartheta^i - \vartheta^j\|_{\mathrm{F}}^2 = \|M(\vartheta^i) - M(\vartheta^j)\|_{\mathrm{F}}^2 = \delta^2 \|M(\underline{\mathbf{L}}^i) - M(\underline{\mathbf{L}}^j)\|_{\mathrm{F}}^2 \leq 2 \, s \, r \, d \, \delta^2, \quad \forall i, j \in [N]. \tag{18}$$

On the other hand, from the construction of $\widetilde{\mathbf{T}}_{0,0}(s, r)$, we also have

$$\|\vartheta^i - \vartheta^j\|_{\mathrm{F}}^2 \geq \tfrac{1}{32} \, s \, r \, d \, \delta^2, \quad \forall i, j \in [N]. \tag{19}$$

Using standard mutual information arguments (Wu, 2016) gives

$$\begin{aligned}
\mathsf{I}(\bar{\mathbf{Y}}; \psi) &\leq \frac{1}{\binom{N}{2}} \sum_{i \neq j} \mathsf{KL}(\vartheta^i \,\|\, \vartheta^j) \\
&= \frac{1}{\binom{N}{2}} \sum_{i \neq j} \frac{n}{2 \, \sigma^2} \|\vartheta^i - \vartheta^j\|_{\mathrm{F}}^2 \\
&\leq \frac{n}{\sigma^2} \, s \, r \, d \, \delta^2,
\end{aligned} \tag{20}$$

where $\mathsf{KL}(\cdot \,\|\, \cdot)$ is the Kullback–Leibler divergence, and the last step uses Eq. (18). By applying Fano's inequality (Cover, 1999), we obtain

$$\mathbb{P}(\hat{\vartheta} \neq \psi) \geq 1 - \frac{\frac{n}{\sigma^2} \, s \, r \, d \, \delta^2 + \log 2}{\log N},$$

where $\psi$ is uniformly distributed over the packing set $\widetilde{\mathbf{T}}_{0,0}(s, r)$. To ensure $\mathbb{P}(\hat{\vartheta} \neq \psi) \geq \frac{1}{2}$, it suffices to choose

$$\delta = \tfrac{1}{2} \sqrt{\left[ \left( c_1 \, r \, s \, d \right) + \left( c_2 \, s \, \log(\tfrac{em}{s}) \right) \right] \frac{\sigma^2}{s \, r \, d \, n}}.$$

Substituting this choice into Eq. (19) and invoking Lemma C.9 completes the proof, showing that

$$\inf_{\hat{\underline{\mathbf{L}}}} \sup_{\underline{\mathbf{L}} \in \mathbf{T}_{0,0}(s,r)} \mathbb{P}\left[ \|\hat{\underline{\mathbf{L}}} - \underline{\mathbf{L}}\|_{\mathrm{F}}^2 \geq \tfrac{c \sigma^2}{n} \left( r \, s \, d + s \, \log \tfrac{em}{s} \right) \right] \geq \tfrac{1}{2}.$$

Finally, a Markov's inequality argument yields the same lower bound in expectation. $\square$

C.2.1. THEORETICAL TOOLS: GILBERT-VARSHAMOV THEOREMS

The proof of case (a) requires a $\frac{srd}{32}$-packing set $\widetilde{\mathbf{T}}_{0,0}(s,r)$. Before proceeding with the construction of this packing set, we first introduce several versions of the Gilbert-Varshamov theorem that will be crucial for our analysis. These results provide guarantees on the existence of well-separated binary and $K$-ary codes.

The first version deals with binary codes containing the zero vector:

**Lemma C.6** (Gilbert-Varshamov Theorem for Binary Codes (Tsybakov, 2009)). *Consider a length-$m$ code with binary symbols (2-ary coding). There exists a subset $\Omega = \{\omega_0, \ldots, \omega_N\}$ of the code book where $\omega_i \in \{0,1\}^m$ that satisfies:*

- *The zero vector is included: $\omega_0 = (0, \ldots, 0)$.*

- *The minimum Hamming distance is bounded: $d_H(\omega_i, \omega_j) \geq m/8$ for all $0 \leq i < j \leq N$.*

- *The set size is exponential: $N \geq 2^{m/8}$.*

This lemma guarantees the existence of a large set of well-separated binary vectors, which will be used to construct the low-rank matrices $M(\underline{\mathbf{L}})_{:,:,k}$ within the $k$-th ($k \in [m]$) frequency component of $\underline{\mathbf{L}} \in \mathbb{R}^{d_1 \times d_2 \times m}$ in the transformed domain defined by $M$ in Eq. (1).

Then, for more general alphabets beyond $\{0,1\}$, we have:

**Lemma C.7** (Gilbert-Varshamov Theorem for $K$-ary Codes). *For a length-$m$ code with $K$-ary symbols, there exists a $\varrho$-separated set whose cardinality is at least:*

$$N_K(m, \varrho) \geq \frac{K^m}{\sum_{i=0}^{\varrho-1} \binom{m}{i}(K-1)^i}$$

*where $\varrho$ denotes the minimum Hamming distance between any two codewords.*

Further, for codes restricted to a Hamming sphere, we have:

**Lemma C.8** (Gilbert-Varshamov Theorem for Bounded-Weight Codes, (Wu, 2016)). *Consider the Hamming sphere of radius $s$ for a length-$m$ code with $K$-ary symbols. There exists a $\varrho$-separated set within this sphere with cardinality at least:*

$$N_K(m, s, \varrho) \geq \frac{\binom{m}{s}(K-1)^s}{\sum_{i=0}^{\varrho-1} \binom{m}{i}(K-1)^i} \tag{21}$$

A particularly useful special case occurs when $\varrho = c_1 s$ and we consider binary coding ($K = 2$):

$$N_2(m, s, \varrho) \geq (em/s)^{c_2 s} \tag{22}$$

where $c_1$ and $c_2$ are absolute constants.

**Lemma C.9** (Existence of a Packing Set). *There exists a packing set $\widetilde{\mathbf{T}}_{0,0}(s,r) \subset \mathbf{T}_{0,0}(s,r)$ satisfying the following properties:*

1. *Frequency sparsity constraint: Each tensor in $\widetilde{\mathbf{T}}_{0,0}(s,r)$ has at most $s$ nonzero frequency components.*

2. *Low-rank structure: Each frequency component is represented by a matrix of rank at most $r$.*

3. *Separation property: Any two distinct tensors $\underline{\mathbf{L}}^1, \underline{\mathbf{L}}^2 \in \widetilde{\mathbf{T}}_{0,0}(s,r)$ satisfy:*

$$\left\|\underline{\mathbf{L}}^1 - \underline{\mathbf{L}}^2\right\|_{\mathrm{F}}^2 \geq \frac{srd_1}{32}. \tag{23}$$

4. *Cardinality: The packing set has size at least:*

$$|\widetilde{\mathbf{T}}_{0,0}(s,r)| \geq (em/s)^{cs} \cdot 2^{rsd_1/32} \geq \exp(c_1 rsd_1 + c_2 s \log(em/s)). \tag{24}$$

*Proof.* We now construct our packing set $\widetilde{\mathbf{T}}_{0,0}(s,r) \subset \mathbf{T}_{0,0}(s,r)$ through a sequence of carefully designed steps. This packing set must be sufficiently rich to capture the essential complexity of $\mathbf{T}_{0,0}(s,r)$. The construction must simultaneously enforce *frequency sparsity bounded by s*, maintain the *low-rank structure within each frequency component characterized by r*, and ensure *good separation properties* across all constructed sub-sets at each step.

**Step 1: Selection of Non-zero Frequencies.** First, we determine a subset $\widetilde{\mathbf{\Gamma}} \subset \mathbf{\Gamma}$ of the support of the $m$ frequency components. This step establishes the *frequency sparsity pattern* of our tensors $\underline{\mathbf{L}}$ in the transformed domain defined by $M(\cdot)$. We proceed as follows:

1) *Ensure frequency sparsity*: First, we use a 2-ary code of length $m$ on the Hamming sphere of radius $s$ to represent possible support patterns.

2) *Ensure sufficient separation*: Second, by requiring the code to be $s/4$-separated, we can guarantee the existence of a set $\widetilde{\mathbf{\Gamma}}$ with a minimum number of frequency patterns by Lemma C.8:

$$N_2(m,s,s/4) \geq (em/s)^{cs} \geq \exp(cs\log(em/s)) \tag{25}$$

where $c$ is an absolute constant.

**Step 2: Construction of Low-rank Matrices.** We then consider constructing appropriate low-rank matrices. Without loss of generality, assume $d_1 \geq d_2$. Motivated by Klopp (2015), we first construct the set of matrices $\mathbf{A}_{\text{low-rank}}$ as follows:

For positions $(i,j)$ of a matrix $\mathbf{A} \in \mathbf{A}_{\text{low-rank}}$ with $i \leq r$ and $j \in [d_2]$, we set $\mathbf{A}_{i,j} \in \{0,1\}$, and for all other positions (i.e., $i > r$), we set $\mathbf{A}_{i,j} = 0$.

Then, we can ensure that $\text{rank}(\mathbf{A}) \leq r, \forall \mathbf{A} \in \mathbf{A}_{\text{low-rank}}$. Further by Lemma C.6, this construction yields a $\{0,1\}$-code of length $rd_1$, for which we can find a subset $\{\mathbf{0}, \mathbf{A}^1, \ldots, \mathbf{A}^{N_0}\}$ satisfying:

i) *Sufficiently many low-rank patterns*: $N_0 \geq 2^{rd_1/8}$.

ii) *Sufficient separation*: The Hamming distance $d_H(\mathbf{A}^i, \mathbf{A}^j) \geq rd_1/8$ for all $0 \leq i < j \leq N_0$.

In sumamry, by construction, we can find the matrix set $\widetilde{\mathbf{A}}_{\text{low-rank}} := \{\mathbf{A}^1, \ldots, \mathbf{A}^{N_0}\}$ that satisfy:

$$|\widetilde{\mathbf{A}}_{\text{low-rank}}| \geq 2^{rd_1/8}, \quad \text{rank}(\mathbf{A}^i) \leq r, \quad \|\mathbf{A}^i\|_{\text{F}}^2 \leq rd_1, \ \|\mathbf{A}^i - \mathbf{A}^j\|_{\text{F}}^2 \geq rd_1/8, \quad \text{and} \quad \|\mathbf{A}^i\|_{\text{F}}^2 \geq rd_1/8. \tag{26}$$

**Step 3: Assign Low-rank Patterns for a Fixed Frequency Pattern.** Now have founded the $s$-sparsity frequency support patterns in $\widetilde{\mathbf{\Gamma}}$ and the $r$-low-rank frequency matrices in $\widetilde{\mathbf{A}}_{\text{low-rank}}$. We then need to assign appropriate low-rank patterns to each selected frequency $\gamma \in \widetilde{\mathbf{\Gamma}}$. This step requires careful consideration of both the separation properties and the cardinality of our construction.

Motivated by the proof of Lemma 3 in Li et al. (2024), we consider the following analysis:

1) From **Step 2**, we know that each frequency component can take $|\widetilde{\mathbf{A}}_{\text{low-rank}}| \geq 2^{rd_1/8}$ different low-rank patterns. This gives us a large alphabet size for coding each frequency component.

2) We can view this as a $|\widetilde{\mathbf{A}}_{\text{low-rank}}|$-ary coding problem for each selected frequency, where we also need to ensure the resulting codes are $s/2$-separated in Hamming distance.

3) For a fixed frequency sparsity pattern $\gamma \in \widetilde{\mathbf{\Gamma}}$, we can lower bound the **cardinality of the resulting set $\widetilde{\mathbf{T}}_\gamma$** according to Lemma C.7 as follows:

$$\begin{aligned}
N_{|\widetilde{\mathbf{A}}_{\text{low-rank}}|}(s,s/2) &\geq \frac{(2^{rd_1/8})^s}{\sum_{i=0}^{s/2-1} \binom{s}{i}(2^{rd_1/8}-1)^i} \\
&\geq \frac{(2^{rd_1/8})^s}{\sum_{i=0}^{s/2-1} \binom{s}{i}(2^{rd_1/8})^{s/2-1}} \\
&\geq \frac{(2^{rd_1/8})^s}{2^s \cdot (2^{rd_1/8})^{s/2-1}} \\
&= \frac{(2^{rd_1/8})^{s/2+1}}{2^s} \\
&= 2^{rsd_1/16-s} \geq 2^{rsd_1/32}
\end{aligned} \tag{27}$$

The inequalities above are derived through the following steps. The first inequality follows from the application of the Gilbert-Varshamov theorem for $K$-ary codes. The second inequality is obtained by upper bounding $(2^{rd_1/8})^i - 1$ with $(2^{rd_1/8})^{s/2-1}$. The third inequality results from bounding the sum of binomial coefficients by $2^s$. Finally, the last inequality holds under the assumption that $rd_1$ is sufficiently large, specifically $rd_1 \geq 32$.

**Step 4: Integration of Frequency Sparsity and Low-rankness.** Now, for each $\gamma \in \tilde{\Gamma}$, we have found a set of tensors $\tilde{\mathbf{T}}_\gamma$ specified by $\gamma$. Then, we totally found a set $\tilde{\mathbf{T}}$ of at least $|\tilde{\Gamma}| \cdot 2^{rsd_1/32}$ tensors. Next, we will show that $\tilde{\mathbf{T}}$ the ideal packing set $\tilde{\mathbf{T}}_{0,0}(s,r)$ of $\mathbf{T}_{0,0}(s,r)$ we are looking for. We consider two cases:

*Case 4.1: Different Frequency Sparsity Patterns* When any two tensors $\underline{\mathbf{L}}^1, \underline{\mathbf{L}}^2 \in \tilde{\mathbf{T}}$ in our construction have different frequency supports $\gamma^1, \gamma^2 \in \tilde{\Gamma}$, then according to the construction of $\tilde{\Gamma}$, the frequency supports $\gamma^1$ and $\gamma^2$ are $s/2$-separated. That means the tensors $\underline{\mathbf{L}}^1, \underline{\mathbf{L}}^2$ have at least $s/4$ different frequency positions, thus the distance

$$\left\| \underline{\mathbf{L}}^1 - \underline{\mathbf{L}}^2 \right\|_F^2 = \left\| M(\underline{\mathbf{L}}^1) - M(\underline{\mathbf{L}}^2) \right\|_F^2 \geq \frac{s}{4} \cdot \frac{rd_1}{8} = \frac{srd_1}{32}$$

.

*Case 4.2: Same Frequency Sparsity Pattern* When any two tensors $\underline{\mathbf{L}}^1, \underline{\mathbf{L}}^2 \in \tilde{\mathbf{T}}$ in our construction have the same frequency support $\gamma \in \tilde{\Gamma}$, then according to the construction of $\tilde{\mathbf{T}}_\gamma$, there are at least $s/2$ different low-rank patterns of $\underline{\mathbf{L}}^1, \underline{\mathbf{L}}^2$. That means the tensors $\underline{\mathbf{L}}^1, \underline{\mathbf{L}}^2$ have at least $s/2$ frequency positions that are $rd_1/8$ separated, thus the distance

$$\left\| \underline{\mathbf{L}}^1 - \underline{\mathbf{L}}^2 \right\|_F^2 = \frac{s}{2} \cdot \frac{rd_1}{8} = \frac{srd_1}{16}$$

Combining both cases, we can conclude that the cardinality of our constructed set is at least:

$$|\tilde{\mathbf{T}}| = \prod_{\gamma \in \tilde{\Gamma}} |\tilde{\mathbf{T}}_\gamma| \geq (em/s)^{cs} \cdot 2^{rsd_1/32} \geq \exp(c_1 rsd_1 + c_2 s \log(em/s)), \tag{28}$$

which completes the proof. $\square$

## C.3. Proofs of Theorem C.2

Before formally providing the proof of upper bounds, we provide some useful technical lemmas.

### C.3.1. TECHNICAL LEMMAS FOR UPPER BOUNDS

We first prove the upper bounds for the covering number of the parameter spaces.

**Lemma C.10** (Upper bounds for the covering number). *Denote $N(\mathbf{T}, \|\cdot\|_F, \varepsilon)$ as the $\varepsilon$-covering number of parameter set $\mathbf{T}$. For $q = 0$ and $\varepsilon \in (0,1]$, let $\mathbb{S}^{md^2-1} := \{\underline{\mathbf{L}} \in \mathbb{R}^{d \times d \times m} : \|\underline{\mathbf{L}}\|_F = 1\}$, then we have*

$$\log N(\varepsilon; \mathbf{T}_{0,0}(s,r) \cap \mathbb{S}^{d^2 m-1}, \|\cdot\|_F) \leq s \log \frac{em}{s} + 2srd \log \frac{1}{\varepsilon/\sqrt{s}}.$$

*Proof.* Any tensor $\underline{\mathbf{L}} \in \mathbf{T}_{0,0}(s,r)$ has at most $s$ nonzero frontal slices in the transformed domain $M(\cdot)$. Denote the nonzero slice indices by

$$\Gamma(\underline{\mathbf{L}}) = \{ i \in [m] : M(\underline{\mathbf{L}})_{:,:,i} \neq \mathbf{0} \}.$$

Since $|\Gamma(\underline{\mathbf{L}})| \leq s$, the total number of ways to pick these supports is bounded by $\binom{m}{s}$. Therefore,

$$\log \binom{m}{s} \leq s \log\left(\frac{em}{s}\right).$$

This accounts for selecting *which* frequency slices are potentially nonzero.

For each fixed support $\gamma \subseteq [m]$ with $|\gamma| \leq s$, we need to cover the set of matrices $\{\mathbf{A} \in \mathbb{R}^{d \times d} : \|\mathbf{A}\|_F \leq 1, \text{rank}(\mathbf{A}) \leq r\}$ in the $\|\cdot\|_F$ metric by balls of radius $\varepsilon/\sqrt{s}$. It is a standard fact (see, e.g., Recht et al. (2007)) that the $\delta$-covering number for rank$\leq r$ matrices of Frobenius norm at most 1 is upper-bounded by

$$N\left(\delta; \{\mathbf{A} : \|\mathbf{A}\|_F \leq 1, \text{rank}(\mathbf{A}) \leq r\}, \|\cdot\|_F\right) \leq \exp\left(C r d \log(\tfrac{1}{\delta})\right),$$

for some constant $C > 0$. Substituting $\delta = \varepsilon/\sqrt{s} \le 1$ gives a covering number of order

$$\exp\!\Big( C\, r\, d\, \log\big(\tfrac{\sqrt{s}}{\varepsilon}\big) \Big).$$

Taking logarithms yields

$$\log N\!\Big( \tfrac{\varepsilon}{\sqrt{s}};\, \{\mathbf{A} : \|\mathbf{A}\|_{\mathrm{F}} \le 1, \mathrm{rank}(\mathbf{A}) \le r\},\, \|\cdot\|_{\mathrm{F}} \Big) \lesssim r\, d\, \log\!\Big( \tfrac{\sqrt{s}}{\varepsilon} \Big).$$

Given $s$ nonzero frequency slices, each can be approximated by a matrix in the covering set with radius $\varepsilon/\sqrt{s}$. Since there are at most $s$ active slices, the total squared error in Frobenius norm sums up to at most $s \cdot (\varepsilon/\sqrt{s})^2 = \varepsilon^2$. Hence,

$$\text{Total covering number} \le \binom{m}{s} \times \Big[ \exp\!\Big( C\, r\, d\, \log\big(\tfrac{\sqrt{s}}{\varepsilon}\big) \Big) \Big]^{s}.$$

Taking logarithms and using $\log \binom{m}{s} \le s \log(\tfrac{em}{s})$ completes the argument:

$$\log N\!\Big( \varepsilon;\, \mathbf{T}_{0,0}(s,r) \cap \mathbb{S}^{md^2-1},\, \|\cdot\|_{\mathrm{F}} \Big) \le s \log\!\Big( \tfrac{em}{s} \Big) + s \Big[ r\, d\, \log\big(\tfrac{\sqrt{s}}{\varepsilon}\big) \Big] = s \log\!\Big( \tfrac{em}{s} \Big) + 2\, s\, r\, d\, \log\!\Big( \tfrac{1}{\varepsilon/\sqrt{s}} \Big),$$

possibly absorbing constants into the notation. $\qquad\square$

**Refined Statement and Explanation.** We begin by defining a function $f(\mathbf{T}; \mathcal{X})$ for a tensor $\mathbf{T} \in \mathbb{R}^{d \times d \times m}$ and some data structure (or random tensor) $\mathcal{X}$. We consider a constrained supremum

$$\sup_{\varrho(\mathbf{T}) \le \nu, \mathbf{T} \in \mathsf{T}} f(\mathbf{T};\, \mathcal{X}),$$

where $\varrho : \mathbb{R}^{d \times d \times m} \to \mathbb{R}^{+}$ is an *increasing* constraint function and $\mathsf{T}$ is any nonempty collection of tensors. Let $\nu > 0$ be a fixed threshold, and define the event

$$\mathcal{E} := \big\{ \mathcal{X} : \exists\, \mathbf{T} \in \mathsf{T} \text{ such that } f(\mathbf{T};\, \mathcal{X}) \ge 2\, g\big(\varrho(\mathbf{T})\big) \big\},$$

where $g : \mathbb{R} \to \mathbb{R}^{+}$ is strictly increasing. Our aim is to bound $\mathbb{P}(\mathcal{E})$, i.e. the probability that there is some tensor $\mathbf{T}$ with constraint $\varrho(\mathbf{T}) \le \nu$ for which $f(\mathbf{T}; \mathcal{X})$ exceeds $2\, g\big(\varrho(\mathbf{T})\big)$. This setup is quite general: for example, $f$ might be a residual or cost function in an empirical process framework, $\varrho(\mathbf{T})$ might measure the size or norm of $\mathbf{T}$, and $g$ could be a nondecreasing penalty or bound we wish to enforce.

**Peeling Bound.** Lemma 9 of Raskutti et al. (2011), reproduced below, gives a powerful "peeling"-type argument. It says that if we can control

$$\mathbb{P}\Big[ \sup_{\mathbf{T}:\varrho(\mathbf{T}) \le \nu} f(\mathbf{T}; \mathcal{X}) \ge g(\nu) \Big] \le 2\, \exp\!\big( -c\, a_n\, g(\nu) \big)$$

for some constants $c > 0$ and $a_n > 0$, then one can derive a stronger tail bound for $\mathbb{P}(\mathcal{E})$. Formally:

**Lemma C.11** (Peeling, Lemma 9 of Raskutti et al. (2011))**.** *Suppose that for all $\nu \ge 0$, we have $g(\nu) \ge \mu$. Then there exists a constant $c > 0$ such that for all $\nu > 0$,*

$$\mathbb{P}\Big[ \sup_{\mathbf{T} \in \mathsf{T},\, \varrho(\mathbf{T}) \le \nu} f(\mathbf{T}; \mathcal{X}) \ge g(\nu) \Big] \le 2\, \exp\!\big( -c\, a_n\, g(\nu) \big).$$

*Hence,*

$$\mathbb{P}(\mathcal{E}) = \mathbb{P}\Big( \exists\, \mathbf{T} \in \mathsf{T} :\, f(\mathbf{T}; \mathcal{X}) \ge 2\, g(\varrho(\mathbf{T})) \Big) \le \frac{2\, \exp\!\big( -4\, c\, a_n\, \mu \big)}{1 - \exp\!\big( -4\, c\, a_n\, \mu \big)}.$$

**Why "Peeling" is Useful.** This lemma effectively "peels" off the largest values of $\varrho(\mathbf{T})$ in layers and bounds the supremum in each layer by $g(\nu)$. One then aggregates or unions over these layers to control the probability that $f(\mathbf{T}; \mathcal{X})$ can exceed $2\, g(\varrho(\mathbf{T}))$ for *any* $\mathbf{T}$. Such arguments often appear in minimax or empirical process proofs, where one partitions the parameter space according to $\varrho(\mathbf{T})$.

**Additional Lemmas for Dual-Level Sparse Spectra.** In our dual-level sparse setting, we require more specialized versions of standard covering or chaining arguments. For instance, Lemma C.12 below extends Lemma 6 of Raskutti et al. (2011) to handle an $\ell_0$-type frequency-sparsity set with $\max(s)$ active slices and $\max(r)$ rank constraints:

$$\mathbf{T}_{0,0}(2s, 2r) = \big\{\underline{\mathbf{L}} : \#\{\text{nonzero freq. slices}\} \leq 2s,\ \text{rank}\big(M(\underline{\mathbf{L}})_{:,:,i}\big) \leq 2r\big\}.$$

We also define

$$\widetilde{\mathsf{S}}\big(\mathbf{T}_{0,0}(2s, 2r), \rho\big) = \Big[\mathbf{T}_{0,0}(2s, 2r)\Big] \cap \Big\{\underline{\mathbf{L}} : \|\underline{\mathbf{L}}\|_{\mathrm{F}} \leq \rho\Big\}.$$

**Lemma C.12.** *There exist positive constants $C_1, C_2 > 0$ such that for any $\rho > 0$,*

$$\sup_{\underline{\mathbf{L}} \in \widetilde{\mathsf{S}}\big(\mathbf{T}_{0,0}(2s,2r),\, \rho\big)} \Big|\langle \underline{\bar{\mathbf{E}}}, \underline{\mathbf{L}}\rangle\Big| \leq C_u\, \sigma\, \rho\, \sqrt{\frac{1}{n}\Big[s\, \log\big(\tfrac{em}{s}\big) + s\, r\, d\Big]},$$

*with probability at least $1 - C_1 \exp\big(-C_2\,[s\log(\tfrac{em}{s}) + s\,r\,d]\big)$, where $\underline{\bar{\mathbf{E}}}$ is the (scaled) noise tensor.*

*Sketch of proof.* This follows from Lemma 6 of Raskutti et al. (2011) if we replace the covering number for a naive $\ell_0$-ball by the more specific covering number of $\mathbf{T}_{0,0}(2s, 2r)$. The detailed estimate of that covering number is provided by part (a) of Lemma C.10. Essentially, one shows that among all frequency-sparse and rank-limited tensors of Frobenius norm up to $\rho$, the uniform covering can be done with cardinality roughly $\exp\big[s\, \log(\tfrac{em}{s}) + s\, r\, d\log(\tfrac{1}{\varepsilon})\big]$, and then translates that into a tail bound on $\sup_{\underline{\mathbf{L}}}\langle \underline{\bar{\mathbf{E}}}, \underline{\mathbf{L}}\rangle$.

### C.3.2. PROOF OF THEOREM C.2

We analyze the constrained MLE estimator in Eq. (14). Note that due to optimality, the estimator $\hat{\underline{\mathbf{L}}}$ satisfies

$$\|\underline{\bar{\mathbf{Y}}} - \hat{\underline{\mathbf{L}}}\|_{\mathrm{F}}^2 \leq \|\underline{\bar{\mathbf{Y}}} - \underline{\mathbf{L}}^\star\|_{\mathrm{F}}^2,$$

rearranging terms gives

$$\|\hat{\underline{\mathbf{L}}} - \underline{\mathbf{L}}^\star\|_{\mathrm{F}}^2 \leq 2|\langle \underline{\bar{\mathbf{E}}}, \hat{\underline{\mathbf{L}}} - \underline{\mathbf{L}}^\star\rangle|. \tag{29}$$

*Proof.* Since both $\hat{\underline{\mathbf{L}}}_0$ and $\underline{\mathbf{L}}^\star$ belong to $\mathbf{T}_{0,0}(s, r)$, their difference satisfies

$$\hat{\underline{\mathbf{L}}}_0 - \underline{\mathbf{L}}^\star \in \mathbf{T}_{0,0}(2s, 2r).$$

Applying Lemma C.12, for any $\rho > 0$, we obtain

$$\sup_{\underline{\mathbf{L}} \in \widetilde{\mathsf{S}}(\mathbf{T}_{0,0}(2s,2r),\rho)} |\langle \underline{\bar{\mathbf{E}}}, \underline{\mathbf{L}}\rangle| \leq C_u \sigma \rho \sqrt{\frac{1}{n}\big(s\log \frac{em}{s} + srd\big)}$$

with probability at least $1 - C_1 \exp\{-C_2(s\log \frac{em}{s} + srd)\}$.

Next, consider the event $\mathcal{E}$ where there exists some $\underline{\mathbf{L}} \in \mathbf{T}_{0,0}(2s, 2r)$ such that

$$|\langle \underline{\bar{\mathbf{E}}}, \underline{\mathbf{L}}\rangle| \geq C_u \sigma \|\underline{\mathbf{L}}\|_{\mathrm{F}} \sqrt{\frac{1}{n}\big(s\log \frac{em}{s} + srd\big)}. \tag{30}$$

By Lemma C.11, the probability of this event satisfies

$$\mathbb{P}[\mathcal{E}] \leq \frac{2\exp(-C_3(s\log \frac{em}{s} + srd))}{1 - \exp(-C_3(s\log \frac{em}{s} + srd))}.$$

This follows by applying Lemma C.11 with function $f(\underline{\mathbf{T}}; \mathcal{X}) = \langle \underline{\bar{\mathbf{E}}}, \underline{\mathbf{L}} \rangle$, set $\mathbf{T} = \mathbf{T}_{0,0}(2s, 2r)$, sequence $a_n = n/\sigma^2$, function $\varrho(\underline{\mathbf{T}}) = \|\underline{\mathbf{T}}\|_{\mathrm{F}}$, and threshold function $g(\nu) = C_u \sigma \nu \sqrt{\frac{1}{n}(s \log \frac{em}{s} + srd)}$. For any $\nu \geq \sigma \sqrt{\frac{1}{n}(s \log \frac{em}{s} + srd)}$, we ensure that $g(\nu) \geq \frac{\sigma^2}{n}(s \log \frac{em}{s} + srd)$, allowing us to apply the lemma.

Combining Eq. (29) and Eq. (30) yields

$$\|\underline{\hat{\mathbf{L}}}_0 - \underline{\mathbf{L}}^\star\|_{\mathrm{F}}^2 \leq C_u \frac{\sigma^2}{n}(s \log \frac{em}{s} + srd).$$

This bound holds with probability at least $1 - C_1 \exp\{-C_2(s \log \frac{em}{s} + srd)\}$, completing the proof of Eq. (15). $\qquad\square$

## C.4. Proof of Theorem C.4

### C.4.1. LOWER BOUND FOR $\ell_p(S_q)$

*Proof.* We aim to prove a minimax lower bound under the dual-level sparse structure imposed by an $\ell_p(S_q)$ quasi-norm. Our proof strategy considers two key parameters that govern dual-level sparsity:

- *Frequency sparsity* $1 \leq s \leq m$, controlling how many frequency components can be nonzero.

- *Within-frequency low-rankness* $1 \leq r \leq d$, limiting the rank of each active frequency component.

*Subspace construction and parameter setting.* We begin with the subspace $\mathbf{T}_{0,0}(s, r)$ of tensors, as introduced in earlier sections, where both frequency indices and within-frequency ranks are hard-constrained. For any tensor in this subspace, suppose the absolute value of each nonzero entry is set to $\delta > 0$. This $\delta$ is chosen so that the $(p, q)$-quasi-norm constraint is satisfied, i.e.,

$$s \cdot \left( r \cdot (\delta \sqrt{d})^q \right)^{\frac{p}{q}} = R.$$

Solving for $\delta$ yields

$$\delta = R^{\frac{1}{p}} s^{-\frac{1}{p}} r^{-\frac{1}{q}} d^{-\frac{1}{2}}.$$

Thus, $\delta$ encodes how large each nonzero entry must be so that the tensor simultaneously satisfies frequency-sparsity and low-rankness in a consistent manner for the $\ell_p(S_q)$-norm. Notice that $\delta$ scales inversely with $s$, $r$, and $\sqrt{d}$, reflecting the interplay among frequency selection, rank constraints, and the Frobenius norm.

*Applying generalized Fano's inequality.* Let us express the probability that any estimator $\underline{\hat{\mathbf{L}}}$ incurs significant estimation error. Using a generalized Fano argument, we obtain

$$\inf_{\underline{\hat{\mathbf{L}}}} \sup_{\underline{\mathbf{L}}^* \in \mathbf{T}_{p,q}(R)} \mathbb{P}\left[ \|\underline{\hat{\mathbf{L}}} - \underline{\mathbf{L}}^*\|_F^2 \geq \frac{1}{16} R^{\frac{2}{p}} s^{1-\frac{2}{p}} r^{1-\frac{2}{q}} \right] \geq 1 - \frac{\frac{n}{\sigma^2} R^{\frac{2}{p}} s^{1-\frac{2}{p}} r^{1-\frac{2}{q}} + \log 2}{\frac{1}{4}(srd + s \log(\frac{m}{s}))}. \tag{31}$$

The denominator $srd + s \log(\frac{m}{s})$ captures (1) the cost of identifying $s$ nonzero frequency slices each of rank at most $r$, plus (2) the combinatorial complexity $s \log(\frac{m}{s})$ for subset selection among $m$ frequencies.

*Reformulating via linear programming.* To analyze Eq. (31) precisely, we adopt a linear programming approach similar to Li et al. (2024) for the $\ell_u(\ell_q)$-ball. Let $y = \log s$ and $x = \log r$. Then

$$R^{\frac{2}{p}} s^{1-\frac{2}{p}} r^{1-\frac{2}{q}} = \exp\left[ \log R^{\frac{2}{p}} + \left(1 - \frac{2}{p}\right) \log s + \left(1 - \frac{2}{q}\right) \log r \right].$$

Hence, maximizing $R^{\frac{2}{p}} s^{1-\frac{2}{p}} r^{1-\frac{2}{q}}$ is equivalent to maximizing

$$z = \left(1 - \frac{2}{p}\right) y + \left(1 - \frac{2}{q}\right) x,$$

subject to constraints bounding $x$ and $y$ (i.e. $0 \leq x \leq \log d$ and $0 \leq y \leq \log m$), plus an additional constraint from balancing numerator and denominator in Eq. (31). Concretely:

$$\begin{cases} 0 \leq x \leq \log d, \quad 0 \leq y \leq \log m, \\ y \geq \min\left\{ -\frac{p}{q} x + \log R + \frac{p}{2}\left(\log n - \log(\sigma^2) - \log d\right), -\left(\frac{p}{q} - \frac{p}{2}\right)x + \log R + \frac{p}{2}\left(\log n - \log(\sigma^2) - \log(\log m)\right) \right\}. \end{cases}$$

The first two lines capture $s \leq m, r \leq d$, while the last line encodes how $\frac{n}{\sigma^2} R^{\frac{2}{p}} s^{1-\frac{2}{p}} r^{1-\frac{2}{q}}$ compares with $s \, r \, d + s \log(\frac{m}{s})$ to ensure a valid Fano-type bound.

*Analyzing slopes and boundary points.* For convenience, define:

$$x_1 = \log R + \frac{p}{2} \log\left(\frac{n}{\sigma^2 \log m}\right), \quad x_2 = \log R + \frac{p}{2} \log\left(\frac{n}{\sigma^2 d}\right),$$

$$y_1 = \frac{q}{p} \log R + \frac{q}{2} \log\left(\frac{n}{\sigma^2 \log m}\right), \quad y_2 = \frac{q}{p} \log R + \frac{q}{2} \log\left(\frac{n}{\sigma^2 d}\right).$$

We then consider the lines:

- *Line A:* $y = -\frac{p}{q} x + \log R + \frac{p}{2}\left(\log n - \log(\sigma^2) - \log d\right)$, with slope $-\frac{p}{q}$ in the $(x, y)$-plane.

- *Line B:* $y = -\left(\frac{p}{q} - \frac{p}{2}\right) x + \log R + \frac{p}{2}\left(\log n - \log(\sigma^2) - \log(\log m)\right)$, whose slope is $\frac{p}{2} - \frac{p}{q}$.

- *Objective slope:* The slope of $z = \left(1 - \frac{2}{p}\right) y + \left(1 - \frac{2}{q}\right) x$ is $\frac{p(q-2)}{q(p-2)}$ in the $(x, y)$-plane.

A standard slope comparison yields these observations:

1. If $p > q$, the slope of $z$ is larger than slopes of Lines A and B, so the maximum of $z$ is attained at boundary points like $(0, y_2)$, $(0, y_1)$, or $(x_1, 0)$.

2. If $p \leq q$, the slope of $z$ is smaller than the slope of A but larger than that of B, so maxima can occur at $(x_1, 0)$, $(x_2, 0)$, or intersections of A and B, depending on $m$ vs. $d$.

Evaluating $z$ at these boundary points, one then obtains the resulting minimax lower bounds:

$$\begin{cases} R\left(\frac{n}{d\sigma^2}\right)^{\frac{p-2}{2}} + R\left(\frac{n}{\sigma^2 \log m}\right)^{\frac{p-2}{2}}, & \text{if } p > q, \\[2mm] R^{\frac{q}{p}}\left(\frac{n}{d\sigma^2}\right)^{\frac{q-2}{2}} + R\left(\frac{n}{\sigma^2 \log m}\right)^{\frac{p-2}{2}}, & \text{if } p \leq q, m \geq d^2, \\[2mm] R^{\frac{q}{p}}\left(\frac{n}{d\sigma^2}\right)^{\frac{q-2}{2}}, & \text{if } p \leq q, m \leq d^2. \end{cases}$$

These match precisely the piecewise expressions for the lower bound in the $\ell_p(S_q)$ setting. Hence, combining with the initial Fano-based argument Eq. (31) concludes the minimax lower bound proof.

$\square$

### C.4.2. COVERING NUMBER OF $\mathbb{B}_{\ell_p(S_q)}(R)$

Before deriving the upper bounds for $\ell_p(S_q)$, we first need to derive the covering number of $\mathbb{B}_{\ell_p(S_q)}(R)$ equipped with the $\ell_p(S_q)$-norm. To this end, we generalize Schütt's theorem for vector-valued sequence spaces (Edmunds & Netrusov, 2014). The analysis relies on entropy numbers and their relationships under different parameter ranges. We introduce several key lemmas and derive the upper bound for $e_k$.

**Lemma C.13** (Schütt's Theorem for Vector-valued Sequence Spaces (Edmunds & Netrusov, 2014))**.** *Let $X$ and $Y$ be $r$-normed quasi-Banach spaces, and let $0 < q < r \leq \infty$. The unit ball $\mathbb{B}_{\ell_q^m(X)}$ is defined as:*

$$\mathbb{B}_{\ell_q^m(X)} = v_1 \mathbb{B}_X \times v_2 \mathbb{B}_X \times \cdots \times v_m \mathbb{B}_X,$$

*where $\mathbb{B}_X$ is the unit ball with $X$-norm, and $v \in \mathbb{B}_q$. For $k, k_0 \in \mathbb{N}$ such that $k_0 \leq k$, let:*

$$D(k_0, k) = \max_{l \in \mathbb{N}, k_0 \leq l \leq k} \left(\frac{l}{k}\right)^{\frac{1}{q} - \frac{1}{r}} e_l(id : X \to Y),$$

$$A(k, m) = \max \left\{ \|id : X \to Y\| \left(\frac{\log(em/k)}{k}\right)^{\frac{1}{q} - \frac{1}{r}}, D(1, k) \right\},$$

*where $\|id : X \to Y\|$ denotes the operator norm, and $e_l(id : X \to Y)$ denotes the l-th entropy number. For $k \geq \log_2(m)$, the entropy numbers satisfy:*

- *If $k \leq m$, then*

$$e_k\left(id : \ell_q^m(X) \to \ell_r^m(Y)\right) \simeq A(k, m).$$

- *If $k \geq m$, then there exist constants $C_1, C_2 > 0$ such that:*

$$D(C_1 k/m, k) \leq e_k\left(id : \ell_q^m(X) \to \ell_r^m(Y)\right) \leq D(C_2 k/m, k).$$

Let $q = p$, $r = 2$, $X = S_q^d$, or $Y = \ell_2^d$ for our problem, so that $\|id : X \to Y\| = 1$ and $e_l(id : X \to Y)$ is given by Eq. (11a), Eq. (11b) and Eq. (11c). Using the results of this lemma, we define the function $\phi(l)$ to analyze the behavior of entropy numbers for $\ell_p(S_q)$-balls. Specifically, we let:

$$\phi(l) := \left(\frac{l}{k}\right)^{\frac{1}{q}-\frac{1}{r}} e_l(id : X \to Y) = \begin{cases} \left(\frac{l}{k}\right)^{\frac{1}{p}-\frac{1}{2}} & 1 \leq l \leq d, \\ \left(\frac{l}{k}\right)^{\frac{1}{p}-\frac{1}{2}} \left(\frac{d}{l}\right)^{\frac{1}{q}-\frac{1}{2}} & d \leq l \leq d^2, \\ \left(\frac{l}{k}\right)^{\frac{1}{p}-\frac{1}{2}} 2^{-\frac{l}{d^2}} d^{\frac{1}{2}-\frac{1}{q}} & l \geq d^2. \end{cases}$$

The monotonicity behavior of $\phi(l)$ across different ranges of $l$ is summarized in Table 2.

*Table 2.* Monotonicity of $\phi(l)$ for $p \leq q$ and $p > q$ in different ranges of $l$ when $p \leq 1$

| Range of $l$ | Expression for $\phi(l)$ | Monotonicity (if $p \leq q$) | Monotonicity (if $p > q$) | Critical Point |
|---|---|---|---|---|
| $1 \leq l \leq d$ | $\left(\frac{l}{k}\right)^{\frac{1}{p}-\frac{1}{2}}$ | Increasing | | None |
| $d \leq l \leq d^2$ | $\left(\frac{l}{k}\right)^{\frac{1}{p}-\frac{1}{2}} \left(\frac{d}{l}\right)^{\frac{1}{q}-\frac{1}{2}}$ | Increasing | Decreasing | None |
| $l \geq d^2$ | $\left(\frac{l}{k}\right)^{\frac{1}{p}-\frac{1}{2}} 2^{-\frac{l}{d^2}} d^{\frac{1}{2}-\frac{1}{q}}$ | Increasing then Decreasing with maximum at $l^* = \frac{\left(\frac{1}{p}-\frac{1}{2}\right)d^2}{\ln 2}$ | | $l^* > d^2$ |
| | | Decreasing | | $l^* \leq d^2$ |

**Lemma C.14** (Entropy Number for $\ell_p(S_q) \hookrightarrow \ell_2(S_2)$). *For $k \geq \max\{\log m, d\}$, the entropy numbers $e_k$ for $\mathbb{B}_{\ell_p(S_q)}(R)$ satisfy:*

$$e_k \simeq_{p,q} \begin{cases} \left(\frac{d}{k}\right)^{\frac{1}{q}-\frac{1}{2}} & p \leq q, \ m \leq d^2 \\ \max\left\{\left(\frac{d}{k}\right)^{\frac{1}{q}-\frac{1}{2}}, \left(\frac{\log(em)}{k}\right)^{\frac{1}{p}-\frac{1}{2}}\right\} & p \leq q, \ m \geq d^2 \\ \left(\frac{\max\{d, \log(em)\}}{k}\right)^{\frac{1}{p}-\frac{1}{2}} & q \leq p. \end{cases} \tag{32}$$

*Proof of Lemma C.14.* Let us prove this lemma by carefully analyzing different cases based on the relationships between $p$, $q$, $m$, and $d$. Our analysis will heavily rely on the behavior of $\phi(l)$ as shown in Table 2 and the application of Lemma C.13.

**Case (a)**: For $p \leq q, m \leq d^2$, we consider the range $d \leq k \leq d^2$ and divide our analysis into two subcases based on the relationship between $k$ and $m$.

(i) First, consider $k \leq m$: According to Lemma C.13, we have:

$$e_k \simeq A(k, m) = \max\left\{\left(\frac{\log(em/k)}{k}\right)^{\frac{1}{p}-\frac{1}{2}}, D(1, k)\right\}.$$

To evaluate $D(1, k)$, we need to analyze $\phi(l)$ in different ranges:

- For $1 \leq l \leq d$:

$$\phi(l) = \left(\frac{l}{k}\right)^{\frac{1}{p}-\frac{1}{2}}.$$

This function is strictly increasing as $\frac{1}{p} - \frac{1}{2} > 0$ for $p \leq 1$. The maximum in this range occurs at $l = d$.

- For $d \leq l \leq d^2$:

$$\phi(l) = \left(\frac{l}{k}\right)^{\frac{1}{p}-\frac{1}{2}} \left(\frac{d}{l}\right)^{\frac{1}{q}-\frac{1}{2}}.$$

Since $p \leq q$, we have $\frac{1}{p} - \frac{1}{q} \geq 0$, making this function increasing. The maximum in this range occurs at $l = k$ since $k \leq d^2$.

Combining these results, we find:

$$D(1,k) = \max_{1 \leq l \leq k} \phi(l) = \left(\frac{d}{k}\right)^{\frac{1}{q}-\frac{1}{2}}.$$

Now, since $m \leq d^2$, we can show:

$$\left(\frac{\log(em/k)}{k}\right)^{\frac{1}{p}-\frac{1}{2}} \lesssim \left(\frac{d}{k}\right)^{\frac{1}{q}-\frac{1}{2}}.$$

Therefore:

$$e_k \simeq \left(\frac{d}{k}\right)^{\frac{1}{q}-\frac{1}{2}}.$$

(ii) Next, consider $m \leq k \leq d^2$: By Lemma C.13, when $k \geq m$, we have:

$$D(C_1 k/m, k) \leq e_k \leq D(C_2 k/m, k).$$

For each $C \in \{C_1, C_2\}$, we need to evaluate:

$$D(Ck/m, k) = \max_{Ck/m \leq l \leq k} \phi(l).$$

Let's analyze $\phi(l)$ in the relevant ranges:

- When $Ck/m \leq l \leq d$:

$$\phi(l) = \left(\frac{l}{k}\right)^{\frac{1}{p}-\frac{1}{2}}.$$

This is increasing, reaching its maximum at $l = d$ if $d$ is in this range.

- When $d \leq l \leq k \leq d^2$:

$$\phi(l) = \left(\frac{l}{k}\right)^{\frac{1}{p}-\frac{1}{2}} \left(\frac{d}{l}\right)^{\frac{1}{q}-\frac{1}{2}}.$$

Since $p \leq q$, this is increasing and reaches its maximum at $l = k$.

The monotonicity of $\phi(l)$ implies that both bounds achieve their maximum at $l = k$:

$$D(Ck/m, k) = \left(\frac{k}{k}\right)^{\frac{1}{p}-\frac{1}{2}} \left(\frac{d}{k}\right)^{\frac{1}{q}-\frac{1}{2}} = \left(\frac{d}{k}\right)^{\frac{1}{q}-\frac{1}{2}}.$$

Therefore:

$$D(C_1 k/m, k) = D(C_2 k/m, k) = \left(\frac{d}{k}\right)^{\frac{1}{q}-\frac{1}{2}}.$$

This gives us:

$$e_k \simeq \left(\frac{d}{k}\right)^{\frac{1}{q}-\frac{1}{2}}.$$

**Case (b)**: For $p \leq q, m \geq d^2$, when $d \leq k \leq m$, Lemma C.13 gives:

$$e_k \simeq \max\{ \left( \frac{\log(em/k)}{k} \right)^{\frac{1}{p} - \frac{1}{2}}, D(1, k)\}.$$

We analyze this in two subcases:

(i) For $d \leq k \leq d^2$: Similar to Case (a), analyzing $\phi(l)$ in three ranges:

- $1 \leq l \leq d$: $\phi(l) = \left( \frac{l}{k} \right)^{\frac{1}{p} - \frac{1}{2}}$ is increasing;
- $d \leq l \leq d^2$: $\phi(l) = \left( \frac{l}{k} \right)^{\frac{1}{p} - \frac{1}{2}} \left( \frac{d}{l} \right)^{\frac{1}{q} - \frac{1}{2}}$ is increasing since $p \leq q$;
- $l \geq d^2$: $\phi(l)$ is decreasing from $l = d^2$.

Therefore:

$$D(1, k) = \left( \frac{d}{k} \right)^{\frac{1}{q} - \frac{1}{2}}.$$

(ii) For $d^2 \leq k \leq m$: In this range:

$$D(1, k) = \max_{1 \leq l \leq k} \phi(l) = \left( \frac{d^2}{k} \right)^{\frac{1}{p} - \frac{1}{2}} d^{\frac{1}{2} - \frac{1}{q}}.$$

When $k \geq d^2$, we can show:

$$\left( \frac{d}{k} \right)^{\frac{1}{q} - \frac{1}{2}} \geq \left( \frac{d^2}{k} \right)^{\frac{1}{p} - \frac{1}{2}} d^{\frac{1}{2} - \frac{1}{q}} = \left( \frac{d}{k} \right)^{\frac{1}{p} - \frac{1}{2}} d^{\frac{1}{p} - \frac{1}{q}}.$$

Therefore, for Case (b):

$$e_k \simeq \max\{ \left( \frac{\log(em/k)}{k} \right)^{\frac{1}{p} - \frac{1}{2}}, \left( \frac{d}{k} \right)^{\frac{1}{q} - \frac{1}{2}}\}.$$

**Case (c)**: For $p > q$, we divide this case into two subcases based on the range of $k$.

(i) For $\max\{d, \log m\} \leq k \leq md$: When $p > q$, we know $\phi(l)$ is decreasing in $[d, d^2]$. Analyzing $\phi(l)$:

- For $1 \leq l \leq d$: Maximum occurs at $l = \max\{d, \log(em)\}$;
- For $d \leq l \leq d^2$: Monotonically decreasing;
- For $l \geq d^2$: Strictly decreasing;

Therefore:

$$D(1, k) = \left( \frac{\max\{d, \log(em)\}}{k} \right)^{\frac{1}{p} - \frac{1}{2}}.$$

(ii) For $md \leq k \leq md^2$: By similar analysis and considering $p > q$:

$$e_k \simeq m^{\frac{1}{q} - \frac{1}{p}} \left( \frac{d}{k} \right)^{\frac{1}{q} - \frac{1}{2}}.$$

Under the assumption $k \geq m \cdot \max\{d, \log(em)\}$:

$$\left( \frac{\max\{d, \log(em)\}}{k} \right)^{\frac{1}{p} - \frac{1}{2}} \geq m^{\frac{1}{q} - \frac{1}{p}} \left( \frac{d}{k} \right)^{\frac{1}{q} - \frac{1}{2}} \iff \left( \frac{\max\{d, \log(em)\}}{d} \right)^{\frac{1}{p} - \frac{1}{2}} \geq m^{\frac{1}{q} - \frac{1}{p}}.$$

This inequality holds under our assumptions.

Combining all three cases yields the result in Eq. (32). □

C.4.3. PROOF OF UPPER BOUNDS FOR $\ell_p(S_q)$

Recall that we define

$$\widetilde{\mathbf{S}}(\mathbf{T}_{p,q}(R), \rho) = \{\underline{\mathbf{L}} \in \mathbb{R}^{d \times m} : \|\underline{\mathbf{L}}\|_F \leq \rho\} \cap \mathbf{T}_{p,q}(R).$$

*Proof.* We prove the theorem by separating into three distinct cases based on the relationships among $p$, $q$, and $m$. In each case, we construct appropriate constants and verify the necessary conditions.

**Case 1: $p \leq q$ and $m \geq d^2$.** We begin by adding a radius factor $R^{\frac{1}{p}}$ to the entropy number bounds in Eq. (32), yielding

$$\epsilon \leq R^{\frac{1}{p}} \left(\frac{d}{k}\right)^{\frac{1}{q}-\frac{1}{2}} + R^{\frac{1}{p}} \left(\frac{\log(em)}{k}\right)^{\frac{1}{p}-\frac{1}{2}}.$$

Solving for $k$ provides an upper bound on the covering number, giving

$$\log N\left(\epsilon; \widetilde{\mathbf{S}}(\mathbf{T}_{p,q}(R), r)\right) \leq d \left(\epsilon^{-1} R^{\frac{1}{p}}\right)^{\frac{2q}{2-q}} + \log(em) \left(\epsilon^{-1} R^{\frac{1}{p}}\right)^{\frac{2p}{2-p}}. \tag{33}$$

To apply Lemma 3.2 from (van de Geer, 2009), we need to construct constants $(\delta, \rho)$ satisfying two key conditions:

$$\sqrt{n}\, \delta \geq C_1\, \rho \qquad\qquad \text{(Condition 1)} \tag{34}$$
$$C_2\, \sqrt{n}\, \delta \geq J(\rho, \delta) \qquad\qquad \text{(Condition 2)} \tag{35}$$

where

$$J(\rho, \delta) = \int_{\frac{\delta}{16}}^{\rho} \sqrt{\log N\left(t; \widetilde{\mathbf{S}}(\mathbf{T}_{p,q}(R), \rho)\right)}\, \mathrm{d}t \leq \int_0^{\rho} \sqrt{d\left(t^{-1}R^{\frac{1}{p}}\right)^{\frac{2q}{2-q}} + \log(em)\left(t^{-1}R^{\frac{1}{p}}\right)^{\frac{2p}{2-p}}}\, \mathrm{d}t.$$

A direct calculation yields:

$$J(\rho, \delta) \leq \sqrt{d}\, R^{\frac{q}{p(2-q)}}\, \rho^{1-\frac{q}{2-q}} + \sqrt{\log(em)}\, R^{\frac{1}{2-p}}\, \rho^{1-\frac{p}{2-p}}.$$

**Choice of constants $\rho, \delta$.** Let

$$\rho = \Omega\left(R^{\frac{q}{2p}}\left(\frac{n}{d}\right)^{\frac{q-2}{4}} + R^{\frac{1}{2}}\left(\frac{n}{\log m}\right)^{\frac{p-2}{4}}\right) \wedge R^{\frac{1}{p}}, \tag{36}$$

$$\delta = C\, \rho\left(R^{\frac{q}{2p}}\left(\frac{n}{d}\right)^{\frac{q-2}{4}} + R^{\frac{1}{2}}\left(\frac{n}{\log m}\right)^{\frac{p-2}{4}}\right). \tag{37}$$

*Verifying Condition Eq. (34).*

$$\frac{\sqrt{n}\, \delta}{\rho} \geq C\left[R^{\frac{q}{p}}\, n^{\frac{q}{2}}\left(\frac{1}{d}\right)^{\frac{q-2}{2}} + R\, n^{\frac{p}{2}}\left(\frac{1}{\log m}\right)^{\frac{p-2}{2}}\right] \geq C_1,$$

where the second inequality follows from Eq. (16).

*Verifying Condition Eq. (35).* Given that $\rho$ is chosen as in Eq. (36), an analogous ratio bound shows

$$\frac{J(\rho, \delta)}{\sqrt{n}\, \delta} = \frac{\sqrt{d}\, R^{\frac{q}{p(2-q)}}\, \rho^{-\frac{q}{2-q}} + \sqrt{\log(m)}\, R^{\frac{1}{2-p}}\, \rho^{-\frac{p}{2-p}}}{C\sqrt{n}\left(R^{\frac{q}{2p}}\left(\frac{n}{d}\right)^{\frac{q-2}{4}} + R^{\frac{1}{2}}\left(\frac{n}{\log m}\right)^{\frac{p-2}{4}}\right)} = \frac{\sqrt{2}}{C},$$

implying $C_2\sqrt{n}\, \delta \geq J(\rho, \delta)$. Additionally, we must check that $(\frac{\delta}{16}, \rho)$ is a valid interval for covering; an argument similar to Eq. (33) and $\rho < R^{\frac{1}{p}}$ ensures

$$\log N\left(\delta; \widetilde{\mathbf{S}}(\mathbf{T}_{p,q}(R), \rho)\right) \geq \max\{d, \log m\},$$

so the condition for Lemma C.11 is met.

Hence, applying Lemma C.11 gives that, with probability at least $1 - C_5 \exp\left\{ -C_6\, n \left[ R^{\frac{q}{p}} \left(\frac{n}{d}\right)^{\frac{q-2}{2}} + R\left(\frac{n}{\log m}\right)^{\frac{p-2}{2}} \right] \right\}$, we have

$$\sup_{\underline{\mathbf{L}} \in \widetilde{\mathbf{S}}(\mathbf{T}_{p,q}(R),\rho)} \left| \langle \bar{\underline{\mathbf{E}}}, \underline{\mathbf{L}} \rangle \right| \leq \left[ R^{\frac{q}{2p}} \left(\frac{n}{d\sigma^2}\right)^{\frac{q-2}{4}} + R^{\frac{1}{2}} \left(\frac{n}{\sigma^2 \log m}\right)^{\frac{p-2}{4}} \right] \rho. \tag{38}$$

(*The other two cases for $p \leq q, m \leq d^2$ and $p \geq q$ follow the same procedure, so we omit full detail here.*)

Combining the arguments for all three regimes and applying Lemma C.11 completes the proof of the upper bounds for $\ell_p(S_q)$. □

### C.5. Preliminary Numerical Illustration of Theorem 4.2

To complement the theoretical analysis, we conduct preliminary simulations to illustrate how the recovery error obtained by our algorithm behaves under different structural parameters in the hard dual sparsity setting. We vary the spectral rank $r$, the frequency-domain sparsity $s$, the sample size $n$, and the noise-to-signal ratio (NSR). These experiments are not intended to establish minimax optimality but rather to provide an empirical illustration of the trends predicted by theory.

In particular, Theorem 4.2 provides an information-theoretic minimax lower bound on the estimation error under dual spectral sparsity, characterizing the fundamental statistical difficulty of the problem. Such results apply to any estimator, efficient or not. We do not claim that our algorithm achieves this minimax rate. Attaining such bounds algorithmically is highly challenging because the $\ell_p$-Schatten-$q$ regularizer is jointly nonconvex in both parameters and introduces cross-frequency coupling, preventing separable optimization. To the best of our knowledge, no known polynomial-time algorithm achieves the exact minimax rate in this tensor setting, reflecting a statistical–computational gap. Our algorithm therefore serves as a practical and interpretable approximation rather than an exact minimax solution.

**Experimental Setting.** We aim to construct a third-order tensor $\underline{\mathbf{L}} \in \mathbb{R}^{d \times d \times m}$ that exhibits low-rank structure in only a few frequency slices while being zero in the others. To this end, we first define an auxiliary tensor $\tilde{\underline{\mathbf{L}}} \in \mathbb{R}^{d \times d \times m}$ in the frequency domain:

(1) select $s$ active slices; for each slice $i$, generate a rank-$r$ matrix $\tilde{\underline{\mathbf{L}}}^{(i)} = \mathbf{A}_i \mathbf{B}_i$, where $\mathbf{A}_i \in \mathbb{R}^{d \times r}$ and $\mathbf{B}_i \in \mathbb{R}^{r \times d}$ are drawn from Gaussian ensembles;

(2) set the remaining $m - s$ slices to zero, thereby inducing spectral sparsity;

(3) apply an inverse DCT along the third mode of $\tilde{\underline{\mathbf{L}}}$ to obtain the spatial-domain tensor $\underline{\mathbf{L}}$.

We then normalize $\|\underline{\mathbf{L}}\|_F = 1$ and add isotropic Gaussian noise $\underline{\mathbf{E}} \sim \mathcal{N}(0, \sigma^2 \mathbf{I})$ with variance

$$\sigma^2 = \frac{\text{NSR}^2}{d^2 m}.$$

The observed tensor is $\underline{\mathbf{Y}} = \underline{\mathbf{L}} + \underline{\mathbf{E}}$. For $n$ i.i.d. observations, the sample mean has effective noise variance $\sigma^2/n$, consistent with our paper's formulation.

We report the squared error

$$\text{SE} = \|\hat{\underline{\mathbf{L}}} - \underline{\mathbf{L}}\|_F^2,$$

where $\hat{\underline{\mathbf{L}}}$ is the recovered tensor. Since $\|\underline{\mathbf{L}}\|_F = 1$, the expected SE scales with the total noise energy $\sigma^2 d^2 m$. Each SE value is averaged over 10 independent trials with independently generated $\underline{\mathbf{L}}$ and noise.

**Results.** Table 3 summarizes the results when varying NSR, rank $r$, and sparsity $s$.

The observed trends in Table 3 are consistent with theoretical predictions:

(1) recovery error grows with NSR;

*Table 3.* Preliminary numerical illustration of Theorem 4.2 (Part I). Squared error (SE) under different settings.

| Varying Factor | $r$ | $s$ | NSR | SE |
|---|---|---|---|---|
| NSR ($r = 1, s = 2$) | 1 | 2 | 0.05 | 2.09e-05 |
| NSR ($r = 1, s = 2$) | 1 | 2 | 0.10 | 3.44e-05 |
| NSR ($r = 1, s = 2$) | 1 | 2 | 0.15 | 4.92e-05 |
| NSR ($r = 1, s = 2$) | 1 | 2 | 0.20 | 7.47e-05 |
| Rank $r$ ($s = 2$, NSR=0.1) | 1 | 2 | 0.10 | 3.44e-05 |
| Rank $r$ ($s = 2$, NSR=0.1) | 2 | 2 | 0.10 | 8.05e-05 |
| Rank $r$ ($s = 2$, NSR=0.1) | 3 | 2 | 0.10 | 1.47e-04 |
| Rank $r$ ($s = 2$, NSR=0.1) | 4 | 2 | 0.10 | 2.29e-04 |
| Sparsity $s$ ($r = 2$, NSR=0.1) | 2 | 2 | 0.10 | 8.05e-05 |
| Sparsity $s$ ($r = 2$, NSR=0.1) | 2 | 4 | 0.10 | 2.16e-04 |
| Sparsity $s$ ($r = 2$, NSR=0.1) | 2 | 6 | 0.10 | 3.58e-04 |
| Sparsity $s$ ($r = 2$, NSR=0.1) | 2 | 8 | 0.10 | 5.45e-04 |

(2) higher rank $r$ increases the error, as more components must be recovered;

(3) larger sparsity $s$ also worsens recovery, since more active frequencies increase complexity.

These results qualitatively match Theorem 4.2 (Part I), providing preliminary empirical support. Moreover, they also serve as an empirical reflection of the statistical–computational gap: while the minimax bound characterizes the best possible rate in principle, our practical algorithm demonstrates consistent yet sub-optimal behavior under increasing complexity. A more comprehensive ablation study is left for future work.

# D. Empirical Evidence for Dual Spectral Sparsity

Dual spectral sparsity, i.e., the simultaneous presence of inter-frequency sparsity and intra-frequency low-rankness in the transform domain, is a central modeling assumption of this work. While we have included examples from medical and hyperspectral data (e.g., *MRI, CT, Salinas A*, and *Indian Pines*, see Figures 1, 4 and 5), we provide here a systematic and statistically grounded validation that goes well beyond illustrative evidence. Our goal is to determine **whether dual spectral sparsity is an inherent structural pattern rather than an artifact of specific examples**. To assess its generality, we quantitatively compare real tensors with principled baselines such as *Gaussian random tensors*, evaluating whether inter-frequency sparsity and intra-frequency low-rankness exhibit statistically significant differences between real tensors and their random counterparts.

Specifically, given a tensor $\mathbf{T} \in \mathbb{R}^{d_1 \times d_2 \times m}$ and a linear transform $M$ applied along the third mode, we seek to evaluate two questions:

(i) **Inter-frequency sparsity:** *Is the energy across frequency slices highly concentrated?*

(ii) **Intra-frequency low-rankness:** *Do the singular values within each slice decay rapidly, indicating low effective rank?*

To answer these questions in a statistically rigorous manner, we first construct quantitative metrics for the two phenomena and then perform hypothesis tests against matched Gaussian random tensors.

## D.1. Quantitative Metrics for Dual Spectral Sparsity

We introduce the following quantitative metrics for the two phenomena:

(i) *Quantifying intra-frequency low-rankness.* For each transformed slice $M(\mathbf{T})_{:,:,i}$, we adopt the *stable rank* (Ipsen & Saibaba, 2025):

$$\text{s-rank}(\mathbf{A}) := \frac{\|\mathbf{A}\|_\text{F}^2}{\|\mathbf{A}\|_\text{spec}^2} = \frac{\sum_j \sigma_j(\mathbf{A})^2}{\max_j \sigma_j(\mathbf{A})^2},$$

which captures the "effective dimension" of a spectrum: a rank-one matrix has stable rank 1, whereas a full-rank matrix with equal singular values achieves the maximum possible stable rank.

We then define the *average frequency-wise stable rank:*

$$\text{F-rank}_\text{avg}(\mathbf{T}) := \frac{1}{m} \sum_{i=1}^m \text{s-rank}(M(\mathbf{T})_{:,:,i}), \qquad \mathfrak{r}(\mathbf{T}) := \frac{\text{F-rank}_\text{avg}(\mathbf{T})}{\min(d_1, d_2)}.$$

Here $\mathfrak{r}(\mathbf{T}) \in (0, 1]$ serves as a normalized indicator of slice-level low-rankness: values closer to 0 indicate strong intra-frequency low-rankness, and values close to 1 indicate nearly full-rank behavior.

(ii) *Quantifying inter-frequency sparsity.* Let $E_i := \|M(\mathbf{T})_{:,:,i}\|_\text{F}^2$ denote the energy of the $i$-th frequency slice. We define

$$\text{F-sparsity}(\mathbf{T}) := \frac{\sum_{i=1}^m E_i}{\max_i E_i}, \qquad \mathfrak{s}(\mathbf{T}) := \frac{\text{F-sparsity}(\mathbf{T})}{m}.$$

The ratio $\mathfrak{s}(\mathbf{T})$ equals 1 if all frequencies carry similar energy and becomes smaller as energy becomes concentrated in a few slices. Thus $\mathfrak{s}(\mathbf{T})$ acts as a normalized measure of inter-frequency sparsity.

## D.2. Hypothesis Testing on Typical Tensor Datasets

To quantitatively assess the universality of dual spectral sparsity, we apply the proposed metrics to two typical tensor datasets: 22 grayscale YUV video tensors[5] and 31 multispectral datasets[6]. Specifically, for each tensor $\mathbf{T}$, we compute:

- $\mathfrak{s}(\mathbf{T})$: the *frequency sparsity ratio*, measuring how concentrated the transform-domain energy is across frequency slices;

---

[5] https://media.xiph.org/video/derf/
[6] https://cave.cs.columbia.edu/repository/Multispectral

*Table 4.* Dual spectral sparsity statistics across 22 YUV video tensors. $\mathfrak{s}(\mathbf{T})$ and $\mathfrak{r}(\mathbf{T})$ quantify inter-frequency sparsity and intra-frequency low-rankness of real tensors; the last two columns report the corresponding values for Gaussian random tensors of the same size.

| Dataset | $\mathfrak{s}(\mathbf{T})$ | | $\mathfrak{r}(\mathbf{T})$ | | $\mathfrak{s}(\mathrm{rand})$ | | $\mathfrak{r}(\mathrm{rand})$ | |
|---|---|---|---|---|---|---|---|---|
| akiyo | 3.33 | $\times 10^{-3}$ | 2.99 | $\times 10^{-2}$ | 9.49 | $\times 10^{-1}$ | 2.84 | $\times 10^{-1}$ |
| bridge-far | 4.76 | $\times 10^{-4}$ | 1.10 | $\times 10^{-1}$ | 9.45 | $\times 10^{-1}$ | 2.84 | $\times 10^{-1}$ |
| bus | 6.73 | $\times 10^{-3}$ | 1.74 | $\times 10^{-2}$ | 9.78 | $\times 10^{-1}$ | 2.80 | $\times 10^{-1}$ |
| carphone | 2.62 | $\times 10^{-3}$ | 4.79 | $\times 10^{-2}$ | 9.54 | $\times 10^{-1}$ | 2.84 | $\times 10^{-1}$ |
| claire | 2.02 | $\times 10^{-3}$ | 4.10 | $\times 10^{-2}$ | 9.54 | $\times 10^{-1}$ | 2.84 | $\times 10^{-1}$ |
| coastguard | 3.33 | $\times 10^{-3}$ | 5.39 | $\times 10^{-2}$ | 9.42 | $\times 10^{-1}$ | 2.83 | $\times 10^{-1}$ |
| container | 3.33 | $\times 10^{-3}$ | 3.82 | $\times 10^{-2}$ | 9.51 | $\times 10^{-1}$ | 2.84 | $\times 10^{-1}$ |
| flower | 4.00 | $\times 10^{-3}$ | 6.39 | $\times 10^{-2}$ | 9.73 | $\times 10^{-1}$ | 2.81 | $\times 10^{-1}$ |
| foreman | 3.34 | $\times 10^{-3}$ | 6.94 | $\times 10^{-2}$ | 9.43 | $\times 10^{-1}$ | 2.84 | $\times 10^{-1}$ |
| grandma | 1.15 | $\times 10^{-3}$ | 1.05 | $\times 10^{-1}$ | 9.50 | $\times 10^{-1}$ | 2.84 | $\times 10^{-1}$ |
| hall | 3.33 | $\times 10^{-3}$ | 5.89 | $\times 10^{-2}$ | 9.51 | $\times 10^{-1}$ | 2.84 | $\times 10^{-1}$ |
| highway | 5.00 | $\times 10^{-4}$ | 7.20 | $\times 10^{-2}$ | 9.43 | $\times 10^{-1}$ | 2.84 | $\times 10^{-1}$ |
| miss-america | 6.67 | $\times 10^{-3}$ | 5.42 | $\times 10^{-2}$ | 9.48 | $\times 10^{-1}$ | 2.84 | $\times 10^{-1}$ |
| mobile | 3.34 | $\times 10^{-3}$ | 9.91 | $\times 10^{-2}$ | 9.44 | $\times 10^{-1}$ | 2.84 | $\times 10^{-1}$ |
| mother-daughter | 3.33 | $\times 10^{-3}$ | 5.86 | $\times 10^{-2}$ | 9.40 | $\times 10^{-1}$ | 2.85 | $\times 10^{-1}$ |
| news | 3.33 | $\times 10^{-3}$ | 4.03 | $\times 10^{-2}$ | 9.31 | $\times 10^{-1}$ | 2.85 | $\times 10^{-1}$ |
| salesman | 2.23 | $\times 10^{-3}$ | 6.69 | $\times 10^{-2}$ | 9.45 | $\times 10^{-1}$ | 2.84 | $\times 10^{-1}$ |
| silent | 3.33 | $\times 10^{-3}$ | 4.27 | $\times 10^{-2}$ | 9.39 | $\times 10^{-1}$ | 2.84 | $\times 10^{-1}$ |
| stefan | 1.11 | $\times 10^{-2}$ | 5.59 | $\times 10^{-2}$ | 9.82 | $\times 10^{-1}$ | 2.80 | $\times 10^{-1}$ |
| suzie | 6.67 | $\times 10^{-3}$ | 4.94 | $\times 10^{-2}$ | 9.63 | $\times 10^{-1}$ | 2.85 | $\times 10^{-1}$ |
| tempete | 3.87 | $\times 10^{-3}$ | 5.96 | $\times 10^{-2}$ | 9.76 | $\times 10^{-1}$ | 2.81 | $\times 10^{-1}$ |
| waterfall | 3.85 | $\times 10^{-3}$ | 5.58 | $\times 10^{-2}$ | 9.74 | $\times 10^{-1}$ | 2.81 | $\times 10^{-1}$ |

- $\mathfrak{r}(\mathbf{T})$: the *stable-rank ratio*, capturing the effective intra-slice low-rankness of the transformed slices;

- $\mathfrak{s}(\mathrm{rand})$ and $\mathfrak{r}(\mathrm{rand})$: the same quantities computed for Gaussian random tensors with identical size and mean energy.

We expect real tensors to exhibit a clear gap from random baselines, reflecting strong inter-frequency sparsity and intra-frequency low-rankness.

As reported in Table 4, both quantities $\mathfrak{s}(\mathbf{T})$ and $\mathfrak{r}(\mathbf{T})$ for video tensors are consistently several orders of magnitude smaller than those of matched Gaussian random tensors, indicating that transform-domain energy concentrates among a few frequency slices and that the active slices are intrinsically low-rank. These findings suggest that *dual spectral sparsity is not an incidental artifact of the transform but an inherent property shared across video data*.

Similar results are also observed in Table 5 for 31 multispectral images, where $\mathfrak{s}(\mathbf{T})$ remains on the order of $10^{-2}$–$10^{-3}$, while Gaussian random tensors of identical size consistently yield values around $0.9$. Similarly, the stable-rank ratios $\mathfrak{r}(\mathbf{T})$ stay between $10^{-3}$ and $10^{-2}$, in sharp contrast to the $\approx 0.25$ ratios observed for the random baselines. This again confirms strong inter-frequency sparsity and pronounced intra-frequency low-rankness, reinforcing the universality of dual spectral sparsity across modalities beyond videos and medical images.

This conclusion is further enhanced by ***formal hypothesis tests*** conducted over all 22 video tensors and 31 multispectral tensors.

Table 6 summarizes the statistical testing between video tensors and matched Gaussian random tensors, which gives rise to the following findings:

- **Statistical test on inter-frequency sparsity $\mathfrak{s}(\mathbf{T})$**: the test video data yield extremely small values and show a complete separation in distribution from random tensors, with both Wilcoxon and KS tests reporting $p < 10^{-6}$ and an exceptionally large effect size (Cohen's $d = -91.84$). This confirms that transform-domain energy is highly concentrated among only a few frequency slices in the test video data.

*Table 5.* Dual spectral sparsity statistics across 31 multispectral images. $\mathfrak{s}(\mathbf{T})$ and $\mathfrak{r}(\mathbf{T})$ denote frequency sparsity and stable-rank ratios, compared against Gaussian random tensors with matched size and energy.

| Dataset | $\mathfrak{s}(\mathbf{T})$ | $\mathfrak{r}(\mathbf{T})$ | $\mathfrak{s}(\text{rand})$ | $\mathfrak{r}(\text{rand})$ |
|---|---|---|---|---|
| balloons | $3.24 \times 10^{-2}$ | $3.45 \times 10^{-3}$ | $9.90 \times 10^{-1}$ | $2.53 \times 10^{-1}$ |
| beads | $3.43 \times 10^{-2}$ | $1.11 \times 10^{-2}$ | $9.81 \times 10^{-1}$ | $2.54 \times 10^{-1}$ |
| cd | $3.30 \times 10^{-2}$ | $7.38 \times 10^{-3}$ | $9.86 \times 10^{-1}$ | $2.53 \times 10^{-1}$ |
| chart_toy | $3.23 \times 10^{-2}$ | $3.19 \times 10^{-3}$ | $9.87 \times 10^{-1}$ | $2.54 \times 10^{-1}$ |
| clay | $4.08 \times 10^{-2}$ | $3.15 \times 10^{-3}$ | $9.89 \times 10^{-1}$ | $2.54 \times 10^{-1}$ |
| cloth | $3.39 \times 10^{-2}$ | $5.20 \times 10^{-3}$ | $9.86 \times 10^{-1}$ | $2.53 \times 10^{-1}$ |
| egyptian_statue | $3.23 \times 10^{-2}$ | $3.64 \times 10^{-3}$ | $9.89 \times 10^{-1}$ | $2.53 \times 10^{-1}$ |
| face | $3.27 \times 10^{-2}$ | $5.25 \times 10^{-3}$ | $9.86 \times 10^{-1}$ | $2.54 \times 10^{-1}$ |
| fake_beers | $3.23 \times 10^{-2}$ | $3.17 \times 10^{-3}$ | $9.92 \times 10^{-1}$ | $2.53 \times 10^{-1}$ |
| fake_food | $3.49 \times 10^{-2}$ | $4.43 \times 10^{-3}$ | $9.87 \times 10^{-1}$ | $2.54 \times 10^{-1}$ |
| fake_lemonslices | $3.24 \times 10^{-2}$ | $6.22 \times 10^{-3}$ | $9.92 \times 10^{-1}$ | $2.52 \times 10^{-1}$ |
| fake_lemons | $3.28 \times 10^{-2}$ | $3.69 \times 10^{-3}$ | $9.93 \times 10^{-1}$ | $2.53 \times 10^{-1}$ |
| fake_peppers | $3.47 \times 10^{-2}$ | $3.43 \times 10^{-3}$ | $9.92 \times 10^{-1}$ | $2.54 \times 10^{-1}$ |
| fake_strawberries | $3.23 \times 10^{-2}$ | $3.57 \times 10^{-3}$ | $9.90 \times 10^{-1}$ | $2.53 \times 10^{-1}$ |
| fake_sushi | $3.27 \times 10^{-2}$ | $3.76 \times 10^{-3}$ | $9.87 \times 10^{-1}$ | $2.53 \times 10^{-1}$ |
| fake_tomatoes | $3.54 \times 10^{-2}$ | $5.49 \times 10^{-3}$ | $9.90 \times 10^{-1}$ | $2.53 \times 10^{-1}$ |
| feathers | $3.25 \times 10^{-2}$ | $3.20 \times 10^{-3}$ | $9.87 \times 10^{-1}$ | $2.53 \times 10^{-1}$ |
| flowers | $3.38 \times 10^{-2}$ | $3.83 \times 10^{-3}$ | $9.88 \times 10^{-1}$ | $2.55 \times 10^{-1}$ |
| glass_tiles | $3.40 \times 10^{-2}$ | $3.84 \times 10^{-3}$ | $9.94 \times 10^{-1}$ | $2.53 \times 10^{-1}$ |
| hairs | $3.24 \times 10^{-2}$ | $3.52 \times 10^{-3}$ | $9.92 \times 10^{-1}$ | $2.54 \times 10^{-1}$ |
| jelly_beans | $3.25 \times 10^{-2}$ | $8.68 \times 10^{-3}$ | $9.91 \times 10^{-1}$ | $2.53 \times 10^{-1}$ |
| oil_painting | $3.24 \times 10^{-2}$ | $4.27 \times 10^{-3}$ | $9.90 \times 10^{-1}$ | $2.53 \times 10^{-1}$ |
| paints | $3.23 \times 10^{-2}$ | $3.04 \times 10^{-3}$ | $9.88 \times 10^{-1}$ | $2.52 \times 10^{-1}$ |
| photo_face | $3.23 \times 10^{-2}$ | $6.02 \times 10^{-3}$ | $9.88 \times 10^{-1}$ | $2.54 \times 10^{-1}$ |
| pompoms | $3.44 \times 10^{-2}$ | $5.32 \times 10^{-3}$ | $9.87 \times 10^{-1}$ | $2.54 \times 10^{-1}$ |
| real_apples | $3.25 \times 10^{-2}$ | $3.73 \times 10^{-3}$ | $9.92 \times 10^{-1}$ | $2.53 \times 10^{-1}$ |
| real_peppers | $3.32 \times 10^{-2}$ | $3.58 \times 10^{-3}$ | $9.91 \times 10^{-1}$ | $2.53 \times 10^{-1}$ |
| sponges | $3.45 \times 10^{-2}$ | $2.51 \times 10^{-3}$ | $9.88 \times 10^{-1}$ | $2.55 \times 10^{-1}$ |
| stuffed_toys | $3.28 \times 10^{-2}$ | $3.42 \times 10^{-3}$ | $9.89 \times 10^{-1}$ | $2.54 \times 10^{-1}$ |
| superballs | $3.50 \times 10^{-2}$ | $4.35 \times 10^{-3}$ | $9.85 \times 10^{-1}$ | $2.53 \times 10^{-1}$ |
| thread_spools | $3.28 \times 10^{-2}$ | $3.97 \times 10^{-3}$ | $9.89 \times 10^{-1}$ | $2.53 \times 10^{-1}$ |

- **Statistical test on intra-frequency low-rankness** $\mathfrak{r}(\mathbf{T})$: the stable-rank ratios are again substantially smaller for real tensors, with both tests reporting $p < 10^{-6}$ and a large effect size (Cohen's $d = -13.88$). This demonstrates that singular values within each active frequency slice decay much more rapidly than in random tensors, revealing strong slice-wise low-rank structure.

Table 7 reports the hypothesis testing results over all 31 multispectral tensors. For both inter-frequency sparsity $\mathfrak{s}(\mathbf{T})$ and intra-frequency stable-rank ratios $\mathfrak{r}(\mathbf{T})$, the Wilcoxon and Kolmogorov–Smirnov tests consistently yield $p < 10^{-6}$, indicating complete distributional separation from their Gaussian random counterparts. The corresponding effect sizes are extremely large (Cohen's $d = -4.26 \times 10^2$ for $\mathfrak{s}$ and $-1.83 \times 10^2$ for $\mathfrak{r}$), reflecting differences of several orders of magnitude. These results confirm that the multispectral datasets exhibit both pronounced frequency sparsity and strong slice-wise low-rankness, further reinforcing the universality of dual spectral sparsity beyond natural videos and medical images.

Together, these results indicate that dual spectral sparsity, understood as energy concentration across slices together with pronounced low-rankness within slices, is not a coincidental pattern but a statistically stable property observed in many forms of real tensor data.

*Table 6.* Hypothesis testing results for dual spectral sparsity across 22 YUV video tensors. $\mathfrak{s}(\mathbf{T})$ measures inter-frequency sparsity, and $\mathfrak{r}(\mathbf{T})$ measures intra-frequency low-rankness. Both metrics for real tensors are compared against size-matched Gaussian random tensors.

| Metric | Wilcoxon $p$-value | KS $p$-value | Cohen's $d$ |
|---|---|---|---|
| Inter-frequency sparsity $\mathfrak{s}(\mathbf{T})$ | $<10^{-6}$ | $<10^{-6}$ | $-91.84$ |
| Intra-frequency low-rankness $\mathfrak{r}(\mathbf{T})$ | $<10^{-6}$ | $<10^{-6}$ | $-13.88$ |

*Table 7.* Hypothesis testing results for dual spectral sparsity across 31 multispectral images. The metrics $\mathfrak{s}(\mathbf{T})$ and $\mathfrak{r}(\mathbf{T})$ denote inter-frequency sparsity and intra-frequency stable-rank ratios, compared against Gaussian random tensors with matched size and energy.

| Metric | Wilcoxon $p$-value | KS $p$-value | Cohen's $d$ |
|---|---|---|---|
| Inter-frequency sparsity $\mathfrak{s}(\mathbf{T})$ | $<10^{-6}$ | $<10^{-6}$ | $-4.26 \times 10^2$ |
| Intra-frequency low-rankness $\mathfrak{r}(\mathbf{T})$ | $<10^{-6}$ | $<10^{-6}$ | $-1.83 \times 10^2$ |

**Consistency of Dual Spectral Sparsity Across Modalities.** The same dual sparsity pattern appears in *natural videos, medical imagery (MRI and CT), hyperspectral cubes such as Salinas A and Indian Pines, and multispectral data*. Across all these modalities, transform-domain energy concentrates in a small subset of frequency slices, the active slices show rapid singular value decay that reflects strong low-rankness, and the statistics of real data differ clearly and consistently from random-tensor baselines.

> *These observations indicate that dual spectral sparsity is a **statistically stable structural feature** present in many real tensor modalities rather than a dataset-specific artifact.*

# E. Experimental Details and Additive Results

This appendix compiles the full experimental and algorithmic details supporting our tensor $\ell_p(S_q)$ framework. We first outline the noisy tensor completion setting and then describe the ADMM-based solver used for optimization, including its update rules, computational cost, and observed convergence behavior. The remaining sections provide extended empirical studies, including ablations, robustness analyses, experiments on additional 3D and 4D tensors, Poisson noise settings, clustering benchmarks, and a comprehensive parameter sensitivity analysis.

## E.1. Experimental Setup

**Noisy Tensor Completion Task Formulation.** The noisy tensor completion problem aims to recover a structured tensor $\underline{\mathbf{L}}^\star$ from a set of noisy and incomplete observations. This problem is particularly relevant in applications such as hyperspectral image restoration, video inpainting, and remote sensing data reconstruction, where missing and corrupted data are common due to sensor limitations or transmission errors.

We consider a third-order tensor $\underline{\mathbf{L}}^\star \in \mathbb{R}^{d_1 \times d_2 \times d_3}$ that represents a clean, fully observed data source. However, due to data corruption and missing values, we only have access to a partially observed noisy tensor $\underline{\mathbf{Y}}$, which is generated as:

$$\underline{\mathbf{Y}} = \underline{\mathbf{B}} \odot (\underline{\mathbf{L}}^\star + \underline{\mathbf{E}}),$$

where:

- $\underline{\mathbf{B}}$ *(Binary Mask)*: A binary tensor of the same size as $\underline{\mathbf{L}}^\star$, where each entry $\underline{\mathbf{B}}_{i,j,k} \in \{0,1\}$ indicates whether the corresponding entry in $\underline{\mathbf{L}}^\star$ is observed ($\underline{\mathbf{B}}_{i,j,k} = 1$) or missing ($\underline{\mathbf{B}}_{i,j,k} = 0$).

- $\odot$ *(Hadamard Product)*: The element-wise product operator ensures that only the observed entries are retained, while unobserved entries are set to zero.

- $\underline{\mathbf{E}}$ *(Noise Tensor)*: Represents random additive noise introduced in the observed entries. Each entry of $\underline{\mathbf{E}}$ is sampled independently from a Gaussian distribution:

$$\underline{\mathbf{E}}_{i,j,k} \sim \mathcal{N}(0, \sigma^2),$$

where the noise level $\sigma$ is set as:

$$\sigma = c\sigma_0, \quad \text{with} \quad c = 0.05, \quad \sigma_0 = \frac{\|\underline{\mathbf{L}}^\star\|_{\mathrm{F}}}{\sqrt{d_1 d_2 d_3}}.$$

Here, $\sigma_0$ represents a normalized noise scale based on the Frobenius norm of the clean tensor.

**Sampling Strategy and Experimental Settings.** We apply a uniform random sampling strategy, where each entry of $\underline{\mathbf{L}}^\star$ is independently observed with probability $p$, meaning that a fraction $1 - p$ of the entries is missing. We consider three different missing ratios: $p \in \{0.05, 0.1, 0.15\}$, which correspond to scenarios where 95%, 90%, and 85% of the entries are missing, respectively. Each experiment is conducted over 10 independent trials to ensure statistical reliability, and the averaged Peak Signal-to-Noise Ratio (PSNR) and Structural Similarity Index (SSIM) are reported to evaluate reconstruction performance.

**Evaluation Metrics.** To assess the quality of tensor reconstruction, we use the following two widely adopted metrics:

- *Peak Signal-to-Noise Ratio (PSNR)*:

$$\mathrm{PSNR} = 10 \log_{10} \left( \frac{\max(\underline{\mathbf{L}}^\star)^2}{\frac{1}{d_1 d_2 d_3} \|\hat{\underline{\mathbf{L}}} - \underline{\mathbf{L}}^\star\|_{\mathrm{F}}^2} \right).$$

A higher PSNR value indicates better reconstruction quality.

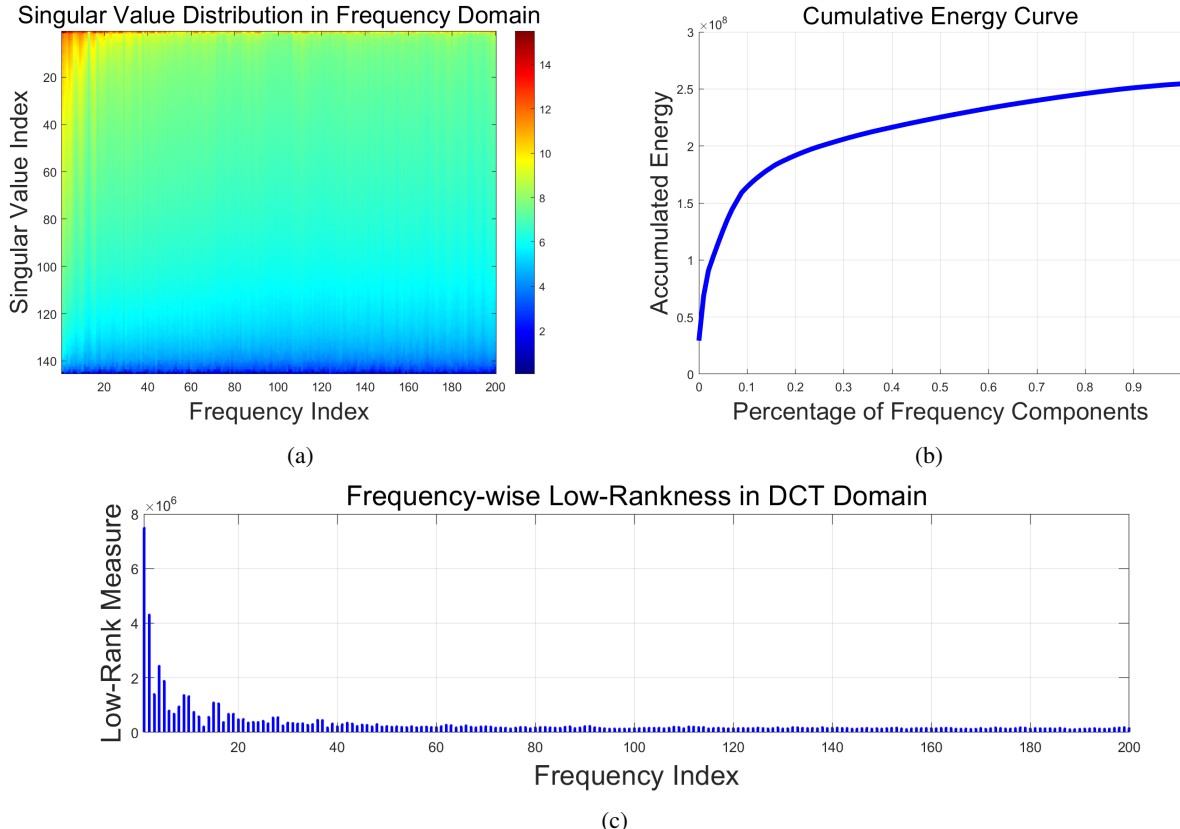

*Figure 4.* Visualization of dual-level sparsity structure using Indian Pines dataset. (a) The singular value heatmap exhibits both inter-frequency sparsity (horizontal variation) and intra-frequency low-rankness (vertical variation). (b)The cumulative energy curve reveals a majority of energy concentration in first 20% frequencies. (c) The frequency-wise low-rank measure $\|\boldsymbol{\sigma}(M(\mathbf{\underline{T}})^{(i)})\|_1$ shows significant peaks in low frequencies and rapid decay afterwards.

- *Structural Similarity Index (SSIM)*:

$$\text{SSIM}(\hat{\underline{\mathbf{L}}}, \underline{\mathbf{L}}^{\star}) = \frac{(2\mu_{\hat{\underline{\mathbf{L}}}}\mu_{\underline{\mathbf{L}}^{\star}} + c_1)(2\sigma_{\hat{\underline{\mathbf{L}}}\underline{\mathbf{L}}^{\star}} + c_2)}{(\mu_{\hat{\underline{\mathbf{L}}}}^2 + \mu_{\underline{\mathbf{L}}^{\star}}^2 + c_1)(\sigma_{\hat{\underline{\mathbf{L}}}}^2 + \sigma_{\underline{\mathbf{L}}^{\star}}^2 + c_2)}.$$

This metric measures perceptual similarity between the recovered tensor $\hat{\underline{\mathbf{L}}}$ and the ground truth $\underline{\mathbf{L}}^{\star}$, where $\mu_{\hat{\underline{\mathbf{L}}}}, \mu_{\underline{\mathbf{L}}^{\star}}$ denote mean values, $\sigma_{\hat{\underline{\mathbf{L}}}}, \sigma_{\underline{\mathbf{L}}^{\star}}$ denote standard deviations, and $\sigma_{\hat{\underline{\mathbf{L}}}\underline{\mathbf{L}}^{\star}}$ represents cross-covariance. Parameters $c_1$ and $c_2$ are small constants to stabilize the division.

These metrics together provide a comprehensive evaluation of the reconstruction performance, ensuring that both numerical fidelity and structural integrity are preserved.

**Benchmark Methods.** We compare the proposed $\ell_p(S_q)$-quasi-norm against several existing low-rank tensor regularization techniques:

- *NN*: Matrix nuclear norm (Candès & Tao, 2010)

- *SNN*: Tucker-based tensor nuclear norm (Liu et al., 2013)

- *TNN-DFT*: TNN with Discrete Fourier Transform (Zhang & Aeron, 2017)

- *TNN-DCT*: TNN with Discrete Cosine Transform (Lu et al., 2019b)

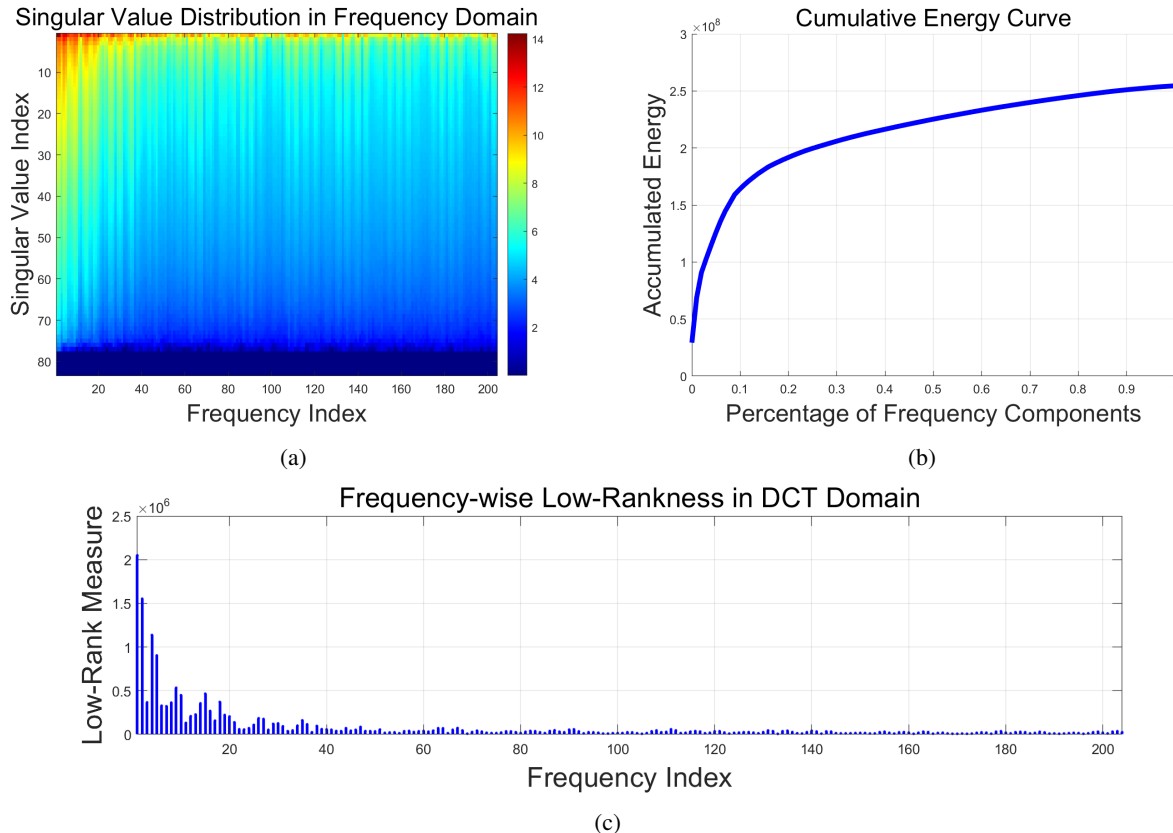

(a)  (b)

(c)

*Figure 5.* Visualization of dual-level sparsity structure using SalinasA dataset. (a) The singular value heatmap exhibits both inter-frequency sparsity (horizontal variation) and intra-frequency low-rankness (vertical variation). (b) The cumulative energy curve reveals a majority of energy concentration in first 20% frequencies. (c) The frequency-wise low-rank measure $\|\boldsymbol{\sigma}(M(\underline{\mathbf{T}})^{(i)})\|_1$ shows significant peaks in low frequencies and rapid decay afterwards.

- *k-Supp*: Tensor $k$-Support norm ($k = 2$) (Wang et al., 2021a)

- $\ell_{1-2}$*-norm*: Tensor $\ell_{1-2}$-norm (Tan et al., 2023)

- *Schatten-p-norm*: Tensor Schatten-$p$-norm ($p = 1/2$) (Kong et al., 2018)

- *LpSq (Proposed)*: The proposed $\ell_p(S_q)$-norm. We employ DCT as the default transform operator $M(\cdot)$.

For further comparison with different fixed transforms, see Appendix E.5; for comparison with adaptive (learnable) transforms, see Appendix E.6.

**Datasets.** The evaluation is conducted across multiple remote sensing datasets, encompassing hyperspectral, multispectral, and thermal imaging data.

1. *Hyperspectral Data.* We conduct noisy tensor completion on subsets of two representative hyperspectral datasets:

- *Indian Pines*: This dataset was collected by the AVIRIS sensor in 1992 over the Indian Pines[7] test site in Northwestern Indiana and consists of $145 \times 145$ pixels and 200 corrected spectral reflectance bands. We use the first 30 bands in the experiments due to computational constraints and parameter tuning.

---

[7]The dataset is publicly available at https://www.ehu.eus/ccwintco/index.php?title=Hyperspectral_Remote_Sensing_Scenes.

- *Salinas A*: Acquired by the AVIRIS sensor over the Salinas Valley, California, in 1998, this dataset consists of 224 bands over a spectral range of 400–2500 nm, with a spatial extent of $86 \times 83$ pixels at a resolution of 3.7m. We use the first 30 bands in the experiments[8].

2. *Multispectral Images.* Multispectral imaging captures image data within specific wavelength ranges across the electromagnetic spectrum and is one of the most widely used modalities in remote sensing. This section presents simulated experiments on multispectral images. The original data consists of three multispectral images: *Cloth*, *Hair*, and *Jelly Beans*, from the Columbia MSI Database[9]. These images contain scenes of a variety of real-world objects, each with a resolution of $512 \times 512 \times 31$, with intensity values scaled to [0,1].

3. *Thermal Imaging Data.* Thermal infrared data provide crucial measurements of surface energy fluxes and temperatures for various remote sensing applications. We conduct experiments on an infrared dataset: *OSU Thermal Database*[10]. The sequences were recorded on the Ohio State University campus during February and March 2005, capturing multiple individuals moving in groups through the scene. We use the first 30 frames of Sequence 1, forming a tensor of size $320 \times 240 \times 30$.

**Parameter Selection for** $(p, q)$**.** The parameters $(p, q)$ control the structural bias of the proposed $\ell_p(S_q)$ regularizer by balancing inter-frequency sparsity and intra-frequency low-rankness. In our experiments, we consider three representative configurations, denoted by **S1–S3**, with $(p, q) = (0.60, 0.61)$, $(0.70, 0.71)$, and $(0.80, 0.81)$, respectively.

As reported in Table 1, these configurations achieve comparable reconstruction performance across datasets and sampling ratios. This indicates that the empirical gains of the proposed framework are not sensitive to the precise choice of $(p, q)$ within a reasonable range, and do not rely on fine-grained parameter tuning. A more detailed dataset-level sensitivity analysis, including heatmaps over the $(p, q)$ grid, is provided in Appendix E.14.

### E.2. Tensor Completion via an ADMM-Based Algorithm

#### E.2.1. ALGORITHM DESCRIPTION

We aim to estimate a structured tensor $\underline{\mathbf{L}}$ from noisy and incomplete observations $\underline{\mathbf{Y}}$. The optimization problem is formulated as:

$$\min_{\underline{\mathbf{L}}} \frac{1}{2}\|\underline{\mathbf{Y}} - \underline{\mathbf{B}} \odot \underline{\mathbf{X}}\|_{\mathrm{F}}^2 + \lambda\|\underline{\mathbf{L}}\|_{\ell_p(S_q)}^p, \tag{39}$$

We introduce an auxiliary variable $\underline{\mathbf{K}}$ and reformulate the problem as:

$$\min_{\underline{\mathbf{L}},\underline{\mathbf{K}}} \quad \frac{1}{2}\|\underline{\mathbf{Y}} - \underline{\mathbf{B}} \odot \underline{\mathbf{K}}\|_{\mathrm{F}}^2 + \lambda\|\underline{\mathbf{L}}\|_{\ell_p(S_q)}^p, \quad \text{s.t.} \quad \underline{\mathbf{L}} = \underline{\mathbf{K}}.$$

This reformulation decouples the data fidelity term from the regularization, making it amenable to optimization via the Alternating Direction Method of Multipliers (ADMM).

We solve this problem using ADMM by iteratively updating $\underline{\mathbf{L}}$, $\underline{\mathbf{K}}$, and the scaled dual variable $\underline{\mathbf{W}}$.

- *Update of* $\underline{\mathbf{L}}$*:* The $\underline{\mathbf{L}}$-subproblem is:

$$\underline{\mathbf{L}}^{t+1} \in \arg\min_{\underline{\mathbf{L}}} \lambda\|\underline{\mathbf{L}}\|_{\ell_p(S_q)}^p + \frac{\mu}{2}\left\|\underline{\mathbf{L}} - \underline{\mathbf{K}}^t + \underline{\mathbf{W}}^t\right\|_{\mathrm{F}}^2.$$

Let $\underline{\mathbf{Z}} = \underline{\mathbf{K}}^t - \underline{\mathbf{W}}^t$, reducing the above problem to:

$$\underline{\mathbf{L}}^{t+1} \in \arg\min_{\underline{\mathbf{L}}} \lambda\|\underline{\mathbf{L}}\|_{\ell_p(S_q)}^p + \frac{\mu}{2}\|\underline{\mathbf{L}} - \underline{\mathbf{Z}}\|_{\mathrm{F}}^2.$$

---

[8]The dataset is available at https://www.ehu.eus/ccwintco/index.php?title=Hyperspectral_Remote_Sensing_Scenes.

[9]Available at https://cave.cs.columbia.edu/repository/Multispectral.

[10]The dataset is available at http://vcipl-okstate.org/pbvs/bench/Data/03/download.html.

Since $M(\underline{\mathbf{L}})$ allows separable updates across frequency components, this problem decomposes into $m$ independent subproblems:

$$\min \frac{1}{2} \left\| M(\underline{\mathbf{L}})_k - M(\underline{\mathbf{Z}})_k \right\|_{\mathrm{F}}^2 + \frac{\lambda}{\mu} \| M(\underline{\mathbf{L}})_k \|_{S_q}^p, \quad \forall k \in [m]. \tag{40}$$

To efficiently handle the Schatten-$q$ term, we employ a weighted $\ell_{1/2}$-norm approximation (see Section E.4 for comparison with $\ell_1$-norm approximation), which can be interpreted as a local majorization of $\| M(\underline{\mathbf{L}})_k \|_{S_q}^p$ obtained via first-order linearization at the current iterate $\underline{\mathbf{L}}^t$, followed by a separable surrogate on individual singular values:

$$\sum_{i=1}^{d} w_{i,k} \cdot \sigma_i(M(\underline{\mathbf{L}})_k)^{1/2},$$

where the weight

$$w_{i,k} = \left( \sum_{j=1}^{d} \varsigma_{j,k}^q + \epsilon \right)^{\frac{p}{q}-1} \left( \varsigma_{j,k}^{1/2} + \epsilon \right)^{2q-1},$$

with $\varsigma_{j,k} = \sigma_j(M(\underline{\mathbf{L}}^t)_k)$, where $\underline{\mathbf{L}}^t$ denotes the tensor $\underline{\mathbf{L}}$ at the $t$-th iteration.

This formulation leads to a closed-form $\ell_{1/2}$-soft-thresholding update for each singular value:

$$\sigma_i^{(t+1)}(M(\underline{\mathbf{L}})_k) = \mathcal{S}_\theta^{\ell_{1/2}}(\sigma_i(M(\underline{\mathbf{Z}})_k)),$$

where $\theta = \frac{\lambda}{\mu} w_{i,k}$ and the $\ell_{1/2}$-soft-thresholding operator (Xu, 2010) is defined as:

$$\mathcal{S}_\theta^{\ell_{1/2}}(\sigma) = \begin{cases} \phi_\theta(\sigma), & |\sigma| > \frac{3\sqrt[3]{54}}{4}\theta^{2/3}, \\ \{\phi_\theta(\sigma), 0\}, & |\sigma| = \frac{3\sqrt[3]{54}}{4}\theta^{2/3}, \\ 0, & |\sigma| < \frac{3\sqrt[3]{54}}{4}\theta^{2/3}. \end{cases} \tag{41}$$

Here, the function $\phi_\theta(\sigma)$ is given by:

$$\phi_\theta(\sigma) = \frac{2}{3}\sigma \left( 1 + \cos \left( \frac{2\pi}{3} - \frac{2}{3}\arccos \left( \frac{\theta}{8}|\sigma|^{-3/2} \right) \right) \right).$$

After updating singular values, the frequency component is reconstructed as:

$$M(\underline{\mathbf{L}}^{t+1})_k = \mathbf{U}_k \cdot \mathtt{diag}(\boldsymbol{\sigma}^{(t+1)}(M(\underline{\mathbf{Z}})_k)) \cdot \mathbf{V}_k^\top, \quad \forall k \in [m],$$

where $\mathbf{U}_k$ and $\mathbf{V}_k$ are the left and right singular matrices of $M(\underline{\mathbf{Z}})_k$. Finally, applying the inverse $M$-transform yields the updated tensor $\underline{\mathbf{L}}^{t+1}$.

- *Update of $\underline{\mathbf{K}}$:* The auxiliary variable $\underline{\mathbf{K}}$ is updated by solving:

$$\underline{\mathbf{K}}^{t+1} = \arg\min_{\underline{\mathbf{K}}} \frac{1}{2} \| \underline{\mathbf{Y}} - \underline{\mathbf{B}} \odot \underline{\mathbf{K}} \|_{\mathrm{F}}^2 + \frac{\mu}{2} \left\| \underline{\mathbf{K}} - \underline{\mathbf{L}}^{t+1} - \underline{\mathbf{W}}^t \right\|_{\mathrm{F}}^2.$$

This step ensures that the solution remains within the feasible constraint region.

- *Dual Variable Update:* The Lagrange multiplier is updated as:

$$\underline{\mathbf{W}}^{t+1} = \underline{\mathbf{W}}^t + (\underline{\mathbf{L}}^{t+1} - \underline{\mathbf{K}}^{t+1}).$$

This ADMM-based algorithm[11] (summarized in Algorithm 1) efficiently solves the dual-level sparse tensor completion problem by iteratively enforcing structured sparsity through proximal updates while maintaining computational efficiency. The proposed weighted $\ell_{1/2}$-soft-thresholding mechanism ensures that the non-convex Schatten-$q$ regularization is effectively handled in each iteration.

---

[11]In practical implementations, the penalty parameter $\mu^t$ can be updated adaptively to improve numerical stability and empirical convergence speed: $\mu^{t+1} = \min\{\gamma\mu^t, \mu_{\max}\}$, where $\gamma > 1$ is a predefined scaling factor.

---

**Algorithm 1** ADMM for dual-sparse tensor completion ($\ell_p$–Schatten-$q$)

---

1: **Input:** Observed tensor $\underline{\mathbf{Y}}$; binary mask $\underline{\mathbf{B}}$; transform $M$ (with inverse $M^{-1}$); regulariser $\lambda$; parameters $p, q$; tolerance $\varepsilon$; penalty $\mu > 0$.
2: **Output:** Completed tensor $\widehat{\underline{\mathbf{L}}}$.

3: ▷ *Initialization*
4: $\underline{\mathbf{L}}^0 \leftarrow \mathbf{0}, \underline{\mathbf{K}}^0 \leftarrow \underline{\mathbf{Y}}, \underline{\mathbf{W}}^0 \leftarrow \mathbf{0}$
5: $t \leftarrow 0$

6: **repeat**

7:    ▷ $\underline{\mathbf{L}}$*–update*
8:    $\underline{\mathbf{Z}} \leftarrow \underline{\mathbf{K}}^t - \underline{\mathbf{W}}^t$
9:    ▷ *Dual correction*
10:    **for** $k = 1, 2, \ldots, m$ **do**
11:       $\mathbf{U}_k \operatorname{diag}(\boldsymbol{\sigma}_k) \mathbf{V}_k^\top \leftarrow \operatorname{SVD}\big(M(\underline{\mathbf{Z}})_k\big)$
12:       Compute weights $w_{i,k} = \big(\sum_j \varsigma_{j,k}^q + \epsilon\big)^{\frac{p}{q}-1} \big(\varsigma_{i,k}^{1/2} + \epsilon\big)^{2q-1}$ where $\varsigma_{i,k} = \sigma_i(M(\underline{\mathbf{L}}^t)_k)$
13:       $\theta_{i,k} \leftarrow (\lambda/\mu)\, w_{i,k}$
14:       $\sigma_{i,k}^{\text{new}} \leftarrow \mathcal{S}_{\theta_{i,k}}^{\ell_{1/2}}(\sigma_{i,k})$
15:       ▷ *Half-thresholding, Eq. (41)*
16:       $M(\underline{\mathbf{L}}^{t+1})_k \leftarrow U_k \operatorname{diag}(\boldsymbol{\sigma}_k^{\text{new}}) V_k^\top$
17:    **end for**
18:    $\underline{\mathbf{L}}^{t+1} \leftarrow M^{-1}\big(\{M(\underline{\mathbf{L}}^{t+1})_k\}_{k=1}^m\big)$

19:    ▷ $\underline{\mathbf{K}}$*–update*
20:    $\underline{\mathbf{K}}^{t+1} \leftarrow \big(\underline{\mathbf{B}} \odot \underline{\mathbf{Y}} + \mu\,(\underline{\mathbf{L}}^{t+1} + \underline{\mathbf{W}}^t)\big) \oslash \big(\underline{\mathbf{B}} + \mu\mathbf{1}\big)$
21:    ▷ *Element-wise division* $\oslash$

22:    ▷ *Dual update*
23:    $\underline{\mathbf{W}}^{t+1} \leftarrow \underline{\mathbf{W}}^t + (\underline{\mathbf{L}}^{t+1} - \underline{\mathbf{K}}^{t+1})$
24:    $t \leftarrow t + 1$
25: **until** $\|\underline{\mathbf{K}}^t - \underline{\mathbf{L}}^t\|_{\text{F}}/\|\underline{\mathbf{Y}}\|_{\text{F}} < \varepsilon$
26: $\widehat{\underline{\mathbf{L}}} \leftarrow \underline{\mathbf{L}}^t$

---

### E.2.2. COMPLEXITY ANALYSIS OF ALGORITHM 1

Each iteration of our algorithm consists of two main computational steps: (i) a linear transform applied to $d_1 d_2$ tubes of length $m$, which can be implemented in $O(d_1 d_2 m \log m)$ time using FFT or DCT; and (ii) $m$ singular value decompositions of $d_1 \times d_2$ matrices, resulting in a complexity of $O(m d_1 d_2 \min(d_1, d_2))$. These operations coincide with the core computational primitives used in standard TNN solvers, implying that the per-iteration complexity remains within the same order of magnitude as existing TNN-based methods. These theoretical complexity estimates are consistent with the empirical runtime behavior observed in our experiments; see Figure 7 for a detailed comparison.

### E.3. Convergence Analysis of Algorithm 1

#### E.3.1. EMPIRICAL CONVERGENCE BEHAVIOR OF ALGORITHM 1

Our ADMM-based algorithm exhibits stable empirical convergence across all experiments. As illustrated in Figure 6, the objective value typically stabilizes within 200 iterations. This behavior is consistent across different sampling ratios and datasets.

#### E.3.2. THEORETICAL CONVERGENCE ANALYSIS OF ALGORITHM 1

**Challenges.** This subsection establishes the convergence properties of Algorithm 1 from a theoretical perspective. The algorithm combines a scaled ADMM splitting with a reweighted spectral surrogate in the $\underline{\mathbf{L}}$-update, designed to handle the nonconvex and nonsmooth nature of the transform-domain penalty. The main technical challenge stems from the interaction of three ingredients: (i) the nonconvex and non-Lipschitz structure of $F$, (ii) the iteration-dependent surrogate $Q_t$ induced

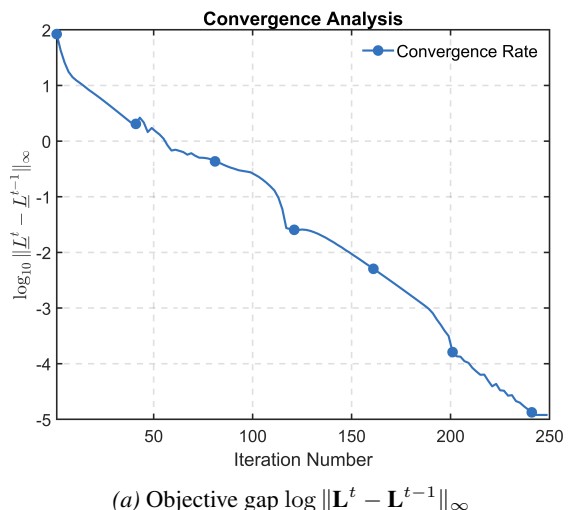

*(a)* Objective gap $\log \|\underline{\mathbf{L}}^t - \underline{\mathbf{L}}^{t-1}\|_\infty$

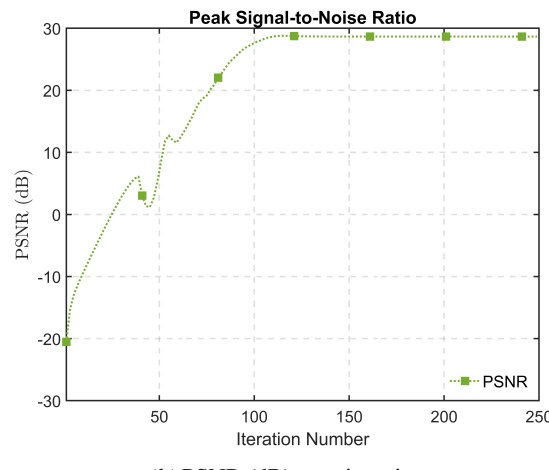

*(b)* PSNR (dB) over iterations

*Figure 6.* Empirical convergence of the proposed ADMM algorithm on the *Salinas A* dataset. Left: the variable gap between consecutive iterates. Right: the evolution of PSNR across iterations.

by reweighting, and (iii) the non-monotone behavior of the scaled augmented Lagrangian caused by the dual update.

**Analysis Strategy.** To address these issues, we construct a modified Lyapunov function that compensates for the surrogate mismatch and the scaled dual step. The convergence analysis is carried out under mild local regularity and consistency assumptions on the surrogate, without requiring global majorization. Then, we show that the resulting iteration enjoys sufficient descent, that the successive increments of the primal variables are square summable, and that the subgradient residual of the Lyapunov function vanishes asymptotically. By invoking the Kurdyka–Łojasiewicz framework (Attouch et al., 2013), these properties imply that the entire sequence generated by the algorithm converges to a critical point of the original constrained problem.

We analyze an ADMM scheme for the original constrained problem

$$\min_{\underline{\mathbf{L}},\underline{\mathbf{K}}} \ \lambda F(\underline{\mathbf{L}}) + g(\underline{\mathbf{K}}) \quad \text{s.t.} \quad \underline{\mathbf{L}} = \underline{\mathbf{K}}, \tag{42}$$

where

$$F(\underline{\mathbf{L}}) := \|M(\underline{\mathbf{L}})\|_{\ell_p(S_q)}^p = \sum_{\ell=1}^m \Big( \sum_{i=1}^d \sigma_i(M(\underline{\mathbf{L}})_\ell)^q \Big)^{\frac{p}{q}}, \tag{43}$$

and

$$g(\underline{\mathbf{K}}) := \frac{1}{2} \|\underline{\mathbf{B}} \odot (\underline{\mathbf{Y}} - \underline{\mathbf{K}})\|_{\mathrm{F}}^2. \tag{44}$$

The ADMM iterations are

$$\underline{\mathbf{L}}^{t+1} \in \arg\min_{\underline{\mathbf{L}}} \ \lambda Q_t(\underline{\mathbf{L}}) + \frac{\mu}{2} \|\underline{\mathbf{L}} - \underline{\mathbf{K}}^t + \underline{\mathbf{W}}^t\|_{\mathrm{F}}^2, \tag{45a}$$

$$\underline{\mathbf{K}}^{t+1} = \arg\min_{\underline{\mathbf{K}}} \ g(\underline{\mathbf{K}}) + \frac{\mu}{2} \|\underline{\mathbf{L}}^{t+1} - \underline{\mathbf{K}} + \underline{\mathbf{W}}^t\|_{\mathrm{F}}^2, \tag{45b}$$

$$\underline{\mathbf{W}}^{t+1} := \underline{\mathbf{W}}^t + (\underline{\mathbf{L}}^{t+1} - \underline{\mathbf{K}}^{t+1}). \tag{45c}$$

Here $Q_t$ is the reweighted spectral surrogate used in the $\underline{\mathbf{L}}$-update, for example, the weighted $\ell_{1/2}$ proxy that admits a closed-form thresholding step.

**Scaled Augmented Lagrangian and Primal Residual.** To analyze the dynamics of the scaled ADMM iterations Eq. (45), we introduce the *scaled augmented Lagrangian*

$$\Psi_\mu(\underline{\mathbf{L}}, \underline{\mathbf{K}}, \underline{\mathbf{W}}) := \lambda F(\underline{\mathbf{L}}) + g(\underline{\mathbf{K}}) + \frac{\mu}{2} \|\underline{\mathbf{L}} - \underline{\mathbf{K}} + \underline{\mathbf{W}}\|_{\mathrm{F}}^2 - \frac{\mu}{2} \|\underline{\mathbf{W}}\|_{\mathrm{F}}^2, \tag{46}$$

which corresponds to the standard augmented Lagrangian expressed in scaled dual variables. This formulation is particularly convenient for tracking the effect of the dual update and for isolating the non-monotone contribution induced by scaling.

In addition, we define the *primal residual* at iteration $t$ as

$$r^t := \underline{\mathbf{L}}^t - \underline{\mathbf{K}}^t,$$

which measures the violation of the linear constraint $\underline{\mathbf{L}} = \underline{\mathbf{K}}$. The residual $r^t$ plays a central role in the subsequent analysis: it directly governs the dual increment $\underline{\mathbf{W}}^{t+1} - \underline{\mathbf{W}}^t$, and its interaction with the primal updates determines whether a global descent property can be recovered by an appropriate Lyapunov function.

**Surrogate Consistency Conditions.**    In contrast to classical MM-based analyses that require the surrogate to globally majorize the original objective, the reweighted spectral proxies used in Algorithm 1 are only required to approximate $F$ with controlled local accuracy. In particular, the surrogate $Q_t$ is designed to be tight at the current iterate and to admit a tractable minimization step, but it need not be a global upper bound of $F$. Instead, we impose a quadratic model-mismatch condition on bounded regions, which is sufficient for absorbing the surrogate error by the square-summable primal increments in the Lyapunov analysis. Throughout this analysis, we assume that the sequence $\{(\underline{\mathbf{L}}^t, \underline{\mathbf{K}}^t, \underline{\mathbf{W}}^t)\}$ generated by Algorithm 1 is bounded. The following conditions formalize this notion of *inexact surrogate consistency* with respect to the original objective $F$.

**Condition E.1** (Local inexact surrogate consistency).  For each $t$, the surrogate $Q_t$ satisfies the following local consistency properties, which are only required on bounded regions and are standard in the analysis of reweighted nonconvex surrogates:

(S1) (Tightness) $Q_t(\underline{\mathbf{L}}^t) = F(\underline{\mathbf{L}}^t)$.

(S2′) (Quadratic model mismatch on bounded sets) There exists $C_M > 0$ such that on bounded sets, for all $\underline{\mathbf{L}}$,

$$F(\underline{\mathbf{L}}) \le Q_t(\underline{\mathbf{L}}) + \frac{C_M}{2}\|\underline{\mathbf{L}} - \underline{\mathbf{L}}^t\|_{\mathrm{F}}^2, \qquad \text{and} \qquad F(\underline{\mathbf{L}}^t) = Q_t(\underline{\mathbf{L}}^t). \tag{47}$$

(S3) (First-order consistency) There exists $C_Q > 0$ such that on bounded sets one can select $\zeta \in \partial Q_t(\underline{\mathbf{L}})$ and $\xi \in \partial F(\underline{\mathbf{L}})$ satisfying

$$\|\xi - \zeta\|_{\mathrm{F}} \le C_Q\|\underline{\mathbf{L}} - \underline{\mathbf{L}}^t\|_{\mathrm{F}}. \tag{48}$$

where $\partial$ denotes the limiting sub-differential (Attouch et al., 2013).

(S4) (Solvability) The $\underline{\mathbf{L}}$-subproblem Eq. (45a) admits an exact minimizer.

*Remark* E.2.  The surrogate sequence $\{Q_t\}$ considered here is not required to satisfy a classical MM majorization property. Instead, the convergence analysis relies on local consistency between $Q_t$ and the original objective $F$ on bounded regions, as captured by Condition E.1. Such conditions are mild and standard in the analysis of reweighted nonconvex spectral methods, and are only introduced to facilitate the Lyapunov-based convergence analysis.

Condition E.1 provides an abstract description of the local approximation properties of the reweighted surrogate $Q_t$ used in the $\underline{\mathbf{L}}$-update. It reflects the fact that the surrogate is constructed to be tight at the current iterate and to capture the local geometry of the original nonconvex spectral penalty $F$.

In the present setting, the reweighting rule is $\epsilon$-regularized[12]. Together with the boundedness of the iterates assumed in the analysis, this regularization is sufficient to justify the local surrogate consistency required in Condition E.1.

We emphasize that the quadratic model-mismatch bound Eq. (47) is only invoked to absorb the surrogate mismatch via the square-summable increment $\|\underline{\mathbf{L}}^{t+1} - \underline{\mathbf{L}}^t\|_{\mathrm{F}}^2$ in the Lyapunov descent argument, and does not affect the algorithmic updates themselves.

---

[12]The reweighting rule is $\epsilon$-regularized in the sense that all weight coefficients are constructed from smoothed singular values of the form $(\sigma_i + \epsilon)^\alpha$ for some fixed $\epsilon > 0$, thereby avoiding singular behavior at vanishing singular values.

**Dual-Step Identity.** The next lemma isolates the contribution of the scaled dual update to the variation of the augmented Lagrangian.

**Lemma E.3** (Dual-step identity for the scaled augmented Lagrangian). *Let $r^{t+1} := \underline{\mathbf{L}}^{t+1} - \underline{\mathbf{K}}^{t+1}$. Then*

$$\Psi_\mu(\underline{\mathbf{L}}^{t+1}, \underline{\mathbf{K}}^{t+1}, \underline{\mathbf{W}}^{t+1}) = \Psi_\mu(\underline{\mathbf{L}}^{t+1}, \underline{\mathbf{K}}^{t+1}, \underline{\mathbf{W}}^t) + \mu\|r^{t+1}\|_{\mathrm{F}}^2. \tag{49}$$

*Proof.* Fix $(\underline{\mathbf{L}}^{t+1}, \underline{\mathbf{K}}^{t+1})$ and expand the quadratic terms in Eq. (46) under $\underline{\mathbf{W}}^{t+1} = \underline{\mathbf{W}}^t + r^{t+1}$. □

**Descent of the $\underline{\mathbf{K}}$-Update and Residual Control.** We first establish a standard descent property of the $\underline{\mathbf{K}}$-update, which will later be combined with the dual-step identity to control the primal residual.

**Lemma E.4** (Strong convexity descent of the $\underline{\mathbf{K}}$-update). *For any fixed $(\underline{\mathbf{L}}^{t+1}, \underline{\mathbf{W}}^t)$, the function $\underline{\mathbf{K}} \mapsto g(\underline{\mathbf{K}}) + \frac{\mu}{2}\|\underline{\mathbf{L}}^{t+1} - \underline{\mathbf{K}} + \underline{\mathbf{W}}^t\|_{\mathrm{F}}^2$ is $\mu$-strongly convex. Consequently,*

$$g(\underline{\mathbf{K}}^t) + \frac{\mu}{2}\|\underline{\mathbf{L}}^{t+1} - \underline{\mathbf{K}}^t + \underline{\mathbf{W}}^t\|_{\mathrm{F}}^2 \geq g(\underline{\mathbf{K}}^{t+1}) + \frac{\mu}{2}\|\underline{\mathbf{L}}^{t+1} - \underline{\mathbf{K}}^{t+1} + \underline{\mathbf{W}}^t\|_{\mathrm{F}}^2 + \frac{\mu}{2}\|\underline{\mathbf{K}}^{t+1} - \underline{\mathbf{K}}^t\|_{\mathrm{F}}^2. \tag{50}$$

*Proof.* The quadratic penalty is $\mu$-strongly convex in $\underline{\mathbf{K}}$ and $g$ is convex, hence their sum is $\mu$-strongly convex. □

The next lemma exploits the specific quadratic structure of $g$ to obtain a sharp control of the primal residual via the dual increment.

**Lemma E.5** (Residual control via dual increment). *Let $g(\underline{\mathbf{K}}) = \frac{1}{2}\|\underline{\mathbf{B}} \odot (\underline{\mathbf{Y}} - \underline{\mathbf{K}})\|_{\mathrm{F}}^2$. Then $\nabla g$ is $1$-Lipschitz and*

$$\|r^{t+1}\|_{\mathrm{F}} \leq \frac{1}{\mu}\|\underline{\mathbf{K}}^{t+1} - \underline{\mathbf{K}}^t\|_{\mathrm{F}}. \tag{51}$$

*Proof.* We have $\nabla g(\underline{\mathbf{K}}) = \underline{\mathbf{B}} \odot (\underline{\mathbf{K}} - \underline{\mathbf{Y}})$, and therefore

$$\|\nabla g(\underline{\mathbf{K}}_1) - \nabla g(\underline{\mathbf{K}}_2)\|_{\mathrm{F}} = \|\underline{\mathbf{B}} \odot (\underline{\mathbf{K}}_1 - \underline{\mathbf{K}}_2)\|_{\mathrm{F}} \leq \|\underline{\mathbf{K}}_1 - \underline{\mathbf{K}}_2\|_{\mathrm{F}},$$

which shows that $\nabla g$ is $1$-Lipschitz. The optimality condition of Eq. (45b) yields

$$\nabla g(\underline{\mathbf{K}}^{t+1}) = \mu(\underline{\mathbf{L}}^{t+1} - \underline{\mathbf{K}}^{t+1} + \underline{\mathbf{W}}^t) = \mu\underline{\mathbf{W}}^{t+1},$$

and similarly $\nabla g(\underline{\mathbf{K}}^t) = \mu\underline{\mathbf{W}}^t$. Subtracting the two relations gives

$$\mu r^{t+1} = \nabla g(\underline{\mathbf{K}}^{t+1}) - \nabla g(\underline{\mathbf{K}}^t).$$

Applying the Lipschitz property completes the proof. □

**An Inexact $\underline{\mathbf{L}}$-Step Inequality for the Original $F$.** The following lemma transfers the optimality of the surrogate $\underline{\mathbf{L}}$-update to an inexact descent inequality for the original objective $F$.

**Lemma E.6** (Inexact descent inequality induced by the $\underline{\mathbf{L}}$-update). *Under Condition E.1, the $\underline{\mathbf{L}}$-update Eq. (45a) satisfies*

$$\lambda F(\underline{\mathbf{L}}^{t+1}) + \frac{\mu}{2}\|\underline{\mathbf{L}}^{t+1} - \underline{\mathbf{K}}^t + \underline{\mathbf{W}}^t\|_{\mathrm{F}}^2 \leq \lambda F(\underline{\mathbf{L}}^t) + \frac{\mu}{2}\|\underline{\mathbf{L}}^t - \underline{\mathbf{K}}^t + \underline{\mathbf{W}}^t\|_{\mathrm{F}}^2 + \frac{\lambda C_M}{2}\|\underline{\mathbf{L}}^{t+1} - \underline{\mathbf{L}}^t\|_{\mathrm{F}}^2. \tag{52}$$

*Proof.* By optimality of Eq. (45a),

$$\lambda Q_t(\underline{\mathbf{L}}^{t+1}) + \frac{\mu}{2}\|\underline{\mathbf{L}}^{t+1} - \underline{\mathbf{K}}^t + \underline{\mathbf{W}}^t\|_{\mathrm{F}}^2 \leq \lambda Q_t(\underline{\mathbf{L}}^t) + \frac{\mu}{2}\|\underline{\mathbf{L}}^t - \underline{\mathbf{K}}^t + \underline{\mathbf{W}}^t\|_{\mathrm{F}}^2.$$

Using the tightness condition (S1) in Condition E.1, we have $Q_t(\underline{\mathbf{L}}^t) = F(\underline{\mathbf{L}}^t)$. Moreover, applying the quadratic model-mismatch bound (S2′) at $\underline{\mathbf{L}} = \underline{\mathbf{L}}^{t+1}$ yields

$$F(\underline{\mathbf{L}}^{t+1}) \leq Q_t(\underline{\mathbf{L}}^{t+1}) + \frac{C_M}{2}\|\underline{\mathbf{L}}^{t+1} - \underline{\mathbf{L}}^t\|_{\mathrm{F}}^2.$$

Combining these two inequalities yields Eq. (52). □

**Lyapunov Function and Sufficient Descent.** To compensate for the non-monotonic effect of the scaled dual update, we introduce a modified Lyapunov function that augments the scaled augmented Lagrangian with residual and memory terms. Define the Lyapunov function

$$\Phi^t := \Psi_\mu(\underline{\mathbf{L}}^t, \underline{\mathbf{K}}^t, \underline{\mathbf{W}}^t) + \frac{\mu}{2}\|r^t\|_{\mathrm{F}}^2 + \frac{\alpha}{2}\|\underline{\mathbf{K}}^t - \underline{\mathbf{K}}^{t-1}\|_{\mathrm{F}}^2, \qquad \alpha \in (0, \mu), \tag{53}$$

with the convention $\underline{\mathbf{K}}^{-1} = \underline{\mathbf{K}}^0$.

For a sufficiently large penalty parameter $\mu$, the following sufficient descent holds.

**Proposition E.7** (Sufficient descent of the Lyapunov function). *Assume Condition E.1 and Lemmas E.3–E.6. There exist $\alpha \in (0, \mu)$ and $c > 0$ such that for all $t \geq 1$,*

$$\Phi^t - \Phi^{t+1} \geq c\Big(\|\underline{\mathbf{K}}^{t+1} - \underline{\mathbf{K}}^t\|_{\mathrm{F}}^2 + \|\underline{\mathbf{K}}^t - \underline{\mathbf{K}}^{t-1}\|_{\mathrm{F}}^2\Big). \tag{54}$$

*Proof.* Let $\Delta\underline{\mathbf{K}}^t := \underline{\mathbf{K}}^{t+1} - \underline{\mathbf{K}}^t$ and $\Delta\underline{\mathbf{L}}^t := \underline{\mathbf{L}}^{t+1} - \underline{\mathbf{L}}^t$. Combine Lemma E.6 and Lemma E.4 to obtain

$$\Psi_\mu(\underline{\mathbf{L}}^t, \underline{\mathbf{K}}^t, \underline{\mathbf{W}}^t) - \Psi_\mu(\underline{\mathbf{L}}^{t+1}, \underline{\mathbf{K}}^{t+1}, \underline{\mathbf{W}}^t) \geq \frac{\mu}{2}\|\Delta\underline{\mathbf{K}}^t\|_{\mathrm{F}}^2 - \frac{\lambda C_M}{2}\|\Delta\underline{\mathbf{L}}^t\|_{\mathrm{F}}^2. \tag{55}$$

By Lemma E.3,

$$\Psi_\mu(\underline{\mathbf{L}}^{t+1}, \underline{\mathbf{K}}^{t+1}, \underline{\mathbf{W}}^{t+1}) = \Psi_\mu(\underline{\mathbf{L}}^{t+1}, \underline{\mathbf{K}}^{t+1}, \underline{\mathbf{W}}^t) + \mu\|r^{t+1}\|_{\mathrm{F}}^2. \tag{56}$$

Combining Eq. (55) and Eq. (56) gives

$$\Psi_\mu(\underline{\mathbf{L}}^t, \underline{\mathbf{K}}^t, \underline{\mathbf{W}}^t) - \Psi_\mu(\underline{\mathbf{L}}^{t+1}, \underline{\mathbf{K}}^{t+1}, \underline{\mathbf{W}}^{t+1}) \geq \frac{\mu}{2}\|\Delta\underline{\mathbf{K}}^t\|_{\mathrm{F}}^2 - \mu\|r^{t+1}\|_{\mathrm{F}}^2 - \frac{\lambda C_M}{2}\|\Delta\underline{\mathbf{L}}^t\|_{\mathrm{F}}^2. \tag{57}$$

By the definition Eq. (53),

$$\Phi^t - \Phi^{t+1} = \Big[\Psi_\mu(\underline{\mathbf{L}}^t, \underline{\mathbf{K}}^t, \underline{\mathbf{W}}^t) - \Psi_\mu(\underline{\mathbf{L}}^{t+1}, \underline{\mathbf{K}}^{t+1}, \underline{\mathbf{W}}^{t+1})\Big] + \frac{\mu}{2}\big(\|r^t\|_{\mathrm{F}}^2 - \|r^{t+1}\|_{\mathrm{F}}^2\big)$$
$$+ \frac{\alpha}{2}\big(\|\Delta\underline{\mathbf{K}}^{t-1}\|_{\mathrm{F}}^2 - \|\Delta\underline{\mathbf{K}}^t\|_{\mathrm{F}}^2\big). \tag{58}$$

Substitute Eq. (57) into Eq. (58) to obtain

$$\Phi^t - \Phi^{t+1} \geq \frac{\mu - \alpha}{2}\|\Delta\underline{\mathbf{K}}^t\|_{\mathrm{F}}^2 + \frac{\alpha}{2}\|\Delta\underline{\mathbf{K}}^{t-1}\|_{\mathrm{F}}^2 + \frac{\mu}{2}\|r^t\|_{\mathrm{F}}^2 - \frac{3\mu}{2}\|r^{t+1}\|_{\mathrm{F}}^2 - \frac{\lambda C_M}{2}\|\Delta\underline{\mathbf{L}}^t\|_{\mathrm{F}}^2. \tag{59}$$

Next we bound $\|\Delta\underline{\mathbf{L}}^t\|_{\mathrm{F}}^2$ by $\|\Delta\underline{\mathbf{K}}^t\|_{\mathrm{F}}^2$ and $\|\Delta\underline{\mathbf{K}}^{t-1}\|_{\mathrm{F}}^2$. Since $\underline{\mathbf{L}}^t = \underline{\mathbf{K}}^t + r^t$,

$$\Delta\underline{\mathbf{L}}^t = \Delta\underline{\mathbf{K}}^t + (r^{t+1} - r^t), \qquad \|r^{t+1} - r^t\|_{\mathrm{F}} \leq \|r^{t+1}\|_{\mathrm{F}} + \|r^t\|_{\mathrm{F}}.$$

Using Lemma E.5 at $k$ and $k - 1$ gives

$$\|r^{t+1}\|_{\mathrm{F}} \leq \frac{1}{\mu}\|\Delta\underline{\mathbf{K}}^t\|_{\mathrm{F}}, \qquad \|r^k\|_{\mathrm{F}} \leq \frac{1}{\mu}\|\Delta\underline{\mathbf{K}}^{t-1}\|_{\mathrm{F}}.$$

Hence

$$\|\Delta\underline{\mathbf{L}}^t\|_{\mathrm{F}} \leq \|\Delta\underline{\mathbf{K}}^t\|_{\mathrm{F}} + \frac{1}{\mu}\|\Delta\underline{\mathbf{K}}^t\|_{\mathrm{F}} + \frac{1}{\mu}\|\Delta\underline{\mathbf{K}}^{t-1}\|_{\mathrm{F}} = \Big(1 + \frac{1}{\mu}\Big)\|\Delta\underline{\mathbf{K}}^t\|_{\mathrm{F}} + \frac{1}{\mu}\|\Delta\underline{\mathbf{K}}^{t-1}\|_{\mathrm{F}}.$$

Using $(a + b)^2 \leq 2a^2 + 2b^2$, we obtain

$$\|\Delta\underline{\mathbf{L}}^t\|_{\mathrm{F}}^2 \leq 2\Big(1 + \frac{1}{\mu}\Big)^2\|\Delta\underline{\mathbf{K}}^t\|_{\mathrm{F}}^2 + \frac{2}{\mu^2}\|\Delta\underline{\mathbf{K}}^{t-1}\|_{\mathrm{F}}^2. \tag{60}$$

Also, Lemma E.5 yields

$$\|r^{t+1}\|_{\mathrm{F}}^2 \leq \frac{1}{\mu^2}\|\Delta\underline{\mathbf{K}}^t\|_{\mathrm{F}}^2. \tag{61}$$

Substitute Eq. (60) and Eq. (61) into Eq. (59) and drop the nonnegative term $\frac{\mu}{2}\|r^t\|_{\mathrm{F}}^2$:

$$\Phi^t - \Phi^{t+1} \geq \Big[\frac{\mu - \alpha}{2} - \frac{3}{2\mu} - \lambda C_M\Big(1 + \frac{1}{\mu}\Big)^2\Big]\|\Delta\underline{\mathbf{K}}^t\|_{\mathrm{F}}^2 + \Big[\frac{\alpha}{2} - \frac{\lambda C_M}{\mu^2}\Big]\|\Delta\underline{\mathbf{K}}^{t-1}\|_{\mathrm{F}}^2. \tag{62}$$

Choose $\alpha \in (0, \mu)$ and $\mu > 0$ such that both bracketed coefficients in Eq. (62) are strictly positive. Then Eq. (54) holds with

$$c := \min\Big\{\frac{\mu - \alpha}{2} - \frac{3}{2\mu} - \lambda C_M\Big(1 + \frac{1}{\mu}\Big)^2, \ \frac{\alpha}{2} - \frac{\lambda C_M}{\mu^2}\Big\} > 0.$$

$\square$

*Remark* E.8 (A convenient parameter choice). A simple sufficient choice is to set $\alpha = \mu/2$ and then pick $\mu$ large enough so that

$$\frac{\mu}{4} - \frac{3}{2\mu} - \lambda C_M\Big(1 + \frac{1}{\mu}\Big)^2 > 0, \qquad \frac{\mu}{4} - \frac{\lambda C_M}{\mu^2} > 0.$$

**Summability and Vanishing Increments.** As a direct consequence of the sufficient descent property of the Lyapunov function, we now establish summability of successive increments and asymptotic primal feasibility, which form the basis for the subsequent KL analysis.

**Corollary E.9** (Summability). $\{\Phi^t\}$ *converges and*

$$\sum_{t=0}^{\infty}\|\underline{\mathbf{K}}^{t+1} - \underline{\mathbf{K}}^t\|_{\mathrm{F}}^2 < \infty, \qquad \sum_{t=0}^{\infty}\|\underline{\mathbf{L}}^{t+1} - \underline{\mathbf{L}}^t\|_{\mathrm{F}}^2 < \infty, \qquad \|r^t\|_{\mathrm{F}} \to 0.$$

*Proof.* By Proposition E.7, $\{\Phi^t\}$ is nonincreasing. Moreover, in the present completion setting $g(\underline{\mathbf{K}}) = \frac{1}{2}\|\mathbf{B} \odot (\underline{\mathbf{Y}} - \underline{\mathbf{K}})\|_{\mathrm{F}}^2$, the $\underline{\mathbf{K}}$-optimality condition together with the scaled dual update yields $\nabla g(\underline{\mathbf{K}}^t) = \mu\underline{\mathbf{W}}^t$ (see the proof of Lemma E.5), and hence $g(\underline{\mathbf{K}}^t) - \frac{\mu}{2}\|\underline{\mathbf{W}}^t\|_{\mathrm{F}}^2 = \big(\frac{1}{2} - \frac{1}{2\mu}\big)\|\mathbf{B} \odot (\underline{\mathbf{Y}} - \underline{\mathbf{K}}^t)\|_{\mathrm{F}}^2 \geq 0$ for $\mu \geq 1$. Therefore $\Phi^t$ is bounded below, and thus $\{\Phi^t\}$ converges. Summing Eq. (54) gives $\sum_t \|\Delta\underline{\mathbf{K}}^t\|_{\mathrm{F}}^2 < \infty$. Then Lemma E.5 gives $r^t \to 0$. Finally, Eq. (60) and $\sum_t \|\Delta\underline{\mathbf{K}}^t\|_{\mathrm{F}}^2 < \infty$ imply $\sum_t \|\Delta\underline{\mathbf{L}}^t\|_{\mathrm{F}}^2 < \infty$. $\square$

**Relative Error Condition.** For notational simplicity, we write $\partial\Phi(\underline{\mathbf{L}}^{t+1}, \underline{\mathbf{K}}^{t+1}, \underline{\mathbf{W}}^{t+1})$ to denote the limiting subgradient of the Lyapunov sequence $\{\Phi^t\}$; this corresponds to the standard extended-variable interpretation commonly adopted in KL-based analyses of descent methods. To invoke the Kurdyka–Łojasiewicz framework (Attouch et al., 2013), it remains to relate the limiting subgradient of the Lyapunov function to the successive variations of the iterates. This is achieved via the following relative error bound.

**Proposition E.10** (Relative error bound). *Under Condition E.1 and Corollary E.9, there exists $C > 0$ such that*

$$\mathrm{dist}\Big(0, \partial\Phi(\underline{\mathbf{L}}^{t+1}, \underline{\mathbf{K}}^{t+1}, \underline{\mathbf{W}}^{t+1})\Big) \leq C\Big(\|\underline{\mathbf{L}}^{t+1} - \underline{\mathbf{L}}^t\|_{\mathrm{F}} + \|\underline{\mathbf{K}}^{t+1} - \underline{\mathbf{K}}^t\|_{\mathrm{F}}\Big). \tag{63}$$

*Proof.* Let $\Delta\underline{\mathbf{L}}^t := \underline{\mathbf{L}}^{t+1} - \underline{\mathbf{L}}^t$ and $\Delta\underline{\mathbf{K}}^t := \underline{\mathbf{K}}^{t+1} - \underline{\mathbf{K}}^t$. We explicitly construct $v^{t+1} \in \partial\Phi(\underline{\mathbf{L}}^{t+1}, \underline{\mathbf{K}}^{t+1}, \underline{\mathbf{W}}^{t+1})$ and bound its norm.

First note that the limiting subdifferential of $\Phi(\underline{\mathbf{L}}, \underline{\mathbf{K}}, \underline{\mathbf{W}})$ at $(\underline{\mathbf{L}}^{t+1}, \underline{\mathbf{K}}^{t+1}, \underline{\mathbf{W}}^{t+1})$ satisfies

$$\partial_{\underline{\mathbf{L}}}\Phi = \lambda\partial F(\underline{\mathbf{L}}^{t+1}) + \mu(\underline{\mathbf{L}}^{t+1} - \underline{\mathbf{K}}^{t+1} + \underline{\mathbf{W}}^{t+1}),$$
$$\nabla_{\underline{\mathbf{K}}}\Phi = \nabla g(\underline{\mathbf{K}}^{t+1}) - \mu(\underline{\mathbf{L}}^{t+1} - \underline{\mathbf{K}}^{t+1} + \underline{\mathbf{W}}^{t+1}) + \alpha(\underline{\mathbf{K}}^{t+1} - \underline{\mathbf{K}}^t),$$
$$\nabla_{\underline{\mathbf{W}}}\Phi = \mu(\underline{\mathbf{L}}^{t+1} - \underline{\mathbf{K}}^{t+1}).$$

From the $\underline{\mathbf{L}}$-optimality condition of Eq. (45a),

$$0 \in \lambda\partial Q_t(\underline{\mathbf{L}}^{t+1}) + \mu(\underline{\mathbf{L}}^{t+1} - \underline{\mathbf{K}}^t + \underline{\mathbf{W}}^t).$$

Select $\zeta^{t+1} \in \partial Q_t(\underline{\mathbf{L}}^{t+1})$ satisfying the inclusion. By (S3), choose $\xi^{t+1} \in \partial F(\underline{\mathbf{L}}^{t+1})$ such that

$$\|\xi^{t+1} - \zeta^{t+1}\|_{\mathrm{F}} \leq C_Q\|\Delta\underline{\mathbf{L}}^t\|_{\mathrm{F}}.$$

Then

$$\lambda\xi^{t+1} + \mu(\underline{\mathbf{L}}^{t+1} - \underline{\mathbf{K}}^{t+1} + \underline{\mathbf{W}}^{t+1}) = \lambda(\xi^{t+1} - \zeta^{t+1}) + \lambda\zeta^{t+1} + \mu(\underline{\mathbf{L}}^{t+1} - \underline{\mathbf{K}}^{t+1} + \underline{\mathbf{W}}^{t+1})$$
$$= \lambda(\xi^{t+1} - \zeta^{t+1}) - \mu(\underline{\mathbf{L}}^{t+1} - \underline{\mathbf{K}}^{t} + \underline{\mathbf{W}}^{t}) + \mu(\underline{\mathbf{L}}^{t+1} - \underline{\mathbf{K}}^{t+1} + \underline{\mathbf{W}}^{t+1})$$
$$= \lambda(\xi^{t+1} - \zeta^{t+1}) + \mu(\underline{\mathbf{K}}^{t} - \underline{\mathbf{K}}^{t+1}) + \mu(\underline{\mathbf{W}}^{t+1} - \underline{\mathbf{W}}^{t}).$$

Since $\underline{\mathbf{W}}^{t+1} - \underline{\mathbf{W}}^{t} = r^{t+1}$ and $\underline{\mathbf{K}}^{t} - \underline{\mathbf{K}}^{t+1} = -\Delta\underline{\mathbf{K}}^{t}$, we obtain

$$\left\|\lambda\xi^{t+1} + \mu(\underline{\mathbf{L}}^{t+1} - \underline{\mathbf{K}}^{t+1} + \underline{\mathbf{W}}^{t+1})\right\|_{\mathrm{F}} \leq \lambda C_Q\|\Delta\underline{\mathbf{L}}^{t}\|_{\mathrm{F}} + \mu\|\Delta\underline{\mathbf{K}}^{t}\|_{\mathrm{F}} + \mu\|r^{t+1}\|_{\mathrm{F}}.$$

Next, the $\underline{\mathbf{K}}$-optimality condition of Eq. (45b) reads

$$\nabla g(\underline{\mathbf{K}}^{t+1}) - \mu(\underline{\mathbf{L}}^{t+1} - \underline{\mathbf{K}}^{t+1} + \underline{\mathbf{W}}^{t}) = 0,$$

hence

$$\nabla g(\underline{\mathbf{K}}^{t+1}) - \mu(\underline{\mathbf{L}}^{t+1} - \underline{\mathbf{K}}^{t+1} + \underline{\mathbf{W}}^{t+1}) = -\mu(\underline{\mathbf{W}}^{t+1} - \underline{\mathbf{W}}^{t}) = -\mu r^{t+1}.$$

Therefore,

$$\left\|\nabla g(\underline{\mathbf{K}}^{t+1}) - \mu(\underline{\mathbf{L}}^{t+1} - \underline{\mathbf{K}}^{t+1} + \underline{\mathbf{W}}^{t+1}) + \alpha(\underline{\mathbf{K}}^{t+1} - \underline{\mathbf{K}}^{t})\right\|_{\mathrm{F}} \leq \mu\|r^{t+1}\|_{\mathrm{F}} + \alpha\|\Delta\underline{\mathbf{K}}^{t}\|_{\mathrm{F}}.$$

Finally,

$$\|\mu(\underline{\mathbf{L}}^{t+1} - \underline{\mathbf{K}}^{t+1})\|_{\mathrm{F}} = \mu\|r^{t+1}\|_{\mathrm{F}}.$$

Collecting the three components, define $v^{t+1} := (v_L^{t+1}, v_K^{t+1}, v_W^{t+1})$ by choosing $v_L^{t+1} \in \lambda\partial F(\underline{\mathbf{L}}^{t+1}) + \mu(\underline{\mathbf{L}}^{t+1} - \underline{\mathbf{K}}^{t+1} + \underline{\mathbf{W}}^{t+1})$ using $\xi^{t+1}$, and $v_K^{t+1}, v_W^{t+1}$ as the displayed gradients above. Then $v^{t+1} \in \partial\Phi(\underline{\mathbf{L}}^{t+1}, \underline{\mathbf{K}}^{t+1}, \underline{\mathbf{W}}^{t+1})$ and

$$\|v^{t+1}\|_{\mathrm{F}} \leq C_1\|\Delta\underline{\mathbf{L}}^{t}\|_{\mathrm{F}} + C_2\|\Delta\underline{\mathbf{K}}^{t}\|_{\mathrm{F}} + C_3\|r^{t+1}\|_{\mathrm{F}}.$$

Using Lemma E.5, $\|r^{t+1}\|_{\mathrm{F}} \leq \frac{1}{\mu}\|\Delta\underline{\mathbf{K}}^{t}\|_{\mathrm{F}}$, we conclude Eq. (63). $\qquad\square$

**KŁ Convergence.** With the sufficient descent property, summability of increments, and the relative error bound established above, we are now in a position to invoke the Kurdyka–Łojasiewicz framework (Attouch et al., 2013). Recall that the Lyapunov function $\Phi$ is bounded below by construction. Moreover, since $p$ and $q$ are rational, the spectral penalty $F$ is a semi-algebraic function (cf. Proposition 4 in Xu et al. (2017)), and hence the Lyapunov function $\Phi$ satisfies the Kurdyka–Łojasiewicz property. By Proposition 4 in Xu et al. (2017), $\|X\|_{S_q}^q = \sum_i \sigma_i(X)^q$ is semi-algebraic for rational $q$, and thus $F$ is semi-algebraic as a finite composition of rational-power spectral functions.

**Theorem E.11** (Global convergence to a critical point of the original objective)**.** *The sequence* $\{(\underline{\mathbf{L}}^{t}, \underline{\mathbf{K}}^{t}, \underline{\mathbf{W}}^{t})\}$ *generated by Eq.* (45) *has finite length and converges to a critical point* $(\underline{\mathbf{L}}^{\star}, \underline{\mathbf{K}}^{\star}, \underline{\mathbf{W}}^{\star})$ *of* $\Phi$ *in the sense of limiting subgradients. Moreover,* $r^t \to 0$*, hence* $\underline{\mathbf{L}}^{\star} = \underline{\mathbf{K}}^{\star}$*, and* $\underline{\mathbf{L}}^{\star}$ *is a stationary point of Eq.* (42)*.*

*Proof.* Proposition E.7 gives sufficient descent of $\Phi^t$. Corollary E.9 gives square-summability of increments and $r^t \to 0$. Proposition E.10 yields the relative error condition. Since $\Phi$ is a sum of KŁ functions and quadratic terms, it is KŁ. The standard KŁ convergence theorem for descent methods implies finite length and convergence to a critical point. Finally, $r^t \to 0$ yields $\underline{\mathbf{L}}^{\star} = \underline{\mathbf{K}}^{\star}$ and stationarity for Eq. (42). $\qquad\square$

### E.4. Ablation Study on Weighted $\ell_{1/2}$ vs. Weighted $\ell_1$ Surrogates

To approximate the nonconvex Schatten-$q$ regularizer in Eq. (40) in a computationally efficient manner, we also considered weighted $\ell_1$ surrogates, in addition to the weighted $\ell_{1/2}$ formulation adopted in Appendix E.2. While the weighted $\ell_1$ formulation is convex and widely used, our initial experimental exploration on Salinas A and Indian Pines suggests that it is generally less competitive than the weighted $\ell_{1/2}$ surrogate for the proposed model.

As shown in Table 8, the weighted $\ell_{1/2}$ surrogate achieves better performance in most tested settings, especially on Indian Pines, where it consistently improves both PSNR and SSIM across all sampling rates. On Salinas A, the two surrogates exhibit mixed behavior: weighted $\ell_{1/2}$ performs better at the lower sampling rate, whereas weighted $\ell_1$ is more favorable for some metrics at higher sampling rates. Overall, these results suggest that the nonconvex weighted $\ell_{1/2}$ penalty can more effectively capture the spectral decay and low-rank characteristics of transformed tensor slices in many practical settings. Based on this empirical evidence, we adopt the weighted $\ell_{1/2}$ approximation throughout our method.

*Table 8.* Comparison of weighted $\ell_1$ and weighted $\ell_{1/2}$ surrogates for approximating the proposed regularizer on Salinas A and Indian Pines datasets under different sampling rates (SR) with $(p, q) = (0.80, 0.81)$.

| Dataset | SR | Metric | Weighted $\ell_1$ | Weighted $\ell_{1/2}$ |
|---|---|---|---|---|
| Salinas A | 5% | PSNR | 27.87 | 28.58 |
| | | SSIM | 0.7220 | 0.7388 |
| | 10% | PSNR | 30.53 | 30.76 |
| | | SSIM | 0.8182 | 0.8080 |
| | 15% | PSNR | 32.25 | 31.19 |
| | | SSIM | 0.8534 | 0.8159 |
| Indian Pines | 5% | PSNR | 25.82 | 26.46 |
| | | SSIM | 0.6015 | 0.6343 |
| | 10% | PSNR | 26.00 | 27.95 |
| | | SSIM | 0.6115 | 0.7112 |
| | 15% | PSNR | 26.49 | 28.49 |
| | | SSIM | 0.6395 | 0.7341 |

*Table 9.* PSNR/SSIM results on Salinas A under different transforms with $(p, q) = (0.80, 0.81)$.

| Method | SR = 5% | SR = 10% | SR = 15% |
|---|---|---|---|
| TNN-DFT | 22.55 / 0.5667 | 25.72 / 0.7027 | 28.06 / 0.7804 |
| Ours-DFT | 23.78 / 0.5556 | 27.17 / 0.6868 | 28.79 / 0.7428 |
| TNN-Random | 16.28 / 0.2331 | 22.21 / 0.4996 | 24.31 / 0.5692 |
| Ours-Random | 18.66 / 0.3073 | 23.70 / 0.4946 | 25.93 / 0.6003 |
| TNN-Oracle | 29.06 / 0.8362 | 32.04 / 0.8924 | 33.77 / 0.9169 |
| Ours-Oracle | **31.08 / 0.8467** | **33.95 / 0.9032** | **35.94 / 0.9322** |

### E.5. Ablation Study on the Effect of Different Transforms

Our main experiments employ the Discrete Cosine Transform (DCT) due to its ability to capture smooth spectral structures. To examine whether the observed gains stem primarily from the transform itself or from the proposed regularizer, we conducted preliminary experiments using alternative invertible transforms: the Discrete Fourier Transform (DFT), random orthogonal transforms (Lu et al., 2019b), and an oracle transform constructed from the left singular vectors of the mode-3 unfolding of the ground-truth tensor (Zhang & Ng, 2021). Results on Salinas A are summarized in Table 9. These results suggest that the proposed regularizer remains effective across different invertible transforms, with consistent advantages over TNN even in the absence of DCT preprocessing. This indicates that the observed improvements stem primarily from the dual spectral modeling itself rather than from a particular transform. At the same time, the oracle case highlights the potential of combining our framework with adaptive or learned transforms.

### E.6. Comparison with Stronger Baselines (Framelet TNN and Adaptive TNN)

To further assess the effectiveness of the proposed framework, we compare against more advanced TNN variants, including the Framelet TNN (Jiang et al., 2020) and Adaptive TNN (Kong et al., 2021). Following Zhang & Ng (2021), we use a variant of the Adaptive TNN which constructs the transform matrix at each iteration using the left singular vectors of the tensor's mode-3 unfolding, thereby avoiding explicit rank selection in Kong et al. (2021). For fairness, we adopt the same adaptive scheme for our method. Results on Salinas A are reported in Table 10.

These results show that our method achieves competitive or superior performance compared to both Framelet and Adaptive TNN. The gain arises not only from transform selection but also from the joint spectral modeling introduced by our regularizer. Moreover, the framework naturally unifies fixed and adaptive transforms, as well as classical low-rank models, within a single formulation.

*Table 10.* Comparison (PSNR/SSIM) with Framelet TNN and Adaptive TNN on Salinas A with $(p, q) = (0.80, 0.81)$.

| Method | SR = 5% | SR = 10% | SR = 15% |
|---|---|---|---|
| Framelet TNN | 26.01 / 0.6382 | 27.76 / 0.7159 | 28.55 / 0.7461 |
| Adaptive TNN | 27.76 / 0.8245 | 30.91 / 0.8846 | 32.90 / 0.9173 |
| Ours-Adaptive | **31.20 / 0.8400** | **33.21 / 0.8811** | **34.03 / 0.8931** |

## E.7. Noiseless Tensor Completion

To further examine the generality of the proposed framework, we conducted experiments on standard noiseless tensor completion. Results on Salinas A with $(p, q) = (0.80, 0.81)$ are reported in Table 11. These results confirm that our method maintains promising performance in the noiseless setting, complementing its robustness under noisy observations. This suggests that the benefits of dual spectral sparsity are not limited to noise suppression, but also extend to enhancing recovery quality in idealized conditions.

*Table 11.* Tensor completion (noiseless) results (PSNR/SSIM) on Salinas A.

| Method | SR = 5% | SR = 10% | SR = 15% |
|---|---|---|---|
| TNN-DFT | 22.95 / 0.5987 | 27.60 / 0.7874 | 28.28 / 0.7985 |
| TNN-DCT | 28.29 / 0.7976 | 33.24 / 0.9159 | 35.89 / 0.9506 |
| $k$-Supp | 22.85 / 0.5976 | 27.49 / 0.7708 | 28.19 / 0.7951 |
| $\ell_{1-2}$ | 23.17 / 0.6129 | 27.88 / 0.7942 | 30.75 / 0.8735 |
| Schatten-$1/2$ | 23.41 / 0.4937 | 28.77 / 0.7571 | 31.40 / 0.8475 |
| Framelet TNN | 28.56 / 0.7060 | 33.44 / 0.8673 | 35.75 / 0.9117 |
| Adaptive TNN | 31.54 / 0.8962 | 35.93 / 0.9538 | 38.45 / 0.9733 |
| Ours-DCT | 31.13 / 0.8393 | 36.11 / 0.9418 | 38.99 / 0.9702 |
| Ours-Adaptive | **33.72 / 0.9103** | **38.47 / 0.9663** | **41.12 / 0.9819** |

## E.8. Robustness to Higher Sampling Ratios

To further examine the stability of parameter settings at higher observation ratios, we evaluated the fixed configuration $(p, q) = (0.80, 0.81)$ with a constant regularization parameter $\lambda = 0.4$ on the Salinas A and OSU Thermal datasets. The corresponding PSNR and SSIM results are reported in Table 12. These results indicate that the fixed configuration generalizes well and continues to improve as more data become available, with no sign of performance collapse or overfitting at higher sampling ratios.

*Table 12.* Performance with $(p, q) = (0.80, 0.81)$ under higher sampling ratios.

| Sampling Ratio | Salinas A | | OSU Thermal | |
|---|---|---|---|---|
| | PSNR | SSIM | PSNR | SSIM |
| 30% | 35.04 | 0.9255 | 37.90 | 0.9693 |
| 40% | 36.28 | 0.9419 | 38.75 | 0.9728 |
| 50% | 37.13 | 0.9510 | 39.26 | 0.9741 |

## E.9. Robustness under Uniform Noise

Our theoretical analysis focuses on the Gaussian setting, which provides a clean statistical model for establishing minimax bounds. Nevertheless, the proposed framework is not inherently restricted to Gaussian noise. To assess robustness in practice, we conducted experiments with uniform noise, where each entry is sampled from $[0, u]$ with $u = 0.05 \cdot \|\underline{\mathbf{L}}\|_{\mathrm{F}} / \sqrt{d_1 d_2 d_3}$. Results on Salinas A are reported in Table 13. These results demonstrate resilience to mild non-Gaussian perturbations.

*Table 13.* Performance of the proposed method with parameter setting $(p, q) = (0.80, 0.81)$ on Salinas A under uniform noise.

| Sampling Ratio | PSNR | SSIM |
|---|---|---|
| 30% | 34.19 | 0.9495 |
| 40% | 34.90 | 0.9618 |
| 50% | 35.30 | 0.9678 |

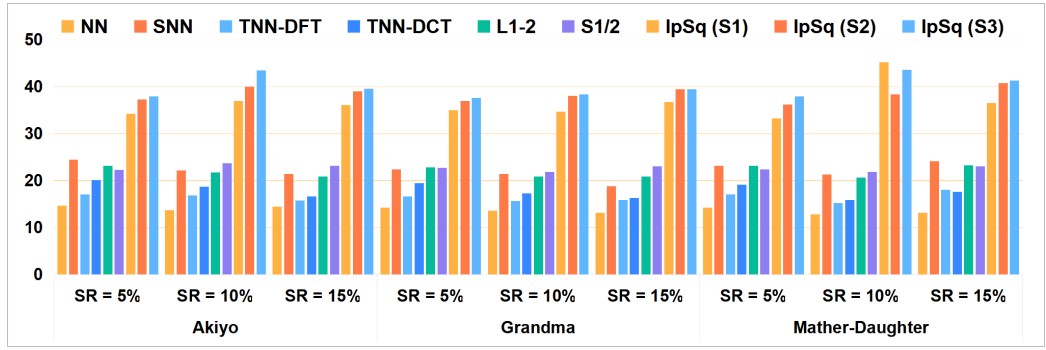

*Figure 7.* Runtime (in seconds) of different tensor completion methods on three YUV video sequences (*Akiyo*, *Grandma*, and *Mother-Daughter*) under varying sampling rates (5%, 10%, 15%).

### E.10. Extended Experiments to More 3D Tensor Data

#### E.10.1. EXTENDED EXPERIMENTS ON MORE VISUAL DATA: YUV VIDEOS

We further evaluate the proposed $\ell_p(S_q)$ regularizer on three YUV video sequences (*Akiyo*, *Grandma*, and *Mother-Daughter*), each converted to a $144 \times 176 \times 30$ tensor using only the luminance (Y) channel. These sequences exhibit diverse motion and spatiotemporal complexity, making them suitable for benchmarking real-world completion performance.

As summarized in Table 14, classical low-rank models such as NN, SNN, and TNN-based variants provide reasonable reconstructions but tend to saturate as the sampling ratio increases. In contrast, the proposed $\ell_p(S_q)$ configurations (**S1–S3**) deliver consistently improved recovery quality, particularly in the higher sampling regimes. This behavior is aligned with our observations on other modalities and suggests that the dual spectral sparsity captured by $\ell_p(S_q)$ remains effective for YUV video data, where the frequency content is more structured yet still exhibits significant variability across frames. To further assess the practical efficiency, we also compare the runtime of competing methods under varying sampling rates. As shown in Figure 7, the proposed $\ell_p(S_q)$ variants (**S1–S3**) incur approximately twice the runtime of the TNN-DCT method across all sampling ratios and datasets. Nonetheless, the runtime remains within the same order of magnitude as TNN-based methods. This observation is consistent with our theoretical complexity analysis. Moreover, given the noticeable gains in reconstruction accuracy, particularly in low-sampling regimes, we regard the modest computational overhead as a reasonable and practically acceptable tradeoff.

#### E.10.2. EXTENDED EXPERIMENTS ON NON-VISUAL DATA: SEISMIC DATA

To evaluate the generality of the proposed $\ell_p(S_q)$ regularizer beyond visual tensors, we further evaluate the proposed $\ell_p(S_q)$ regularizer on a *seismic* dataset[13], a representative example of non-visual tensor modalities. Due to computational constraints, we extract the first 30 frontal slices from *original_data_128* (depth 64), yielding a tensor of size $64 \times 128 \times 30$. Seismic data exhibit oscillatory and heterogeneous spectral patterns that differ markedly from the structured textures found in visual datasets. As shown in Table 15, the proposed $\ell_p(S_q)$ family achieves stable and high reconstruction quality across different sampling ratios on seismic tensors. The trend is consistent with what we observe on imaging data and suggests that the dual spectral structure captured by $\ell_p(S_q)$ naturally appears in non-visual tensor fields with richer spectral variability.

### E.11. Extended Experiments to Higher-Order Tensors

Our regularizer extends directly to higher-order tensors as long as a suitable transform and slice level singular values are well defined. Under this basic condition, the dual spectral sparsity principle carries over naturally to arbitrary tensor orders. This section outlines such an extension and provides preliminary evidence showing that the same modeling behavior persists beyond the third-order setting.

**A Strategy to Extend to Higher-Order Tensors.** For higher-order tensors with order $K > 3$, a natural extension of the proposed $\ell_p(S_q)$ regularizer is to generalize the spectral shrinkage scheme to multiple orientations. Following the spirit of orientation-invariant TNN (Wang et al., 2023b), one can first define $K$ distinct 3-way tensor unfoldings $\mathbf{T}_{[k]} := \mathfrak{F}_k(\mathbf{T})$

---

[13]The data is available at http://pan.baidu.com/s/1qYwI1IG.

*Table 14.* Noisy tensor completion results on three typical video datasets. The proposed $\ell_p(S_q)$ is reported under four parameter choices: Setting **S1** $(p, q) = (0.60, 0.61)$, Setting **S2** $(p, q) = (0.70, 0.71)$, and Setting **S3** $(p, q) = (0.80, 0.81)$.

| Dataset | SR | Metric | NN | SNN | TNN-DFT | TNN-DCT | $\ell_{1-2}$ | $S_{1/2}$ | S1 | S2 | S3 |
|---|---|---|---|---|---|---|---|---|---|---|---|
| *Akiyo* | 5% | PSNR | 15.37 | 19.49 | 27.04 | 26.58 | 27.00 | 26.89 | 27.63 | 27.97 | **28.09** |
| | | SSIM | 0.1864 | 0.6047 | 0.8019 | **0.8233** | 0.7974 | 0.7313 | 0.7715 | 0.7845 | 0.7875 |
| | 10% | PSNR | 18.01 | 22.54 | 29.18 | 28.91 | 29.39 | 28.95 | 30.53 | 30.57 | **30.59** |
| | | SSIM | 0.2858 | 0.7186 | 0.8556 | **0.8899** | 0.8576 | 0.7853 | 0.8625 | 0.8631 | 0.8580 |
| | 15% | PSNR | 19.64 | 24.37 | 30.60 | 29.96 | 30.70 | 29.98 | 32.30 | 32.33 | **32.39** |
| | | SSIM | 0.3694 | 0.7812 | 0.8791 | **0.9133** | 0.8803 | 0.8011 | 0.8973 | 0.8963 | 0.8945 |
| *Grandma* | 5% | PSNR | 16.52 | 19.13 | 28.53 | 28.26 | 28.68 | 29.12 | 29.96 | 30.01 | **30.14** |
| | | SSIM | 0.1928 | 0.5503 | 0.8135 | 0.8217 | 0.8134 | 0.7874 | 0.8205 | 0.8259 | **0.8263** |
| | 10% | PSNR | 18.34 | 22.52 | 31.44 | 31.15 | 31.67 | 31.77 | 33.50 | 33.56 | **33.68** |
| | | SSIM | 0.2992 | 0.6755 | 0.8822 | 0.8932 | 0.8852 | 0.8563 | 0.9031 | **0.9041** | 0.9037 |
| | 15% | PSNR | 19.66 | 24.88 | 33.02 | 32.55 | 33.23 | 32.95 | 35.43 | 35.49 | **35.52** |
| | | SSIM | 0.3757 | 0.7525 | 0.9088 | 0.9164 | 0.9097 | 0.8784 | **0.9332** | 0.9327 | 0.9307 |
| *Mother-Daughter* | 5% | PSNR | 16.17 | 20.22 | 27.66 | 26.85 | 27.51 | 27.28 | 28.18 | 28.45 | **28.60** |
| | | SSIM | 0.1895 | 0.5824 | 0.7491 | **0.7670** | 0.7441 | 0.6742 | 0.7317 | 0.7445 | 0.7504 |
| | 10% | PSNR | 18.55 | 23.48 | 29.38 | 28.76 | 29.56 | 28.34 | 30.62 | **30.66** | 30.52 |
| | | SSIM | 0.2780 | 0.6840 | 0.8035 | **0.8327** | 0.8058 | 0.7067 | 0.8212 | 0.8226 | 0.8135 |
| | 15% | PSNR | 20.32 | 25.31 | 30.49 | 29.56 | 30.54 | 28.80 | 31.62 | **31.69** | 31.65 |
| | | SSIM | 0.3548 | 0.7405 | 0.8293 | **0.8554** | 0.8305 | 0.7188 | 0.8474 | 0.8504 | 0.8457 |

via mode-$(k, k+1)$ 3D-unfolding, each capturing the spectral structure along a different orientation. Then, the $\ell_p(S_q)$ regularization can be applied independently to each $\underline{\mathbf{T}}_{[k]}$ in the transformed domain (e.g., DCT or DFT), and the total regularizer becomes a weighted sum across all orientations:

$$\mathcal{R}(\underline{\mathbf{T}}) := \sum_{k=1}^{K} \omega_k \| \mathcal{S}_q(\mathfrak{F}_k(\underline{\mathbf{T}})) \|_p^p,$$

where $\omega_k$ are positive weights satisfying $\sum_k \omega_k = 1$. We call this Orientation-Invariant $\ell_p(S_q)$ (OI-$\ell_p(S_q)$). This formulation allows $\ell_p(S_q)$ to capture joint spectral sparsity patterns across multiple dimensions, enhancing the model's flexibility and robustness for high-dimensional data. In practice, a simple yet effective choice is to set all weights equally and restrict to circular unfolding with $k = 1, \ldots, K$.

*Table 15.* Noisy tensor completion results on the *Seismic* dataset.

| Method | SR = 5% | | SR = 10% | | SR = 15% | |
|---|---|---|---|---|---|---|
| | PSNR | SSIM | PSNR | SSIM | PSNR | SSIM |
| TNN-DFT | 24.46 | 0.8830 | 27.64 | 0.9293 | 30.75 | 0.9558 |
| TNN-DCT | 32.31 | 0.9688 | 37.99 | 0.9878 | 41.25 | 0.9930 |
| $k$-Supp | 24.32 | 0.8785 | 27.72 | 0.9334 | 30.72 | 0.9521 |
| $\ell_{1-2}$ | 23.22 | 0.8563 | 28.38 | 0.9355 | 31.32 | 0.9583 |
| $S_{1/2}$ | 36.72 | 0.9783 | 41.20 | 0.9895 | 43.64 | 0.9935 |
| $\ell_p(S_q)$ (0.50,0.51) | 36.30 | 0.9776 | 42.15 | 0.9912 | 44.72 | 0.9948 |
| $\ell_p(S_q)$ (0.60,0.61) | 36.87 | 0.9765 | 41.99 | 0.9909 | 44.67 | 0.9943 |
| $\ell_p(S_q)$ (0.70,0.71) | 36.33 | 0.9779 | 42.16 | 0.9912 | 44.71 | 0.9942 |
| $\ell_p(S_q)$ (0.80,0.81) | 37.65 | 0.9821 | **43.35** | **0.9922** | 45.74 | **0.9953** |
| $\ell_p(S_q)$ (0.90,0.91) | 37.75 | **0.9830** | 43.04 | 0.9921 | 45.64 | 0.9952 |
| $\ell_p(S_q)$ (0.94,0.95) | **38.46** | 0.9828 | 43.15 | 0.9917 | **46.06** | 0.9951 |

**Preliminary Results.** We further assess the proposed model on a fourth-order tensor derived from the *Ground* dataset[14], commonly used in small object detection for aerial surveillance. The input consists of the first 30 frames, each of size $216 \times 288$ with 3 RGB channels, forming a $216 \times 288 \times 3 \times 30$ tensor.

To benchmark against matrix-based baselines, NN is applied along each unfolding mode of the tensor; the best result across all four modes is reported. Since TNN-based methods (TNN-DCT and TNN-DFT) are tailored to third-order inputs, we extract three grayscale videos corresponding to each color channel and apply these methods independently to each $216 \times 288 \times 30$ slice. For SqNN (Mu et al., 2014), we adopt the mode partition $\{1, 4\}$ vs. $\{2, 3\}$ and set $\lambda = \lambda_\iota \sigma \sqrt{p d_1 d_4 \log(d_1 d_4 + d_2 d_3)}$. SNN (Tomioka et al., 2011) uses weights $(1, 1, 0.01, 1)$ with global tuning. For OITNN and the proposed OI-$\ell_p(S_q)$, we adopt weights 1:100:1:1 and set $\lambda = \lambda_\iota \sigma \sqrt{p d_1 d_3 d_4 \log(d_1 d_3 d_4 + d_2 d_3)}$. We evaluate the methods under two sampling ratios ($p = 5\%, 10\%$).

Table 16 shows that the OI-$\ell_p(S_q)$ variant attains the strongest PSNR and SSIM at both sampling ratios on the *Ground* data, indicating that dual spectral sparsity remains effective in higher-order settings.

*Table 16.* The PSNR and SSIM values obtained by seven tensor norms (NN (Klopp et al., 2014), SqNN (Mu et al., 2014), SNN (Tomioka et al., 2011), TNN-DFT (Lu et al., 2019b), TNN-DCT (Lu et al., 2019b), OITNN-O (Wang et al., 2023b), and the proposed OI-$\ell_p(S_q)$ with $(p, q) = (0.50, 0.51)$ for noisy tensor completion on the 4D *Ground* data.

| SR | | NN | SqNN | SNN | TNN-DFT | TNN-DCT | OITNN-O | OI-$\ell_p(S_q)$ |
|---|---|---|---|---|---|---|---|---|
| 5% | PSNR | 24.83 | 26.80 | 25.03 | 30.90 | 30.94 | 32.85 | **34.35** |
| | SSIM | 0.7697 | 0.8011 | 0.8055 | 0.8955 | 0.8997 | 0.9306 | **0.9512** |
| 10% | PSNR | 27.03 | 29.10 | 27.36 | 32.27 | 32.49 | 34.25 | **36.47** |
| | SSIM | 0.8395 | 0.8725 | 0.8559 | 0.9083 | 0.9149 | 0.9382 | **0.9669** |

### E.12. Extended Experiments to Poisson Tensor Completion

We next extend our framework to the Poisson tensor completion setting, which arises naturally in photon-limited imaging and other count-based sensing modalities. Following the masked Poisson observation model introduced by Zhang & Ng (2021), each observed entry is generated through a signal-dependent Poisson process, leading to heteroscedastic noise and significant challenges in low-intensity regions.

We adopt the Bi-Module Tensor Regularization (BTR) framework of Wang et al. (2025a) and replace the quadratic fidelity term with the Poisson negative log-likelihood. The $\ell_p(S_q)$ spectral regularizer is then applied to both bi-module unfoldings, yielding a Bi–$\ell_p(S_q)$ formulation that preserves bidirectional spectral structure while remaining compatible with count-based measurements.

For experimentation, we follow the non-uniform sampling protocol of Wang et al. (2025a), in which structured dropout is applied along all tensor modes to obtain sampling ratios $\mathfrak{p} \in \{20\%, 30\%\}$. Given a ground-truth tensor $\underline{\mathbf{X}}^\star$, we normalize its dynamic range to $\|\underline{\mathbf{X}}^\star\|_\infty = 100$, set the background constant to $c = 5$, and generate Poisson measurements $\underline{\mathbf{Y}}$ via

$$\underline{\mathbf{P}}_{ijk} \sim \mathfrak{P}(\underline{\mathbf{X}}^\star_{ijk} + c), \quad \underline{\mathbf{Y}} = \underline{\mathbf{B}} \circledast \underline{\mathbf{P}},$$

where $\underline{\mathbf{B}}$ is the binary mask tensor, and $\circledast$ denotes element-wise product. We evaluate the resulting Bi–$\ell_p(S_q)$ model on hyperspectral, multispectral, LiDAR, and infrared tensors, providing a diverse testbed for assessing robustness under Poisson corruption and highly irregular sampling.

Table 17 summarizes the performance of Bi-$\ell_p(S_q)$ under non-uniform Poisson sampling. We compare against a broad set of representative baselines, including matrix-based PoissonNN (Cao & Xie, 2015), Tucker-based HaLRTC/SNN (Liu et al., 2013), factorization-based TMac (Xu et al., 2015), and transform-based TNN-DFT/DCT (Zhang & Ng, 2021). For completeness, we also include the recent bi-module tensor regularizers BTR-DFT and BTR-DCT proposed by Wang et al. (2025a). Across the datasets and sampling ratios, Bi-$\ell_p(S_q)$ consistently achieves the best or near-best PSNR and SSIM, outperforming both classical low-rank priors and modern transform-based tensor models. These results demonstrate that Bi-$\ell_p(S_q)$ provides a robust and adaptable prior for Poisson tensor completion across diverse sensing modalities and noise conditions.

---

[14] http://www.loujing.com/rss-small-target

*Table 17.* Poisson tensor completion results under non-uniform sampling at sampling rates of 20% and 30%. The compared methods include PoissonNN (Cao & Xie, 2015), SNN/HaLRTC (Liu et al., 2013), TMac (Xu et al., 2015), TNN-DFT/DCT (Zhang & Ng, 2021), and the two Bi-Module regularizers BTR-DFT and BTR-DCT from Wang et al. (2025a), abbreviated as BTR-F and BTR-C, respectively. Bi-$\ell_p(S_q)$ uses $(p, q) = (0.8, 0.81)$.

| Dataset | SR | Metric | PoissonNN | SNN | TMac | TNN-F | TNN-C | BTR-F | BTR-C | Bi-$\ell_p(S_q)$ |
|---|---|---|---|---|---|---|---|---|---|---|
| *Indian Pines* | 20% | PSNR | 7.77 | 23.07 | 24.21 | 24.65 | 25.90 | 28.73 | 28.44 | **30.05** |
| | | SSIM | 0.0089 | 0.4081 | 0.5481 | 0.4068 | 0.5044 | 0.7202 | 0.7151 | **0.7358** |
| | 30% | PSNR | 8.54 | 22.91 | 24.65 | 24.82 | 26.59 | 29.38 | 29.65 | **31.15** |
| | | SSIM | 0.0169 | 0.4163 | 0.5819 | 0.4221 | 0.5313 | 0.7273 | 0.7620 | **0.7881** |
| *SalinasA* | 20% | PSNR | 7.62 | 23.13 | 25.01 | 21.74 | 25.75 | 28.72 | 29.73 | **32.28** |
| | | SSIM | 0.0340 | 0.4511 | 0.6725 | 0.3637 | 0.5516 | 0.7589 | 0.7671 | **0.7839** |
| | 30% | PSNR | 8.40 | 23.17 | 25.39 | 22.28 | 26.92 | 30.48 | 31.48 | **34.55** |
| | | SSIM | 0.0487 | 0.4414 | 0.6888 | 0.3857 | 0.5963 | 0.8076 | 0.8161 | **0.8695** |
| *Cloth* | 20% | PSNR | 13.36 | 21.20 | 21.11 | 22.93 | 24.57 | 25.71 | 25.96 | **27.97** |
| | | SSIM | 0.0905 | 0.4450 | 0.4174 | 0.5686 | 0.6299 | 0.7212 | 0.7521 | **0.8005** |
| | 30% | PSNR | 13.36 | 21.50 | 21.83 | 23.36 | 25.24 | 26.63 | 27.16 | **28.51** |
| | | SSIM | 0.1451 | 0.5290 | 0.4915 | 0.6544 | 0.7261 | 0.7548 | 0.7912 | **0.8088** |
| *Oilpainting* | 20% | PSNR | 15.11 | 23.33 | 22.58 | 27.52 | 28.40 | 28.96 | 29.72 | **32.04** |
| | | SSIM | 0.0677 | 0.3391 | 0.3528 | 0.4211 | 0.4656 | 0.5538 | **0.6384** | 0.6262 |
| | 30% | PSNR | 13.95 | 22.30 | 21.38 | 26.33 | 27.23 | 29.57 | 30.66 | **33.07** |
| | | SSIM | 0.1357 | 0.4580 | 0.4525 | 0.5717 | 0.6240 | 0.5525 | 0.6334 | **0.6471** |
| *SenerioBDis* | 20% | PSNR | 13.24 | 17.61 | 18.54 | 20.11 | 20.05 | 20.18 | 20.13 | **21.53** |
| | | SSIM | 0.0767 | 0.2180 | 0.2285 | 0.3550 | 0.3907 | 0.4125 | **0.4492** | 0.4439 |
| | 30% | PSNR | 14.08 | 18.75 | 18.89 | 21.40 | 21.41 | 21.12 | 21.09 | **22.58** |
| | | SSIM | 0.1234 | 0.2628 | 0.2526 | 0.3848 | 0.4207 | 0.4309 | **0.4668** | 0.4572 |
| *SenerioBIntens* | 20% | PSNR | 14.19 | 18.06 | 17.76 | 20.45 | 20.47 | 21.06 | 20.65 | **21.80** |
| | | SSIM | 0.1205 | 0.3269 | 0.2688 | 0.4690 | 0.4769 | 0.5306 | 0.5106 | **0.5560** |
| | 30% | PSNR | 14.89 | 19.01 | 18.10 | 21.36 | 21.41 | 21.93 | 21.58 | **23.14** |
| | | SSIM | 0.1959 | 0.3998 | 0.2967 | 0.5279 | 0.5369 | 0.5664 | 0.5594 | **0.6041** |
| *Thermal* | 20% | PSNR | 10.53 | 19.29 | 15.46 | 25.66 | 26.01 | 26.08 | 26.50 | **29.38** |
| | | SSIM | 0.0724 | 0.3806 | 0.2096 | 0.5788 | 0.6195 | 0.6571 | **0.7083** | 0.7056 |
| | 30% | PSNR | 11.29 | 20.64 | 15.83 | 26.59 | 27.42 | 27.23 | 27.88 | **30.46** |
| | | SSIM | 0.1142 | 0.4391 | 0.2447 | 0.5918 | 0.6392 | 0.6639 | **0.7185** | 0.7039 |

## E.13. Extended Experiments to Clustering

To examine whether the proposed $\ell_p(S_q)$ prior remains effective beyond completion tasks, we further apply it to *unsupervised image clustering*. In this setting, the goal is to recover the underlying identity or digit category by exploiting structural correlations across samples. Following the self-representation paradigm widely used in subspace clustering, each image is expressed as a combination of the remaining images, while the coefficient tensor is regularized through the bidirectional t-linear structure imposed by BTR (Wang et al., 2025a). Embedding $\ell_p(S_q)$ into this framework yields a nonconvex spectral shrinkage along both t-modes, producing a more discriminative affinity matrix for clustering.

*Datasets.* We use four benchmark datasets with varying numbers of classes and resolutions:

- *FRDUE*: 3040 facial images from 152 individuals, each of size $25 \times 22$.

- *FRDUE100*: A subset of FRDUE with 2500 images from 100 subjects, same resolution.

- *PIE10*: 680 images of 10 people under varying lighting/pose, size $22 \times 22$.

- *USPS1000*: 1000 handwritten digits from USPS dataset, 10 classes, $16 \times 16$ resolution.

*Data Tensor Formation.* All images are normalized to $[0, 1]$ and stacked into a third-order tensor $\underline{\mathbf{X}} \in \mathbb{R}^{h \times n \times w}$, where

each lateral slice $\underline{\mathbf{X}}_{:,i,:}$ corresponds to one image. No pretrained features or handcrafted descriptors are used, allowing the clustering performance to reflect purely the modeling ability of the tensor regularizers.

*Table 18.* Clustering performance comparison on four benchmark datasets (*FRDUE*, *FRDUE-100*, *PIE-10*, and *USPS1000*) in terms of accuracy (ACC), normalized mutual information (NMI), and purity (PUR).

| Dataset | Metric | R-TPCA | | OR-TPCA | R-TLRR | | OR-TLRR | | BTR | | Bi-$\ell_p(S_q)$ | |
| --- | --- | --- | --- | --- | --- | --- | --- | --- | --- | --- | --- | --- |
| | | DFT | DCT | DFT | DFT | DCT | DFT | DCT | DFT | DCT | DFT | DCT |
| *FRDUE* | ACC | 0.761 | 0.778 | 0.764 | 0.843 | 0.840 | 0.836 | 0.739 | 0.859 | 0.859 | **0.862** | 0.856 |
| | NMI | 0.909 | 0.914 | 0.913 | 0.951 | 0.951 | 0.947 | 0.904 | 0.958 | 0.957 | **0.960** | 0.956 |
| | PUR | 0.796 | 0.808 | 0.799 | 0.876 | 0.872 | 0.864 | 0.778 | 0.890 | 0.889 | **0.892** | 0.887 |
| *FRDUE-100* | ACC | 0.791 | 0.791 | 0.797 | 0.861 | 0.860 | 0.866 | 0.762 | 0.886 | 0.877 | **0.896** | 0.885 |
| | NMI | 0.913 | 0.914 | 0.922 | 0.953 | 0.953 | 0.956 | 0.906 | 0.964 | 0.961 | **0.966** | 0.963 |
| | PUR | 0.820 | 0.820 | 0.827 | 0.888 | 0.888 | 0.893 | 0.796 | 0.913 | 0.903 | **0.920** | 0.909 |
| *PIE-10* | ACC | 0.428 | 0.427 | 0.540 | 0.590 | 0.583 | 0.459 | 0.198 | 0.600 | 0.611 | 0.603 | **0.638** |
| | NMI | 0.667 | 0.662 | 0.736 | 0.756 | 0.762 | 0.705 | 0.515 | 0.770 | 0.778 | 0.771 | **0.781** |
| | PUR | 0.447 | 0.446 | 0.559 | 0.606 | 0.600 | 0.480 | 0.205 | 0.619 | 0.633 | 0.622 | **0.656** |
| *USPS1000* | ACC | 0.355 | 0.354 | 0.337 | 0.426 | 0.409 | 0.519 | 0.450 | 0.580 | 0.534 | **0.589** | 0.524 |
| | NMI | 0.307 | 0.299 | 0.283 | 0.383 | 0.392 | 0.510 | 0.449 | 0.586 | 0.539 | **0.591** | 0.534 |
| | PUR | 0.447 | 0.454 | 0.443 | 0.526 | 0.508 | 0.633 | 0.591 | 0.686 | 0.666 | **0.687** | 0.665 |

*Model Configuration.* The dictionary is fixed as $\underline{\mathbf{D}} = \underline{\mathbf{X}}$, enforcing a self-expressive relation. BTR regularization weight $\alpha$ is tuned over $\{0.1, 0.5, 0.9\}$. The regularization parameter $\lambda$ is tuned over $\{0.1, 0.5, 1, 2, 5\}$ based on validation. We set $(p, q) = (0.8, 0.81)$. ADMM is used to solve the optimization problem.

*Clustering Protocol.* After optimization, the coefficient tensors from the two t-modules are symmetrized and aggregated to form an affinity matrix $\hat{\mathbf{Z}}$. Spectral clustering with normalized cuts is then applied to obtain cluster assignments. Accuracy (ACC), normalized mutual information (NMI), and purity (PUR) are reported. Each experiment is repeated ten times to mitigate the effect of random initialization. The results, summarized in Table 18, demonstrate that incorporating the nonconvex $\ell_p(S_q)$ regularizer into BTR consistently enhances the discriminability of the learned affinities across all datasets.

### E.14. Sensitivity Analysis of the Shape Parameters $(p, q)$

This subsection reports an extended empirical sensitivity analysis of the proposed $\ell_p(S_q)$ regularizer with respect to the shape parameters $(p, q)$, which control the geometry of the induced dual spectral regularization. Rather than aiming to identify an optimal choice, the analysis is intended to examine whether the empirical behavior of the method remains stable across a range of parameter values and experimental settings.

**Why is selecting $(p, q)$ intrinsically challenging?**   A natural question is how the parameters $(p, q)$ should be chosen in the proposed $\ell_p(S_q)$ regularizer. Unlike conventional scalar hyperparameters, $(p, q)$ do not act as simple weighting coefficients. Instead, they jointly determine the *inductive geometry* of the model: the parameter $p$ controls sparsity across frequency slices, while $q$ governs low-rankness within each slice. As such, $(p, q)$ define the structural bias of the regularizer itself and cannot be absorbed into optimization scaling or adjusted post hoc, playing a role closer to model specification than standard regularization tuning.

From a theoretical standpoint, this difficulty is intrinsic rather than algorithm-specific. Even in vector and matrix settings, nonconvex quasi-norms such as $\ell_p$ penalties are known to lack provably consistent or statistically efficient parameter selection rules under limited data (Xu, 2010). In the tensor setting considered here, the coupling between inter-frequency sparsity and intra-frequency low-rankness further complicates identifiability, and current theory offers little actionable guidance for choosing $(p, q)$. These observations motivate a systematic empirical investigation into whether stable and transferable parameter regimes emerge across different data modalities.

**Experimental design.**   To study the sensitivity of $\ell_p(S_q)$ to $(p, q)$, we evaluate three tensor modalities with distinct spectral characteristics: hyperspectral data (*SalinasA*), video sequences (*Grandma*), and seismic volumes (*Seismic*). For each dataset we form tensors using the first 50, 80, and 120 frontal slices. Increasing the number of slices introduces more frequency components and raises spectral complexity, which helps us evaluate the stability of $\ell_p(S_q)$ under varying tensor sizes.

For each tensor instance we sweep a two-dimensional parameter grid with $p \in \{0.5, 0.6, 0.7, 0.8, 0.9\}$ and $q = p + \Delta q$, where $\Delta q$ spans eight decreasing offsets down to $0$. All configurations are evaluated with a fixed sampling ratio of ten percent and a noise level of five percent. Reconstruction quality is measured by PSNR and SSIM.

*Table 19.* Sensitivity analysis of $\ell_p(S_q)$ on the *SalinasA* dataset under different frequency counts (i.e., using the first 50, 80, and 120 frames). Each sub-table corresponds to a fixed frequency count, with rows indexed by $p$ and columns indexed by $q - p$. Each entry reports PSNR/SSIM.

| # Freq. | $p$ | $q - p = 0$ | 0.0012 | 0.0025 | 0.005 | 0.01 | 0.015 | 0.02 | 0.025 | 0.03 |
|---|---|---|---|---|---|---|---|---|---|---|
| | **0.5** | 30.81/0.8128 | 30.89/0.8128 | 30.98/0.8130 | 30.99/0.8135 | 31.13/0.8138 | 31.13/0.8144 | 31.11/0.8174 | 31.02/0.8186 | 30.83/0.8165 |
| | **0.6** | 31.14/0.8235 | 31.14/0.8235 | 31.18/0.8243 | 31.17/0.8234 | 31.15/0.8212 | 31.25/0.8238 | 30.95/0.8125 | 31.11/0.8136 | 30.64/0.7894 |
| **50** | **0.7** | 31.20/0.8270 | 31.23/0.8275 | 31.22/0.8267 | 31.21/0.8256 | 31.32/0.8275 | 31.42/0.8281 | 31.35/0.8224 | 31.26/0.8159 | 30.51/0.7783 |
| | **0.8** | 31.42/0.8335 | 31.43/0.8337 | 31.48/0.8343 | 31.54/0.8348 | 31.52/0.8315 | 31.43/0.8273 | 31.35/0.8200 | 31.10/0.8052 | 30.54/0.7767 |
| | **0.9** | 31.56/0.8366 | 31.59/0.8378 | 31.57/0.8362 | 31.60/0.8367 | 31.57/0.8322 | 31.45/0.8248 | 31.30/0.8161 | 31.10/0.8055 | 30.95/0.7971 |
| | **0.5** | 31.62/0.8401 | 31.65/0.8412 | 31.68/0.8420 | 31.75/0.8409 | 31.62/0.8344 | 31.67/0.8360 | 31.78/0.8404 | 31.68/0.8428 | 31.53/0.8373 |
| | **0.6** | 31.87/0.8511 | 31.87/0.8499 | 31.90/0.8518 | 31.98/0.8528 | 32.01/0.8510 | 32.00/0.8513 | 31.85/0.8452 | 31.83/0.8400 | 31.36/0.8248 |
| **80** | **0.7** | 31.97/0.8545 | 32.05/0.8558 | 32.05/0.8553 | 32.06/0.8544 | 32.11/0.8541 | 32.14/0.8526 | 32.13/0.8488 | 32.02/0.8422 | 30.99/0.8012 |
| | **0.8** | 32.25/0.8614 | 32.26/0.8610 | 32.25/0.8603 | 32.26/0.8598 | 32.29/0.8569 | 32.20/0.8512 | 32.02/0.8419 | 31.58/0.8230 | 30.91/0.7950 |
| | **0.9** | 32.40/0.8636 | 32.41/0.8630 | 32.41/0.8625 | 32.38/0.8607 | 32.32/0.8559 | 32.18/0.8481 | 31.93/0.8379 | 31.72/0.8295 | 31.55/0.8227 |
| | **0.5** | 33.19/0.8730 | 33.26/0.8740 | 33.34/0.8756 | 33.44/0.8761 | 33.63/0.8785 | 33.70/0.8795 | 33.70/0.8804 | 33.52/0.8795 | 33.20/0.8745 |
| | **0.6** | 33.65/0.8840 | 33.62/0.8835 | 33.68/0.8839 | 33.75/0.8854 | 33.78/0.8857 | 33.84/0.8858 | 33.83/0.8853 | 33.61/0.8772 | 33.79/0.8796 |
| **120** | **0.7** | 33.82/0.8884 | 33.83/0.8884 | 33.86/0.8886 | 33.92/0.8891 | 34.00/0.8894 | 34.13/0.8908 | 34.20/0.8905 | 34.21/0.8889 | 33.85/0.8768 |
| | **0.8** | 34.03/0.8928 | 34.07/0.8931 | 34.08/0.8933 | 34.13/0.8934 | 34.19/0.8932 | 34.22/0.8914 | 34.23/0.8892 | 34.13/0.8843 | 33.49/0.8641 |
| | **0.9** | 34.29/0.8976 | 34.32/0.8975 | 34.34/0.8975 | 34.37/0.8968 | 34.36/0.8949 | 34.34/0.8930 | 34.26/0.8899 | 34.21/0.8878 | 34.11/0.8842 |

**Observed trends across datasets.** The quantitative results in Tables 19 -21 reveal consistent trends across all datasets and frequency settings.

- *Effect of small $p$:* Settings with $p = 0.5$ consistently yield lower PSNR across all three modalities. This pattern is visible in the *SalinasA*, *Grandma*, and *Seismic* results. A likely explanation is that very small $p$ imposes overly aggressive inter-frequency shrinkage, which suppresses informative frequency components and leads to under-representation of useful structure.

- *Effect of large $q$ (large $q - p$):* Increasing $q$ beyond a moderate range yields diminishing or slightly reduced SSIM. This behavior appears in the 80-frame *Grandma* results and similarly in the *SalinasA* and *Seismic* experiments, suggesting that excessive intra-slice shrinkage suppresses structural detail.

- *High-performing region:* The highest and most stable PSNR and SSIM values occur when $p$ lies between $0.7$ and $0.9$ and $q$ slightly exceeds $p$. This pattern is consistent across all datasets. For instance, in the 120-frame *Seismic* results the best PSNR appears near $(p, q - p) \approx (0.8, 0.01)$, with similar behavior observed in *SalinasA* and *Grandma*.

Overall, the tables show that neighboring $(p, q)$ configurations often achieve comparable reconstruction quality, and that this phenomenon appears consistently across modalities and frequency numbers. The performance landscape of $\ell_p(S_q)$ is therefore smooth rather than sharply peaked in the tested range, indicating that the method does not depend on fine-grained parameter tuning.

The heatmaps in Figures 8-10 further support these observations. In the low-frequency-number regime (50 slices), the PSNR surface shows a clear ridge around $\Delta q$ between $0.01$ and $0.015$, with a visible peak near $(p, q) \approx (0.8, 0.81)$. As the number of slices increases to 80 and 120, the sensitivity decreases and the high-performing region expands, indicating that richer spectral information stabilizes the behavior of the regularizer.

**Conclusions based on Empirical Evidence.** The collective evidence from Tables 19 -21 and Figures 8-10 suggests that $(p, q)$ does not require fine tuning. A broad region yields near-optimal results, and the high-performing band around $(p, q) \approx (0.8, 0.81)$ appears consistently across modalities and tensor depths. More importantly, the trends observed in the hyperspectral, video, and seismic experiments agree strongly, implying that the geometric effect of the $\ell_p(S_q)$ prior interacts smoothly with spectral complexity.

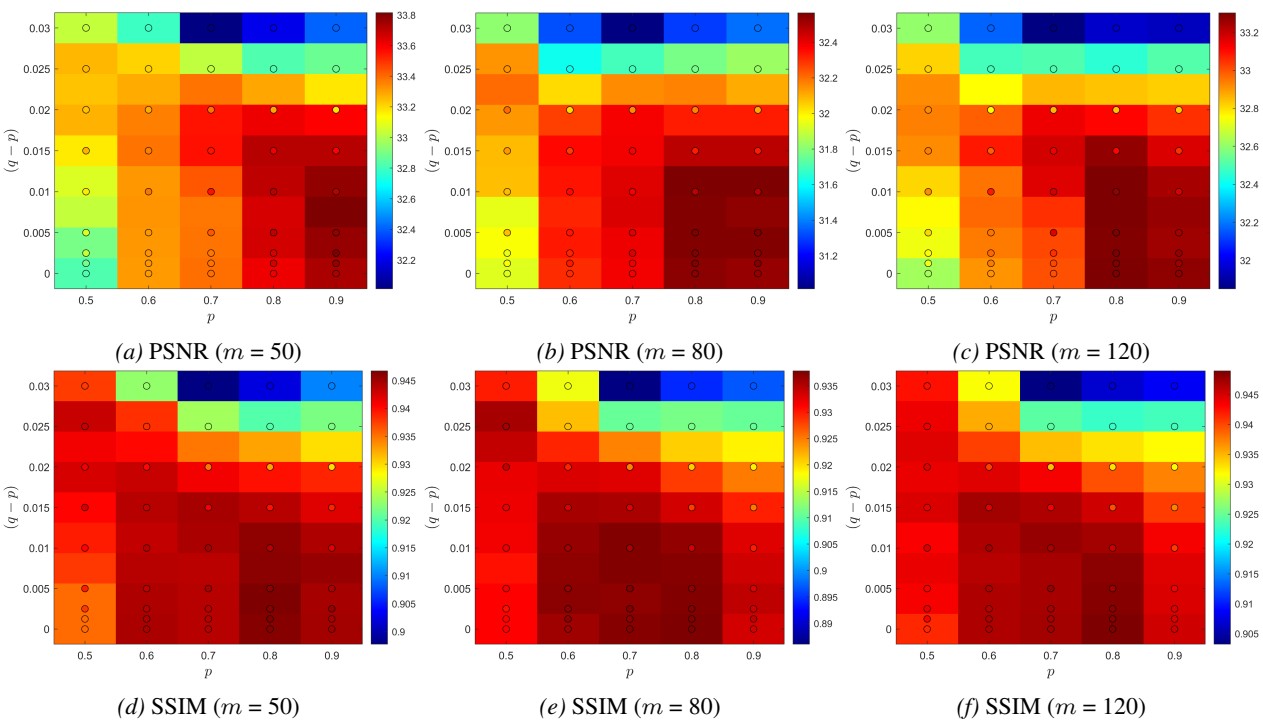

*Figure 8.* Heatmap visualization of PSNR and SSIM scores for $\ell_p(S_q)$ under varying $(p, q-p)$ configurations. Rows correspond to dataset *Grandma* with metrics (PSNR, SSIM); columns represent frequency counts (50, 80, 120 frames).

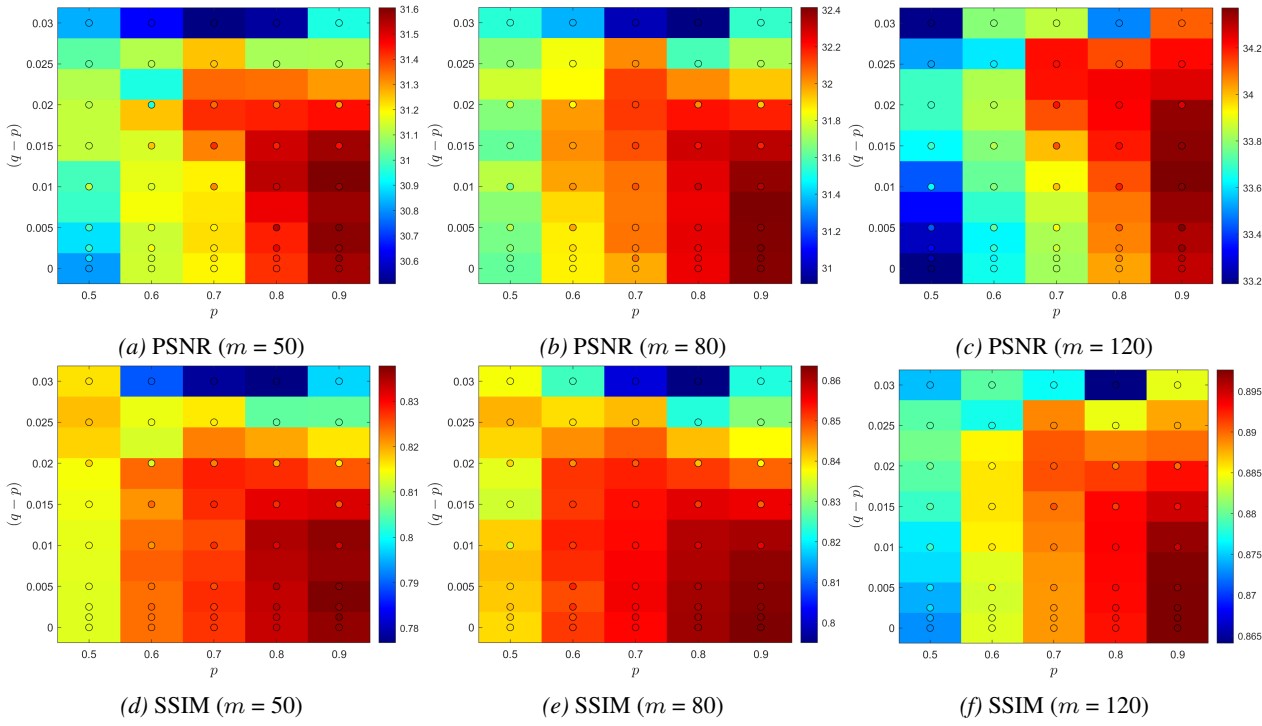

*Figure 9.* Heatmap visualization of PSNR and SSIM scores for $\ell_p(S_q)$ under varying $(p, q-p)$ configurations. Rows correspond to dataset *SanilasA* with metrics (PSNR, SSIM); columns represent frequency counts (50, 80, 120 frames).

*Table 20.* Sensitivity analysis of $\ell_p(S_q)$ on the *grandma* dataset under different frequency counts (i.e., using the first 50, 80, and 120 frames). Each sub-table corresponds to a fixed frequency count, with rows indexed by $p$ and columns indexed by $q - p$. Each entry reports PSNR/SSIM.

| # Freq. | $p$ | $q - p = 0$ | 0.0012 | 0.0025 | 0.005 | 0.01 | 0.015 | 0.02 | 0.025 | 0.03 |
|---|---|---|---|---|---|---|---|---|---|---|
| | **0.5** | 32.83/0.9353 | 32.90/0.9354 | 33.03/0.9378 | 33.07/0.9391 | 33.17/0.9404 | 33.27/0.9425 | 33.24/0.9411 | 33.26/0.9433 | 33.03/0.9377 |
| | **0.6** | 33.31/0.9446 | 33.32/0.9444 | 33.33/0.9440 | 33.32/0.9434 | 33.38/0.9439 | 33.36/0.9432 | 33.28/0.9401 | 33.22/0.9382 | 32.79/0.9231 |
| **50** | **0.7** | 33.39/0.9440 | 33.38/0.9441 | 33.37/0.9438 | 33.43/0.9446 | 33.56/0.9448 | 33.55/0.9412 | 33.38/0.9348 | 33.02/0.9239 | 32.01/0.8977 |
| | **0.8** | 33.61/0.9463 | 33.66/0.9468 | 33.65/0.9461 | 33.70/0.9458 | 33.72/0.9440 | 33.62/0.9398 | 33.29/0.9327 | 32.83/0.9201 | 32.20/0.9022 |
| | **0.9** | 33.74/0.9449 | 33.77/0.9447 | 33.82/0.9455 | 33.77/0.9443 | 33.71/0.9420 | 33.58/0.9387 | 33.18/0.9297 | 32.87/0.9219 | 32.41/0.9101 |
| | **0.5** | 31.93/0.9313 | 31.98/0.9312 | 31.95/0.9304 | 32.07/0.9318 | 32.08/0.9320 | 32.14/0.9322 | 32.21/0.9344 | 32.14/0.9358 | 31.81/0.9298 |
| | **0.6** | 32.30/0.9362 | 32.33/0.9372 | 32.31/0.9369 | 32.34/0.9366 | 32.36/0.9356 | 32.27/0.9331 | 32.04/0.9295 | 31.62/0.9216 | 31.33/0.9176 |
| **80** | **0.7** | 32.39/0.9373 | 32.39/0.9369 | 32.42/0.9376 | 32.42/0.9375 | 32.39/0.9355 | 32.38/0.9329 | 32.16/0.9246 | 31.70/0.9105 | 31.01/0.8860 |
| | **0.8** | 32.55/0.9378 | 32.56/0.9375 | 32.55/0.9374 | 32.57/0.9367 | 32.47/0.9334 | 32.33/0.9282 | 32.17/0.9208 | 31.77/0.9115 | 31.29/0.8944 |
| | **0.9** | 32.53/0.9336 | 32.56/0.9345 | 32.54/0.9337 | 32.56/0.9329 | 32.47/0.9297 | 32.33/0.9249 | 32.11/0.9189 | 31.83/0.9106 | 31.38/0.8968 |
| | **0.5** | 32.63/0.9414 | 32.73/0.9436 | 32.76/0.9442 | 32.81/0.9434 | 32.92/0.9447 | 32.93/0.9440 | 32.92/0.9446 | 32.82/0.9438 | 32.60/0.9425 |
| | **0.6** | 32.91/0.9468 | 32.94/0.9467 | 32.97/0.9464 | 32.95/0.9467 | 33.08/0.9473 | 32.98/0.9446 | 32.75/0.9402 | 32.49/0.9356 | 32.17/0.9316 |
| **120** | **0.7** | 33.00/0.9473 | 33.02/0.9471 | 33.05/0.9470 | 33.16/0.9478 | 33.18/0.9468 | 33.14/0.9435 | 32.86/0.9346 | 32.51/0.9244 | 31.85/0.9032 |
| | **0.8** | 33.30/0.9490 | 33.29/0.9485 | 33.30/0.9484 | 33.30/0.9473 | 33.27/0.9452 | 33.11/0.9396 | 32.84/0.9330 | 32.45/0.9227 | 31.95/0.9066 |
| | **0.9** | 33.28/0.9453 | 33.26/0.9447 | 33.27/0.9446 | 33.24/0.9435 | 33.17/0.9404 | 33.05/0.9372 | 32.82/0.9319 | 32.51/0.9234 | 31.94/0.9085 |

*Table 21.* Sensitivity analysis of $\ell_p(S_q)$ on the *Seismic* dataset under different frequency counts (i.e., using the first 50, 80, and 120 frames). Each sub-table corresponds to a fixed frequency count, with rows indexed by $p$ and columns indexed by $q - p$. Each entry reports PSNR/SSIM.

| # Freq. | $p$ | $q - p = 0$ | 0.0012 | 0.0025 | 0.005 | 0.01 | 0.015 | 0.02 | 0.025 | 0.03 |
|---|---|---|---|---|---|---|---|---|---|---|
| | **0.5** | 41.83/0.9882 | 41.83/0.9882 | 42.15/0.9901 | 42.15/0.9901 | 42.52/0.9911 | 42.53/0.9911 | 42.53/0.9911 | 42.54/0.9911 | 42.54/0.9911 |
| | **0.6** | 42.51/0.9911 | 43.07/0.9914 | 42.63/0.9905 | 42.90/0.9905 | 42.82/0.9912 | 43.02/0.9912 | 42.82/0.9912 | 43.06/0.9913 | 43.06/0.9912 |
| **50** | **0.7** | 42.55/0.9911 | 42.17/0.9901 | 42.17/0.9901 | 42.17/0.9901 | 42.17/0.9901 | 42.17/0.9901 | 42.18/0.9901 | 42.72/0.9914 | 43.44/0.9923 |
| | **0.8** | 43.90/0.9920 | 43.90/0.9919 | 43.91/0.9919 | 44.10/0.9925 | 44.12/0.9925 | 44.12/0.9925 | 44.12/0.9924 | 44.33/0.9926 | 44.33/0.9926 |
| | **0.9** | 43.90/0.9919 | 43.91/0.9919 | 43.92/0.9919 | 44.08/0.9924 | 44.11/0.9924 | 44.14/0.9924 | 44.16/0.9925 | 44.15/0.9924 | 44.20/0.9924 |
| | **0.5** | 42.21/0.9864 | 42.22/0.9864 | 42.22/0.9864 | 42.97/0.9886 | 43.14/0.9898 | 43.15/0.9898 | 43.15/0.9898 | 43.16/0.9897 | 43.16/0.9897 |
| | **0.6** | 43.40/0.9897 | 43.41/0.9897 | 43.40/0.9897 | 43.40/0.9897 | 43.68/0.9906 | 43.58/0.9904 | 43.41/0.9896 | 43.41/0.9895 | 43.58/0.9903 |
| **80** | **0.7** | 42.98/0.9886 | 42.99/0.9885 | 42.99/0.9885 | 42.99/0.9885 | 43.16/0.9897 | 43.16/0.9897 | 43.56/0.9904 | 43.40/0.9896 | 44.02/0.9907 |
| | **0.8** | 43.82/0.9905 | 43.92/0.9909 | 43.94/0.9909 | 44.17/0.9913 | 44.42/0.9913 | 44.60/0.9916 | 44.61/0.9916 | 44.61/0.9916 | 44.73/0.9915 |
| | **0.9** | 43.66/0.9905 | 43.84/0.9905 | 43.94/0.9909 | 44.04/0.9911 | 44.40/0.9914 | 44.52/0.9916 | 44.59/0.9916 | 44.61/0.9915 | 44.67/0.9916 |
| | **0.5** | 42.33/0.9852 | 42.65/0.9862 | 42.65/0.9863 | 42.65/0.9863 | 43.11/0.9874 | 43.23/0.9877 | 43.24/0.9876 | 43.08/0.9871 | 43.08/0.9871 |
| | **0.6** | 43.38/0.9876 | 43.29/0.9871 | 43.23/0.9871 | 43.76/0.9885 | 43.63/0.9877 | 43.47/0.9873 | 43.61/0.9877 | 43.65/0.9881 | 43.55/0.9874 |
| **120** | **0.7** | 43.40/0.9873 | 43.29/0.9870 | 43.29/0.9870 | 43.29/0.9870 | 43.29/0.9869 | 43.43/0.9872 | 43.43/0.9872 | 43.71/0.9877 | 44.56/0.9899 |
| | **0.8** | 44.24/0.9890 | 44.33/0.9891 | 44.34/0.9891 | 44.47/0.9893 | 44.61/0.9895 | 44.84/0.9902 | 45.02/0.9904 | 45.04/0.9904 | 45.08/0.9902 |
| | **0.9** | 44.23/0.9891 | 44.24/0.9890 | 44.30/0.9891 | 44.46/0.9894 | 44.60/0.9899 | 44.73/0.9902 | 44.86/0.9902 | 45.03/0.9903 | 45.02/0.9902 |

**Empirical Recommendation of** $(p, q)$**.** Based on the overall trends observed in Tables 19–21 and the heatmaps in Figures 9–10, a practical choice in many cases is

$$p \in [0.8, 0.9], \qquad q - p \in [0.005, 0.015].$$

These ranges provide stable reconstruction quality across the tested datasets, and may serve as a useful starting point when no task-specific tuning is available.

*Remark* E.12. While adaptive or bilevel parameter selection strategies may further improve performance, they remain computationally expensive and statistically unstable under nonconvex objectives and limited data. Developing scalable and principled methods for quasi-norm parameter selection is an important open direction, but is beyond the scope of this work, which focuses on the formulation, theoretical properties, and empirical benefits of the proposed $\ell_p(S_q)$ regularizer.

### E.15. Sensitivity Analysis of the Regularization Parameter $\lambda$

The regularization parameter $\lambda$ controls the trade-off between data fidelity and the proposed $\ell_p(S_q)$ regularization. A very small $\lambda$ may lead to insufficient spectral shrinkage and retain noise or sampling artifacts, whereas an overly large $\lambda$ may

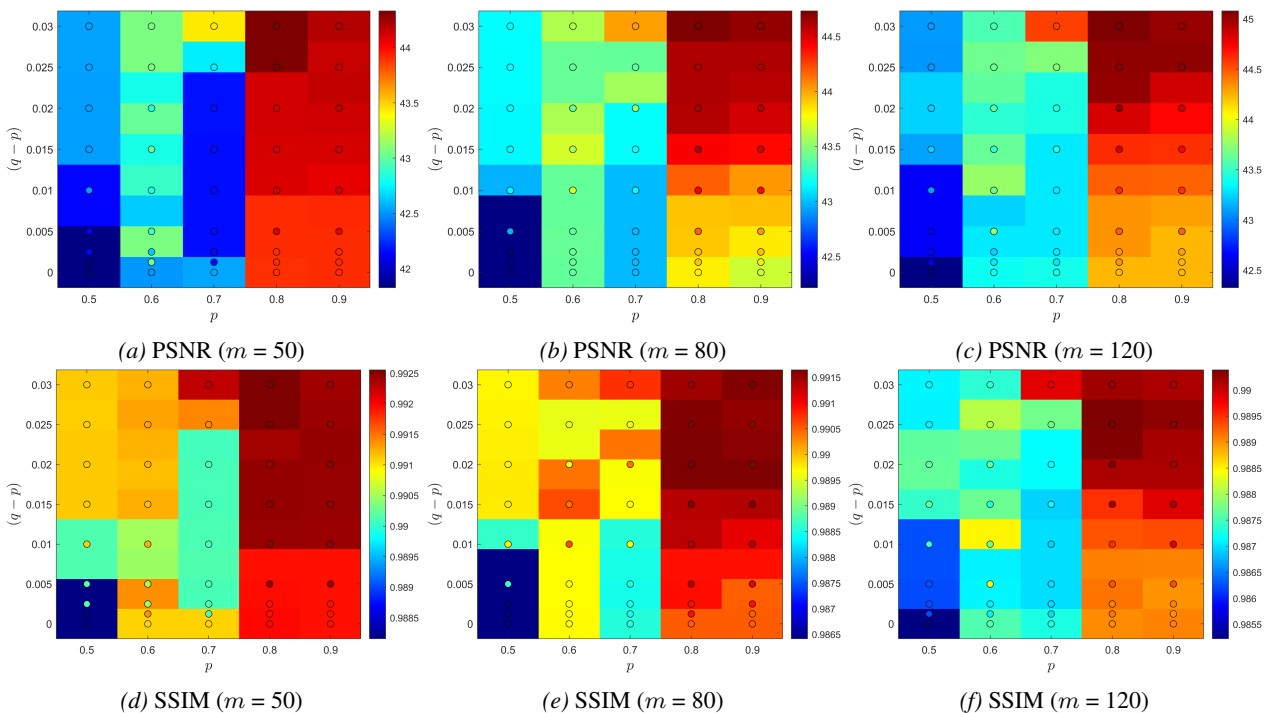

*(a)* PSNR ($m = 50$)   *(b)* PSNR ($m = 80$)   *(c)* PSNR ($m = 120$)

*(d)* SSIM ($m = 50$)   *(e)* SSIM ($m = 80$)   *(f)* SSIM ($m = 120$)

*Figure 10.* Heatmap visualization of PSNR and SSIM scores for $\ell_p(S_q)$ under varying $(p, q - p)$ configurations. Rows correspond to dataset *Seismic* with metrics (PSNR, SSIM); columns represent frequency counts (50, 80, 120 frames).

over-penalize transformed singular values and remove useful signal components. Hence, we examine whether the proposed method is overly sensitive to the choice of $\lambda$.

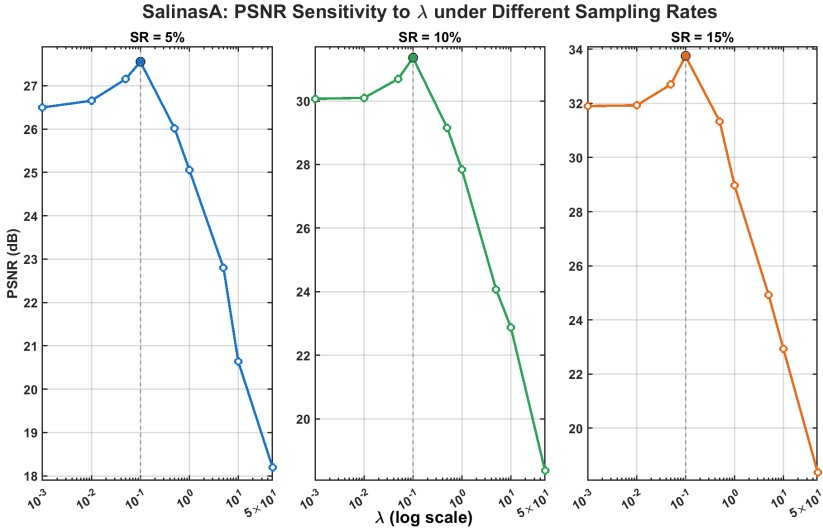

*Figure 11.* Preliminary sensitivity of the proposed model to the regularization parameter $\lambda$ on Salinas A under three sampling ratios ($\text{SR} = 5\%, 10\%, 15\%$), with $(p, q) = (0.80, 0.81)$ fixed. The horizontal axis uses a logarithmic scale for $\lambda$, and the vertical axis reports PSNR. Each panel corresponds to one sampling ratio and illustrates how reconstruction quality varies as $\lambda$ changes.

We first conduct a one-dimensional sensitivity study on the Salinas A dataset under three sampling ratios, $\text{SR} = 5\%, 10\%$, and $15\%$. In this experiment, we fix $(p, q) = (0.80, 0.81)$ and vary $\lambda$ over a logarithmic grid. The reconstruction quality is evaluated by PSNR. As shown in Figure 11, the PSNR curves exhibit a similar trend across the three sampling ratios. When

$\lambda$ is too small, the regularization effect is insufficient; when $\lambda$ is too large, excessive shrinkage deteriorates reconstruction quality. In all three settings, the best or near-best performance is achieved around $\lambda = 0.1$, suggesting that a moderate regularization strength provides a favorable balance between preserving signal components and suppressing noise.

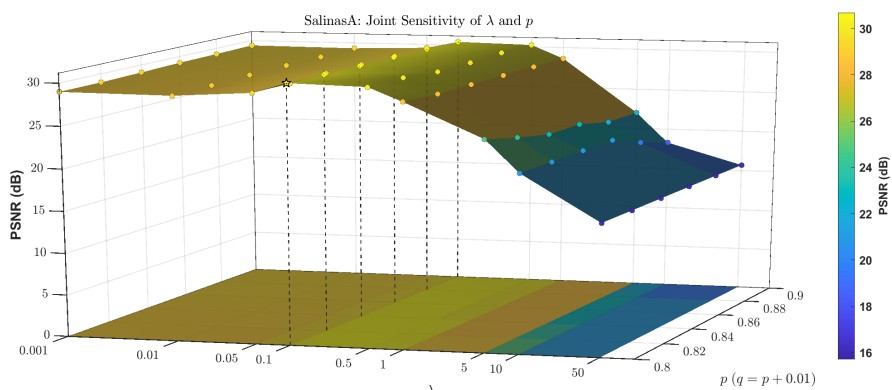

*Figure 12.* Preliminary joint sensitivity of the proposed algorithm to $\lambda$ and $p$ on Salinas A under SR $= 10\%$ and NSR $= 5\%$, with $q = p + 0.01$. The tested values of $p$ are chosen within the empirical range discussed in Appendix E.14. The horizontal axis shows $\lambda$, the second axis shows $p$, and the vertical axis reports PSNR. The plot provides a controlled view of the interplay between the overall regularization strength and the $\ell_p(S_q)$ shape parameter.

We further examine the joint sensitivity of $\lambda$ and $p$ under SR $= 10\%$ and NSR $= 5\%$, with $q = p + 0.01$. The tested values of $p$ are chosen within the empirically recommended range discussed in Appendix E.14. As shown in Figure 12, the favorable region of $\lambda$ remains centered around $\lambda = 0.1$ across different values of $p$. This indicates that the choice of $\lambda$ is not strongly coupled with small variations of $(p, q)$ within the tested range.

Overall, these results suggest that $\lambda$ mainly controls the overall strength of spectral shrinkage, while the qualitative behavior of the method remains stable within a moderate range around $\lambda = 0.1$. Accordingly, we use $\lambda = 0.1$ as the default value in the reported experiments unless otherwise specified.

# F. Discussion of Scope and Future Directions

This work introduces the $\ell_p$-Schatten-$q$ quasi-norm as a unified regularization framework for modeling dual spectral sparsity in tensor data, together with its statistical characterization and an implementable solver. Complementing the limitations discussed in the concluding remarks, we further clarify the scope of the present study and outline several natural directions for future development.

**Parameter selection.**    The parameters $(p, q)$ determine the geometry and inductive bias of the proposed regularizer. While no statistically optimal selection theory is currently available, even in simpler settings, we provide practical recommendations for choosing $(p, q)$ and demonstrate through experiments that the performance of $\ell_p(S_q)$ is robust across different datasets and sampling regimes. Developing adaptive or data-driven selection mechanisms, for instance via bilevel optimization, is an important but technically challenging direction that lies beyond the present scope.

**Theoretical scope.**    Our minimax analysis is conducted under a Gaussian location model, which serves as a canonical setting to isolate the intrinsic statistical effect of dual spectral sparsity. This theory characterizes how estimation difficulty depends jointly on inter-frequency sparsity and intra-frequency spectral decay. Extending equally sharp guarantees to task-specific inverse problems, such as tensor completion or Poisson observations, would require incorporating the corresponding sampling operators and likelihood models into the analysis.

**Computational aspects.**    The current implementation uses full SVDs for all transformed slices, which is transparent but not yet optimized for large-scale tensors. A more scalable implementation could exploit several forms of structure. First, since the reweighted shrinkage sets many singular values to zero, each slice update can be computed by truncated SVD, retaining only components above the current adaptive threshold. Second, the active frequency set usually changes slowly across iterations, so low-energy slices can be screened or updated less frequently. Third, consecutive ADMM iterates have similar spectral subspaces, making warm-started Lanczos iterations or subspace recycling natural choices. Fourth, the transformed slice updates are embarrassingly parallel. Finally, randomized SVD and stochastic frequency-slice sampling could further reduce the per-iteration cost, especially when the number of transformed slices is large. These strategies could lead to a more scalable solver while preserving the dual spectral bias.

**Potential applications.**    Beyond classical tensor recovery, dual spectral regularization may be useful in high-dimensional learning problems such as LLM KV-cache compression (Li et al., 2025), parameter-efficient adaptation of foundation models (Tao et al., 2025), and multi-view clustering (Huang et al., 2025). These settings may exhibit nonuniform importance across heads, layers, modules, or views, together with low-rank redundancy within selected components. An $\ell_p(S_q)$-type regularizer could therefore provide a way to combine structural sparsification with within-group low-rank compression. Since the appropriate transform may not be fixed in advance, a natural future direction is to jointly learn the transform domain and the dual spectral regularization structure.

In summary, the proposed $\ell_p(S_q)$ quasi-norm opens a new way to model dual spectral sparsity, but **its statistical, geometric, and algorithmic properties are still only partially understood**. We view the present work as a starting point for further exploration of dual spectral regularization in broader statistical and computational settings.

