# OpenReview forum: "Refining Dual Spectral Sparsity in Transformed Tensor Singular Values"
_ICML.cc/2026/Conference — ICML 2026 regular_

### Official Review · Reviewer_dfFo · 2026-03-02

**Soundness:** 3
**Presentation:** 3
**Significance:** 3
**Originality:** 3
**Overall Recommendation:** 5
**Confidence:** 3

**Summary:**

This paper addresses the limitations of the Tensor Nuclear Norm (TNN), which assumes single-level spectral sparsity and fails to capture the complex multi-level structures in real-world data. The authors propose the tensor $l_p$-Schatten-q ($l_p(S_q)$) quasi-norm to model "dual spectral sparsity"—simultaneously capturing inter-frequency selection and intra-frequency low-rankness. The work provides a comprehensive theoretical framework, including minimax error bounds and a unified view of existing tensor norms. A reweighted proximal algorithm is developed and validated through experiments on tensor completion and clustering tasks, demonstrating superior performance over state-of-the-art methods.

**Compliance With Llm Reviewing Policy:**

Affirmed.

**Final Justification:**

The author resolved most of my issues, so I've upgraded my score to 5.

**Key Questions For Authors:**

See weakness

**Limitations:**

yes

**Strengths And Weaknesses:**

Strengths:

1.The problem studied by this paper is the misalignment between traditional TNN and the heterogeneous spectral distributions of real data. By decoupling the frequency-domain sparsity into two levels, the model offers a much more refined representation of tensor structures.

2.The paper establishes both upper and lower minimax error bounds, providing solid statistical guarantees for the proposed dual-layer sparsity model. The generalization of several existing norms into a single framework is also a significant contribution.

3.This paper is clearly written and well structured. The author's analysis, proof, and experiments on this issue are comprehensive and thorough.

Weaknesses:
1.The introduction of two parameters ($p$ and $q$) increases the model's flexibility but also its tuning complexity. Although the author discusses its shortcomings, I believe these two parameters will hinder the function's practicality. Moreover, given that this framework is computationally intensive by nature, it raises the question of whether methods like TNN remain more practical in real-world applications.

2.In comparison, I directly employed weighted TNN to constrain dual spectral sparsity—could this achieve the same effect? Essentially, it boils down to finding a suitable weighting scheme to measure the contributions of different frequencies and singular values.

3.Does it remain effective for non-image and non-video data? Currently, low-rank tensor recovery is extensively used in multi-view clustering, but its samples lack continuous properties. Can this method also be applied to this field?

4.Since the energy in high-frequency slices is negligible, can I achieve faster constraints by directly truncating the frequency slices?

---

> ### Author Rebuttal · Authors · 2026-03-30
>
> Thank you for your recognition of our refined representation and statistical guarantees. We address your specific concerns below.
>
> ---
>
> > **W1: Tuning Complexity & Practicality vs. TNN**
>
> We agree that $p$ and $q$ increase flexibility and tuning complexity relative to TNN. This is a practical trade-off: TNN is effective when uniform spectral regularization is adequate, while our model is intended for tensors exhibiting dual spectral sparsity in the transformed domain. Our goal is not to replace TNN universally, but to provide a more expressive alternative for dual-spectral structures.
>
> * **Sensitivity:** As shown in Appendix E.14, performance is not sharply sensitive to a single $(p,q)$ choice; across the recommended range, the method consistently attains competitive performance without requiring fine-grained tuning.
> * **Cost:** Per-iteration cost is $O(d_1 d_2 m \log m + m d_1 d_2 \min(d_1,d_2))$, the *same complexity class* as TNN.
> * **Runtime:** Preliminary tests ([Scalability Figures](https://anonymous.4open.science/r/LpSq-8B38/Scalability.png)) show LpSq incurs a `1.7x–1.9x` runtime overhead but achieves consistent gains of `+1.6` to `+2.8 dB` PSNR, suggesting that the added cost behaves like a stable multiplicative overhead across the tested settings.
>
> > **W2. Comparison with Weighted TNN.**
>
> We agree that weighted TNN is closely related to our method, as both involve a non-uniform treatment of spectral information. In our case, the weighting arises as part of the formulation tailored to the proposed $\ell_p$-Schatten-$q$ regularization.
>
> The main difference lies in how this non-uniformity is defined. In conventional weighted TNN, weights are typically assigned independently to individual singular values, often based on heuristic choices or previous iterates, while different transformed frequency slices are usually treated uniformly. In our formulation, the weighting behavior is instead induced by the underlying $\ell_p$-Schatten-$q$ regularization and reflects the collective spectral structure within each transformed slice.
>
> Therefore, while a carefully designed weighted TNN may exhibit similar effects in certain cases, our method is better understood as a structured dual-spectral modeling framework associated with the proposed LpSq, rather than a direct instance of conventional weighted TNN.
>
> > **W3. Non-image/video data and Multi-view Clustering.**
>
> We agree that multi-view clustering differs from image/video data, since the sample/view dimension often lacks explicit spatial or temporal continuity, making a literal frequency interpretation less direct. Empirically, our experiments already go beyond videos, covering hyperspectral, multispectral, thermal, and seismic data, as well as clustering datasets, suggesting that the method is not tied to a single modality.
>
> From the modeling side, our formulation is defined in the transformed domain and does not fundamentally rely on continuity itself; rather, it depends on whether the chosen transform can reveal nonuniform cross-slice importance together with within-slice low-rank structure. Our clustering experiments, conducted without explicit spatial priors, together with additional results under multiple transforms, including adaptively learned ones, provide preliminary evidence that the method is not tied to a fixed physical frequency interpretation. More broadly, the consistent empirical gains of LpSq over TNN across these diverse settings suggest that the proposed dual-spectral structure may also be useful in the multi-view setting. A more dedicated study of multi-view clustering is beyond the scope of the current paper, but we agree it is an important future direction and will add this discussion to Appendix F.
>
> > **W4. Frequency Truncation for Acceleration.**
>
> We agree that direct truncation of high-frequency slices is a natural and practical way to reduce computational cost. Motivated by this idea, we added a preliminary experiment on SalinasA to examine the resulting speed-accuracy trade-off. See the figures for [Running Time](https://anonymous.4open.science/r/LpSq-8B38/TruncFreq-Time.png) and [PSNR](https://anonymous.4open.science/r/LpSq-8B38/TruncFreq-PSNR.png).
>
>
> | Kept low-freq ratio | Speedup | PSNR gap |
> |---|---:|---:|
> | 50% | 1.73x | -1.05 dB |
> | 70% | 1.38x | -0.53 dB |
> | 90% | 1.09x | -0.17 dB |
>
> These results confirm that hard truncation can accelerate the solver, but with some loss in reconstruction quality. This is consistent with the underlying mechanism: hard truncation assumes some transformed slices can be removed beforehand, whereas our model uses a softer, data-dependent selection. As a result, stronger truncation gives more speedup but may also remove weak yet useful components. We therefore view this as a meaningful acceleration direction, and the results further suggest that adaptive frequency truncation may be an interesting future direction.
>
> ---
>
> Thank you again for these thoughtful suggestions.

---

> > ### Author Rebuttal · Reviewer_dfFo · 2026-04-02
> >
> > Thank you for the authors' reply. I will keep my positive score.

---

> > > ### Author Response · Authors · 2026-04-04
> > >
> > > Thank you for taking the time to review our response. We appreciate your acknowledgment that the concerns have been fully addressed, and we value your continued positive recommendation. We will incorporate these points more explicitly into the final version.

---

### Official Review · Reviewer_Xfh4 · 2026-03-11

**Soundness:** 3
**Presentation:** 1
**Significance:** 3
**Originality:** 3
**Overall Recommendation:** 4
**Confidence:** 3

**Summary:**

The paper introduces the tensor $\ell_p$-Schatten-$q$ quasi-norm as a generalization of the Tensor Nuclear Norm (TNN) to better capture multi-level spectral structures in tensors. The proposed regularizer jointly enforces inter-frequency sparsity and intra-frequency low-rankness, providing a more flexible model than TNN and related methods. The authors establish minimax error bounds and propose a reweighted algorithm to solve the resulting nonconvex optimization problem. Experiments on tensor completion and image clustering demonstrate improved performance and robustness.

**Compliance With Llm Reviewing Policy:**

Affirmed.

**Final Justification:**

I kept my initial score. The paper presented good contributions, and I would like to see their paper appearing in conferences such as ICML

**Key Questions For Authors:**

- Can you explain the algorithm and main related results in the main text?
- Can you show us some denoised images to compare the methods?
- Is definition 3.2 completely novel, or you are inspired by concepts from other contexts?

**Strengths And Weaknesses:**

- I found the paper interesting. Exploring the low-rank properties of tensors is an important topic.
- The theoretical results in Sections 4 and 5 seem interesting.
- The presentation is, however, too implicit in my opinion. I have a hard time understanding Section 2 and need to read the reference (Lu et al. 2019) to understand the concepts. Similar for Section 4.2: I find it difficult to understand Equation (6). Would the reader minding explaining me what $\hat{L}$ is? Several acronyms such as PSNR and SSIM are used before being introduced.
- The algorithm to solve the correponding optimization seems unnatural? Are there any specific reason to approximate with $\ell_{1,2}$ norm?

---

> ### Author Rebuttal · Authors · 2026-03-30
>
> We appreciate the constructive feedback. We will refine the manuscript to be more self-contained.
>
> ---
>
> > **1.  Too dependent on prior references (Sec. 2).**
>
> We will expand Sec. 2 to make the core modeling self-contained, including the transformed frontal slices, the role of tensor singular values under t-SVD, and the interpretation of standard TNN as uniform shrinkage.
>
> > **2. Eq. (6) and the theoretical presentation.**
>
> Eq. (6) defines the *minimax risk*, a standard concept in statistical learning theory used to measure the fundamental estimation limit for a structured tensor class $\mathsf{T}$.
>
> - $\underline{\mathbf{L}}^\star$ denotes the *ground-truth tensor* that we aim to recover.
> - $\underline{\widehat{\mathbf{L}}}$ denotes any possible *estimator*  constructed from the observations.
> - The infimum-supremum ($\inf_{\underline{\widehat{\mathbf{L}}}} \sup_{\underline{\mathbf{L}}^\star \in \mathsf{T}}$) structure represents a "game" where we seek the best possible estimator against the "hardest" (worst-case) tensor within the defined class $\mathsf{T}$. Essentially, Eq. (6) gives the best possible worst-case error over $\mathsf{T}$.
>
> > **3. Acronyms such as PSNR.**
>
> PSNR refers to Peak Signal-to-Noise Ratio, and SSIM refers to Structural Similarity Index; both are currently defined in App. E.1. We will ensure all acronyms are defined upon their first use.
>
> > **4. On the optimization procedure and approximation.**
>
> The proposed $\ell_p$-Schatten-$q$ regularizer is designed to jointly capture two distinct structural properties: *inter-frequency sparsity* and *intra-frequency spectral decay*. Due to this *coupled nonconvex structure*, the objective does not permit a straightforward proximal update like standard TNN formulations.
>
> We introduced this approximation to preserve dual-spectral priors while ensuring a computationally efficient solver. Specifically, the *weighted $\ell_{1/2}$ proxy* admits a *closed-form half-thresholding update* for each singular value, supported by the well-established $\ell_{1/2}$ theory [R1].
>
> To justify this proxy, we also evaluated a weighted $\ell_1$ alternative. As shown in App. E.4, the $\ell_{1/2}$ proxy consistently outperforms the $\ell_1$ version on both SalinasA and Indian Pines across multiple sampling rates. This suggests the non-convex $\ell_{1/2}$ proxy better preserves the inherent spectral decay and low-rank structure of transformed slices. We will incorporate this motivation into the revision.
>
> [R1]  Xu, Z. Data modeling: Visual psychology approach and $L_{1/2}$ regularization theory. In *International Congress of Mathematicians (ICM)*, 2010.
>
> > **5. Explain the algorithm and the main results.**
>
> We provide a more intuitive summary:
>
> - The proposed algorithm is designed for the $\ell_p$-Schatten-$q$ regularized problem, where sparsity across transformed frequency slices and spectral shrinkage within each slice are coupled in one objective. All transformed slices are updated throughout the iterations, and all undergo singular-value shrinkage. Crucially, the effective shrinkage strength is adaptive: slices with weaker spectral contribution are penalized more heavily (driven closer to zero), while those with stronger contribution are penalized less heavily to retain more spectral information.
> - The main theorem establishes minimax rates under both hard dual spectral sparsity and soft $\ell_p(S_q)$ constraints. In the hard case, the risk consists of two parts: locating informative frequency slices and estimating the low-rank structure within them. In the soft case, the rate is jointly governed by the $\ell_p$ frequency effect and the Schatten-$q$ within-slice decay, with different regimes depending on the relationship between $p$ and $q$.
>
> Both the algorithm and the theory share the same principle: estimation difficulty is jointly determined by the concentration of tensor energy across frequency slices and the compressibility of singular values within each slice.
>
> > **6. Visual comparisons.**
>
> We provide representative visual results for the challenging Poisson tensor completion setting on Indian Pines at [Visualization](https://anonymous.4open.science/r/LpSq-8B38/Visualization.png).
>
> > **7. Novelty of Def. 3.2.**
>
> The novelty of Def. 3.2 lies in the *hierarchical coupling* of two structural effects into a single tensor regularizer:
>
> - It uses an *outer $\ell_p$ term* to model sparsity across transformed frequency slices.
> - It uses an *inner Schatten-$q$ term* to model spectral decay within each slice.
>
> This is structurally different from TNN (which applies a uniform penalty across slices) and mixed $\ell_u(\ell_q)$ norms used in vector settings where group structures are fixed. In our case, the "groups" are induced by the tensor transform and represent spectral frequencies. This dual-level coupling is specific to the t-SVD framework and has not been explored in prior work.
>
> ---
>
> We appreciate the opportunity to clarify these points.

---

> > ### Author Rebuttal · Reviewer_Xfh4 · 2026-04-01
> >
> > In my opinion, most problems of this paper reside in its presentation form, as well as the implicitness in certain results. Since the author promised to work on these issues in the final draft, my concerns are addressed, and I will keep my score as it is.

---

> > > ### Author Response · Authors · 2026-04-04
> > >
> > > We sincerely thank you for your constructive follow-up. We are encouraged that the concerns regarding presentation and the implicitness of certain results have been addressed. In the final version, we will make the manuscript more self-contained and present these points more clearly.

---

### Official Review · Reviewer_WS8R · 2026-03-15

**Soundness:** 3
**Presentation:** 3
**Significance:** 3
**Originality:** 3
**Overall Recommendation:** 4
**Confidence:** 4

**Summary:**

The work proposes an extension to the tensor nuclear norm (TNN)-based low-rank modeling on tensors using the concept of dual spectral sparsity. Here, low-rankness within the frequency components co-exists along with sparsity across them, unlike TNN that assumes element-wise sparsity on frequency-domain. Using an l_p Shatten q_quasi norm regularization, the approach designs a formulation that controls inter-frequency sparsity and intra-frequency low-rankness. Theoretical analysis including minimax bounds are established for the proposed optimization problem. Several experiments on real-data are promising.

**Compliance With Llm Reviewing Policy:**

Affirmed.

**Key Questions For Authors:**

Please see “Weaknesses” section. In addition, I have a few other questions.

1.	In Table 1, do you present results without using the new regularization? The baseline names are a bit confusing. Please present an ablation study that shows the improvements from considering only frequency sparsity, only low-rankness, both, and none.

2.	Have you looked at if the theoretical results with minimax rates have correlation with empirical results?

3. It would be better to present some synthetic data results in the main page to understand the framework and the interplay of the hyperparameters better.

**Limitations:**

Please discuss limitations in terms of the generality of the prior assumptions (co-existence of low-rankness and sparsity) for various settings. Also, include discussions on comparing computational complexity with baselines and also any other limitations.

**Strengths And Weaknesses:**

Strengths:

1.	The paper is well-organized and presented. Motivation for the proposed modeling is clearly detailed.

2.	Extensive experiments are provided, especially many real-world datasets are considered for evaluating the proposed approach.

Weaknesses:

1.	Limited novelty in algorithm development which seems like computationally intensive. No comparison is presented about the computational complexity and about scalability in handling large-scale tensor data, compared to the baselines.

2.	Connections to low-rank models such as block-term decomposition are not discussed. In experiments as well, tensor completion algorithms using models such as CP, Tucker, and block-term are not considered. These models also assumes low-rankness and the associated algorithms are the state-of-the-art in tensor completion tasks.

3.	Sensitivity to the regularizer lambda and related analysis is missing. Synthetic simulations may help to understand the trade-off and the interplay between hyperparameters lambda, p, and q across diverse scenarios and noise settings.

---

> ### Author Rebuttal · Authors · 2026-03-30
>
> Thanks for your constructive feedback and positive comments on the motivation and empirical results. Our work develops a dual-spectral modeling framework, its theoretical characterization, and a tailored optimization algorithm for $\ell_p$-Schatten-$q$ regularization, jointly capturing inter-frequency sparsity and intra-frequency low-rankness. We address your concerns regarding algorithmic complexity, scalability, and relations to other tensor models below.
>
> ---
>
> > **W1. On complexity and scalability.**
>
> We agree that computational aspects should be clarified more explicitly.
>
> - **Complexity Analysis:** As analyzed in Appendix E.2.2, the per-iteration cost is dominated by the transform (under DFT or DCT) and slice-wise SVD operations:
>   $O(d_1 d_2 m \log m) + O(m d_1 d_2 \min(d_1, d_2))$.
>   This is of the same order as standard TNN methods. The additional cost mainly comes from the reweighting step, which involves slice-wise arithmetic and does not change the overall complexity order.
> - **Preliminary Scaling Analysis:** To characterize the scaling behavior of our algorithm, we conducted preliminary experiments by increasing tensor dimensions along both spectral depth and spatial resolution (see [Scalability Figures](https://anonymous.4open.science/r/LpSq-8B38/Scalability.png)). These results indicate that as the tensor size grows, the runtime overhead relative to TNN remains *relatively stable* (approx. `1.7x–1.9x`).  This suggests that, at larger scales, our method *scales in a manner consistent with the complexity order of standard TNN*. In these preliminary experiments, this additional constant-factor cost is accompanied by a reconstruction gain of `+1.6` to `+2.8 dB` PSNR.
> - **Practical considerations:** The algorithm follows the same slice-wise structure as TNN, supports parallelization across frequency slices, and is compatible with truncated or randomized SVD for further acceleration as discussed in Appendix F.
>
> > **W2. On relation to CP / Tucker / block-term models.**
>
> We appreciate the suggestion to consider these fundamental tensor models. Following your suggestion, we have added preliminary comparisons with CP, Tucker, and Block-Term Decomposition methods; the results are available at [Experimental Results](https://anonymous.4open.science/r/LpSq-8B38/CTB.png). We will include these comparison results and a discussion on the structural connections between these models in the revised manuscript.
>
> > **W3. On sensitivity to $\lambda$ and hyperparameters.**
>
> To clarify the role of $\lambda$, we conducted a sensitivity analysis on *SalinasA* under three distinct sampling settings (SR = 5%, 10%, 15%) (see [Lambda Results](https://anonymous.4open.science/r/LpSq-8B38/Lam-SR.png)) and a joint sensitivity study over $\lambda$ and $p$, with $q=p+0.01$ (see [Joint Sensitivity](https://anonymous.4open.science/r/LpSq-8B38/Lam-pq.png)). Across these diverse observation and parameter settings, the optimal value of $\lambda$ consistently remains around 0.1. These preliminary results suggest that the performance is relatively robust to the choice of $\lambda$ within a moderate range, which simplifies hyperparameter tuning in practice. We will include a more comprehensive sensitivity analysis in the revised manuscript.
>
> > **Q1. On Table 1 and ablation.**
>
> Thank you for pointing this out. All results in Table 1 are obtained using the corresponding regularized formulations. The potential confusion arises from the baseline naming, which we will revise for clarity. We have also added preliminary ablation results (see [Ablation Results](https://anonymous.4open.science/r/LpSq-8B38/Ablation.png)), demonstrating that combining frequency sparsity and intra-slice low-rankness yields consistently better performance.
>
> > **Q2. On minimax theory vs. empirical behavior.**
>
> Yes. We provide numerical illustrations in Appendix C.5 showing that the recovery error increases with noise level, rank $r$, and sparsity $s$. This trend is consistent with the dependence predicted by the minimax rates, providing qualitative empirical support for the theoretical analysis.
>
> > **Q3. On synthetic experiments.**
>
> We agree that synthetic data helps illustrate the framework. We already include synthetic illustrations in Appendix C.5, showing how recovery error varies with structural parameters. We will further refine the presentation to better highlight these results.
>
> > **Limitations.**
>
> We will clarify the limitations more explicitly in the revision:
>
> - Generality: The dual spectral prior is most effective when spectral energy is concentrated in transformed components; its advantage may diminish if the energy is uniformly distributed across frequencies.
> - Computation: The current solver relies on slice-wise SVDs, which may limit scalability for extremely large tensors without acceleration techniques (e.g., randomized SVD).
>
> A more detailed discussion is provided in Appendix F.
>
> ---
>
> Thank you again for these helpful suggestions.

---

> > ### Author Rebuttal · Reviewer_WS8R · 2026-04-03
> >
> > Thank you for the response. I am keeping my score.

---

> > > ### Author Response · Authors · 2026-04-04
> > >
> > > Many thanks for your thoughtful engagement. We are encouraged by your acknowledgment that our response has fully addressed the concerns. In the final version, we will incorporate these points more explicitly into the manuscript.

---

### Official Review · Reviewer_snXw · 2026-03-20

**Soundness:** 3
**Presentation:** 3
**Significance:** 3
**Originality:** 3
**Overall Recommendation:** 5
**Confidence:** 5

**Summary:**

This paper proposed a novel tensor L_p quasi norm to solve the propolem that "existing TNN assumes single-level spectral sparsity, which is misaligned with the multi-level spectral structures prevalent in real-world data".

**Compliance With Llm Reviewing Policy:**

Affirmed.

**Key Questions For Authors:**

See Strengths And Weaknesses

**Limitations:**

See Strengths And Weaknesses

**Strengths And Weaknesses:**

This paper addresses a long-standing and significant challenge by proposing a novel $L_p$-norm approach. This represents a new trajectory for explicitly controlling dual spectral sparsity, effectively bringing the TNN closer to the multi-level spectral structures inherent in real-world data.

My primary concern involves the optimization of the $L_p$-quasi-norm, which is notoriously difficult to solve compared to the classic TNN. The authors should clarify how the accuracy of the solution is guaranteed. Furthermore, the complexity of the algorithm is a concern; the authors need to demonstrate how they have simplified the solver to ensure it is computationally efficient and accessible for implementation.

---

> ### Author Rebuttal · Authors · 2026-03-30
>
> Thank you for your careful reading and for recognizing the significance of modeling dual spectral sparsity. We appreciate your constructive comments regarding optimization reliability and computational complexity.
>
> ---
>
> > **Weakness 1 (Clarification of Optimization):** *My primary concern involves the optimization of the Lq-quasi-norm, which is notoriously difficult to solve compared to the classic TNN. The authors should clarify how the accuracy of the solution is guaranteed.*
>
> We agree that optimizing the proposed non-convex $\ell_p(S_q)$ regularization is substantially more difficult than the classic convex TNN case.  For this reason, our intention is not to claim exact minimization of the non-convex objective. Instead, the proposed solver is designed to compute a stationary point of the original formulation through a reweighted proximal scheme within an ADMM framework.
> To support the reliability of the solution procedure, Appendices E.2 and E.3 provide the full update rules together with empirical convergence behavior and supporting convergence analysis. In this sense, the guarantee we aim for is convergence of the solution procedure to a stationary point of the original non-convex formulation. We will revise the manuscript to make this point clearer in the main text.
>
> > **Weakness 2 (On computational complexity and implementation accessibility):** *Furthermore, the complexity of the algorithm is a concern; the authors need to demonstrate how they have simplified the solver to ensure it is computationally efficient and accessible for implementation.*
>
>  We agree that the solver complexity should be clarified more explicitly. The proposed solver is designed to remain computationally aligned with standard TNN pipelines, rather than introducing a fundamentally different optimization framework.
>
> **(1) Complexity Analysis**
>
> - **Complexity order:** The per-iteration cost is dominated by the transform and slice-wise SVD operations. When the transform along the third mode is implemented using FFT or DCT, the transform cost is $O(d_1 d_2 m \log m)$, and the slice-wise SVD cost is $O(m d_1 d_2 \min(d_1,d_2))$. Therefore, as analyzed in Appendix E.2.2, the overall per-iteration complexity is
>   $O(  d_1 d_2m \log m + m d_1 d_2 \min(d_1,d_2)),$ which is of the *same order as standard TNN*.
> - **Solver simplification:** The implementation follows the same transform-plus-slice-wise-update structure as standard TNN methods. The additional overhead mainly comes from the reweighting step, which only adds a *lower-order cost* compared with the dominant t-SVD operations, rather than from a fundamentally more complex solver architecture.
> - **Practical accessibility:** Because the algorithm preserves the same slice-wise computation pattern as TNN, it can be incorporated into existing pipelines with only modest modification. As noted in Appendix F, more scalable variants could be obtained by using randomized or truncated SVD for further acceleration.
>
> We will revise the manuscript to make these implementation-oriented points more explicit.
>
> **(2) Empirical Support**
>
> To provide additional empirical support with the complexity analysis, we also include preliminary runtime comparisons with TNN on SalinasA under two controlled scaling settings: increasing the number of slices, and increasing the spatial resolution of each slice by interpolation. In both settings, LpSq is compared directly with TNN under the same observation and noise conditions.
>
> See the figures for [preliminary experimental results](https://anonymous.4open.science/r/LpSq-8B38/Scalability.png). Using TNN as the reference baseline, LpSq consistently incurs additional runtime but also delivers a stable reconstruction gain on *SalinasA*. Along the spectral-depth direction, LpSq requires about `1.70x–1.78x` the runtime of TNN, while improving PSNR by about `+1.89` to `+2.88 dB`. Along the spatial-resolution direction, LpSq requires about `1.88x–1.95x` the runtime of TNN, while improving PSNR by about `+1.60` to `+1.99 dB`.
>
> Overall, these preliminary results suggest that, relative to the TNN baseline, the additional cost of LpSq appears to behave like a moderate multiplicative overhead, while being accompanied by a consistent PSNR gain across both scaling studies.
>
> ---
>
> We thank you again for these constructive comments, which helped us clarify both the optimization perspective and the practical computational trade-offs of the proposed method.

---

> > ### Author Rebuttal · Reviewer_snXw · 2026-04-03
> >
> > My concerns on optimization-complexity-experiment support have been fully resolved.

---

> > > ### Author Response · Authors · 2026-04-04
> > >
> > > Thank you for the positive follow-up. We appreciate your acknowledgment that our clarification on optimization, complexity, and experimental support has fully addressed your concerns. We will make these points clearer in the final version.

---

### Decision · Program_Chairs · 2026-04-30

**Decision:**

Accept (regular)

**Comment:**

The paper proposes a tensor Lp-Schatten-q quasi-norm for modeling dual spectral sparsity, supported by theory (minimax bounds), optimization, and experiments. Overall, the reviews are positive (multiple weak-to-strong accepts), recognizing both the novelty of the dual spectral modeling and strong empirical gains. Initial concerns mainly relate to optimization clarity, computational complexity, and presentation. Importantly, all reviewers indicate their concerns are resolved after rebuttal. The area chair agrees with the reviewers' assessment and follows their recommendation.